# EPISODIC MEMORIES GENERATION AND EVALUATION BENCHMARK FOR LARGE LANGUAGE MODELS

**Alexis Huet**[*]**, Zied Ben Houidi**[*,†]**, Dario Rossi**
Huawei Technologies Co., Ltd., Paris, France
`{first(.mid).last}@huawei.com`
[*]Equal contribution; [†]Corresponding author and principal investigator

## ABSTRACT

Episodic memory – the ability to recall specific events grounded in time and space – is a cornerstone of human cognition, enabling not only coherent storytelling, but also planning and decision-making. Despite their remarkable capabilities, Large Language Models (LLMs) lack a robust mechanism for episodic memory: we argue that integrating episodic memory capabilities into LLM is essential for advancing AI towards human-like cognition, increasing their potential to reason consistently and ground their output in real-world episodic events, hence avoiding confabulations. To address this challenge, we introduce a comprehensive framework to model and evaluate LLM episodic memory capabilities. Drawing inspiration from cognitive science, we develop a structured approach to represent episodic events, encapsulating temporal and spatial contexts, involved entities, and detailed descriptions. We synthesize a unique episodic memory benchmark, free from contamination, and release open source code and datasets to assess LLM performance across various recall and episodic reasoning tasks. Our evaluation of state-of-the-art models, including GPT-4 and Claude variants, Llama 3.1, and o1-mini, reveals that even the most advanced LLMs struggle with episodic memory tasks, particularly when dealing with multiple related events or complex spatio-temporal relationships – even in contexts as short as 10k-100k tokens.

## 1 INTRODUCTION

*Episodic memory* – the ability to recall specific events grounded in time and space – is a cornerstone of human cognition. Unlike *semantic memory*, which stores general knowledge, episodic memory is intimately tied to *time*, *space*, and *details* of specific *events* (Tulving et al., 1972; Tulving & Thomson, 1973). Both memories are declarative, activated through *cues* – specific triggers that can bring back other facts (semantic) or a rich recollection of past events (episodic). For instance, hearing the word "birds" can call for "fly"; while hearing "France" might remind someone of their first trip abroad, bringing back details of that entire event.

Episodic memory is not only vital for personal identity and coherent storytelling but also plays a crucial role in planning (Pfeiffer & Foster, 2013; Ólafsdóttir et al., 2015), reasoning (Dusek & Eichenbaum, 1997) and decision-making (Barron et al., 2013). It enables individuals to track the states of entities they care about, on both physical (Burgess et al., 2002) and virtual space (e.g. virtual reality (Cushman et al., 2008), digital folders Benn et al. (2015)). It further enables envisioning future scenarios through mental time travel, a process known as future episodic thinking (Atance & O'Neill, 2001; Schacter & Madore, 2016). The critical role of spatio-temporal processing in episodic memory is exemplified best by the *hippocampus*, a specialized brain region that acts as a cognitive map and spatio-temporal index (O'Keefe & Nadel, 1978; Teyler & DiScenna, 1986), needed for forming (and navigating through) past, present and future episodic memories (Tanaka et al., 2014; Teyler & Rudy, 2007). It is within the hippocampal formation that neuroscience research has identified specialized space and time neurons: *place cells*, which fire when we occupy specific locations (O'Keefe & Dostrovsky, 1971), but also express current, past and future locations (Moser et al., 2015); *grid cells*, which create a coordinate system for spatial navigation (Hafting et al., 2005); and *time cells*, which segment events into distinct temporal sequences (MacDonald et al., 2011).

The above findings highlight the fundamental importance of spatial and temporal context in episodic memory processes, and support the *hippocampal indexing theory* (Teyler & DiScenna, 1986; Teyler & Rudy, 2007), according to which the hippocampus stores compressed representations of neocortical activity patterns, serving as an index to reactivate these patterns during memory recall. Additionally, research on source and reality monitoring (Johnson & Raye, 1981; Johnson et al., 1993; Garrison et al., 2017) shows that memories of actually perceived events typically contain richer spatial and temporal contextual information – as opposed to memories of imagined events, making it easier to discern their origins. Thus, episodic grounding in time and space is what allows us to distinguish between what is real and what is imaginary: everything that happened in our collective agreed-upon space and time is real, and everything else is not. It is this ability to ground memories in specific spatio-temporal contexts (e.g. attributing the source a piece of knowledge came from) that gives episodic memory its power and reliability.

Yet, despite its fundamental significance for human cognition, episodic memory remains underexplored within Large Language Models (LLMs) research. Despite remarkable LLM capabilities, two significant drawbacks related to lack of episodic memory limit their full potential. First, LLMs tend to *hallucinate*, i.e., confabulate information that is coherent yet factually incorrect, i.e. not grounded in episodic reality. Second, while humans engage in many step-by-step trials and errors, continuously updating their episodic memory and knowledge, LLMs are constrained by their inability to retain information beyond their context window. This transient nature of LLMs' memory contrasts sharply with human episodic memory, which allows for the long-term storage and retrieval of detailed experiences across an entire project or even lifespan. We argue that integration of episodic memory into LLMs held potential to significantly enhance their reasoning, consistency, and factual accuracy.

In this paper, as a first step to assess this gap, we propose a comprehensive framework to model and evaluate episodic memory capabilities in LLMs. To the best of our knowledge, this aspect has not been previously evaluated: although more and more challenging tasks for long context comprehension are emerging, existing approaches to extending the memory capabilities of LLMs (such as retrieval-augmented generation, or various in-context extension methods), are still tested against relatively simple benchmarks (see Sec. 2). Diverging from such benchmarks, we specifically target episodic memory, with *events* rich of *contextual* information, and involving specific *entities* happens at specific *time* and *space* locations. We synthesize a unique episodic memory benchmark, free from contamination, that ensures coherence and control over the generated narratives and their corresponding ground truth answers. This benchmark not only evaluates how well current LLMs handle episodic tasks, but also lays the groundwork for future research aimed at incorporating dynamic, context-sensitive episodic memory into AI systems. Our contributions include:

**Modeling episodic memory.** Drawing inspiration from cognitive science, we develop a structured approach to model episodic events within LLMs, encapsulating temporal and spatial contexts, involved entities, and detailed event descriptions.

**Benchmark code and dataset.** We introduce a framework for generating synthetic episodic memory datasets[1] comprising narratives of events and corresponding question-answer pairs– that could also be used to generate synthetic tasks for training purposes. We further release 11 datasets[2], differing in size and diversity, to evaluate LLM performance across various episodic memory tasks.

**Assessing LLM performance.** We evaluate various LLMs (GPT-4, Claude, Llama 3.1, and o1-mini), assessing their performance under different configurations: in-context learning, retrieval-augmented generation and fine-tuning. We demonstrate that even the most advanced LLMs struggle with episodic memory tasks, particularly when dealing with multiple related events or complex spatio-temporal relationships, even for very small context size – confirming the need for episodic memory benchmarks like the one we propose to further improve LLM performance. We further showcase the inadequacy of existing fine-tuning strategies for embedding episodic knowledge into LLMs, demonstrating the need for novel training methodologies tailored to episodic memory integration.

## 2 RELATED WORK

For brevity, we summarize here and refer the reader to Appendix A for a broader overview.

---

[1]Code and data available at Huet et al. (2025).
[2]List available in Tab. 28.

**Assessing memory in human subjects.** Existing tests of episodic memory in humans often involve asking the person to recall specific events in their life. The Autobiographical Memory Interview (AMI) (Kopelman, 1994) and the Autobiographical Interview (Levine et al., 2002) are widely used tests that assess recall of specific past events, which are then scored based on the level of detail and accuracy. These tests employ structured interviews that classify memories across dimensions including time periods, locations, and specific event details, which directly informs our approach of using cues related to time, space, entities, and event contents. Other episodic memory tests (Wilson et al., 1985; Delis et al., 2000) can detect impairments at various levels such as encoding, retrieval, and recognition of familiarity – which we take into account in our benchmark definition.

**Assessing memory in LLMs.** Existing benchmarks primarily focus on simple *retrieval* or *reasoning* over long contexts, with limited assessment of episodic memory capabilities (Vodrahalli et al., 2024).

*Retrieval-oriented benchmarks* follow a "needle-in-a-haystack" paradigm (Kamradt, 2023), requiring models to find specific pieces of information within extensive (and highly irrelevant) contexts, failing to evaluate the model's understanding of temporal sequences or state changes. Extensions to *multiple needles* (Reid et al., 2024; Hsieh et al., 2024; Li et al., 2024; Zhang et al., 2024) still lack cue differentiation, and do not incorporate temporal nor spatial awareness. Similarly, *question answering* benchmarks on long contexts (Bohnet et al., 2024; Zhang et al., 2024) do not probe model ability to track entity states or temporal relationships. Other synthetic datasets for *reasoning tasks* such as the bAbI (Weston et al., 2015) (and its long-context extension bAbILong (Kuratov et al., 2024)) or Li et al. (2024), often involve highly artificial scenarios lacking complexity and realism – opening the door to shortcut reasoning by exploiting dataset biases or patterns.

*Other spatio-temporal benchmarks* have been created for temporal question answering (Jia et al., 2018; Saxena et al., 2021; Dhingra et al., 2022; Chen et al., 2021; Kasai et al., 2024; Tan et al., 2023). Compared to our work, the point of view adopted is information retrieval instead of episodic memory analysis, hence the main differences are: (i) the cue refers to a single fact, instead of a trace of events; (ii) there is no systematic approach in the different dimensions of time, space, entity or content; (iii) they are based on limited existing data sources that are already public, usually the Freebase (Bollacker et al., 2008) or the Wikidata knowledge bases; (iv) the output answer is always a single closed answer, while we allow the retrieval of zero or more than one free-form answers. Recently, Vodrahalli et al. (2024) introduced the Latent Structure Queries (LSQ) framework as first good step to address the limitations of current reasoning over long context benchmarks. Close to our design philosophy, it works by embedding a hidden structure within a large context, which the model must understand to answer queries: for example, an object list is modified throughout a long text, and the model must track these changes to determine the list final state, which probes LLM ability to track state across the entire context, going beyond simple retrieval. Our work is similar in spirit but broader in scope, systematically incorporating a range of key elements crucial to episodic memory.

**Limits of existing benchmarks.** Summarizing, we identify several limits in current benchmarks: (1) existence of *shortcuts*; (2) prioritization of simple *retrieval over reasoning*; (3) *data leakage*, where training data is contaminated with test data; (4) *out of distribution distractors* where needles are hidden in a completely different context, biasing the retrieval task; (5) *labor-intensive* creation effort, which makes it difficult to scale adapt, or incrementally improve benchmarks.

**Need for an episodic memory benchmark.** The above limitations hinder the evaluation of LLMs episodic memory capabilities. We hence propose a benchmark that adheres to the following guidelines: (1) explicitly *incorporate temporal and spatial contexts*; (2) force models to *track entity states*; (3) use *varied retrieval cues*, based on different combinations of event attributes, mirroring human cue-based recall; (4) *avoid data leakage*, ensuring that the evaluation is free from contamination; and finally (5) *balance complexity and realism*, providing tasks that are both challenging and representative of real-world tasks, without incurring excessive human labor to retain flexibility.

## 3 MODELING EPISODIC MEMORY FOR LANGUAGE MODELS

Inspired by human episodic memory (Tulving et al., 1972), we model episodic memory for LLMs using two key components, namely *entities* and *events*. Our goals are to (i) create appropriate episodic events, (ii) design systematic memory tasks, and (iii) develop methods to evaluate the ability of LLMs to recall these events accurately. Similarly to human memory tests, we construct tasks and scenarios

for LLMs to assess their ability to (i) recall specific episodes with details, (ii) understand entity states, (iii) comprehend temporal and spatial contexts, and (iv) avoid confabulation.

## 3.1 ENTITIES, EPISODIC EVENTS AND WORLD MODELING

**Entities.** Entities ($ent_j$) are fundamental subjects in the world, that can participate in or be affected by events: their attributes or relationships may change over time due to episodic events. More generally, each entity has an associated state $state_{j,t}$ at any discrete time $t$. This state evolves based on events and includes all details about the entity at a given time, such as its location and any observation such as actions or interactions. The latest state of an entity is denoted as $state_{j,L}$, where $L$ denotes the latest point in time.

**Episodic events.** Episodic events ($event_i$) transcend the scope of the current benchmark, and are defined as actions or observations that lead to changes in the state of the world or its entities, including the mere progression of time. Specifically, each event $event_i$ is characterized by a tuple $(t_i, s_i, ent_i, c_i)$, where $t_i$ represents the time at which the event occurs, $s_i$ the location where the event takes place, $ent_i$ the set of entities involved in the event, and finally $c_i$ corresponds to the event content, detailing what happened. This formulation captures the essential elements of episodic memory: what happened, where and when.

**World model.** This model of episodic events and entities aligns with the hippocampus's role in maintaining a dynamic cognitive map of the world. By tracking the states of entities across time and space, our framework mirrors the brain's process of continuously updating its representation of the environment based on new experiences. This approach not only captures the context, space and temporal properties of events, but also how these events transform our understanding of the world and its constituents. In line with the encoding specificity principle (Tulving & Thomson, 1973), each event is associated with specific details that differentiate it from others: parts of these details will serve as *retrieval cues* during memory tasks. Now in LLMs, these episodic memory components are represented as sentences or paragraphs describing the events, and entity states are inferred from the context provided in the text.

## 3.2 SYSTEMATIC TASK DESIGN: CUE-BASED RECALL AND RETRIEVAL

We model cue-based recall as a key-value retrieval system, where the cue (key) is any combination of elements from the event tuple $(t_i, s_i, ent_i, c_i)$, and the associated event details serve as the memory trace (value). By systematically varying the cues, we can assess the model's ability to retrieve specific information based on different aspects of the events.

memory —— cue *(\*,s,\*,\*)* —→ events —— trace *Entities* —→ [recalled list]

Figure 1: Memory recall process implemented in the benchmark: a set of events match a given cue, from which a list of elements are recalled (here, all entities that have been seen in a given space).

**Cue composition and retrieval types.** To create a comprehensive set of tasks, we consider all possible combinations of the event tuple elements as cues. For instance, a cue of $(t, *, *, *)$ would prompt the retrieval of events that occurred at a specific time $t$, while a cue of $(*, s, *, *)$, as shown in Fig. 1, would query for events that took place at a particular location $s$. Tab. 1 presents example combinations of different cue compositions, the descriptions of the tasks, the types of information to be retrieved, showing a few templated questions from our actual implementation for each case (the full list is deferred to Tab. 10 in the Appendix; example question/answer pairs from the templates are shown in Appendix C).

**Encoding specificity principle and cue overload.** According to the encoding specificity principle (Tulving & Thomson, 1973), the effectiveness of a retrieval cue depends on its similarity to the original encoding context. Specific cues that closely match the encoded event lead to precise retrieval, while broader cues may result in interference with other events sharing similar features, a phenomenon known as cue overload. By varying the specificity of the cues as exemplified in Tab. 1, we test the model's ability to handle both precise and ambiguous queries. For example, a highly

Table 1: Episodic memory questions based on cue composition and retrieval types (full list in Tab. 10).

| Cue | Description | Retrieved trace (id) | Template question (corresponding to ⋆) |
|---|---|---|---|
| $(t, *, *, *)$ | Events at a specific time | - Spaces (0)
- Entities (1) ⋆
- Contents (2) | ⋆ Consider all events that happened on {t}. Provide a list of all protagonists involved in any of these events, without describing the events themselves. |
| $(*, s, ent, *)$ | Events involving entities at a specific location | - Times (18)
- Contents (19) ⋆ | ⋆ Reflect on {ent}'s experiences at {s}. Describe all the key events they've been involved in at this location, focusing on what happened rather than when it occurred. |
| $(*, s, ent, c)$ | Events with specific location, entities, and content | - Times (27) ⋆ | ⋆ Consider all events involving both {ent} and {c} at {s}. Provide a list of all dates when these events occurred, without describing the events. |
| $(t, s, ent, c)$ | Events with specific time, location, entities, and content | - Full event details (29) ⋆ | ⋆ Provide a comprehensive account of what happened involving {ent} and {c} at {s} on {t}. Include all relevant details about the event(s), including what occurred and any other pertinent information. |
| $(*, *, ent, *)$ | Retrieves the most recent known location of an entity | - Times [latest] (30)
- Spaces [latest] (31) ⋆
- Contents [latest] (32) | ⋆ What is the most recent location where {ent} was observed in the story's chronological timeline? |
| $(*, *, ent, *)$ | Retrieves a chronological list of dates when an entity was observed | - Times [chrono.] (33) ⋆
- Spaces [chrono.] (34)
- Contents [chrono.] (35) | ⋆ Provide a chronological list of all dates when {ent} was observed, from earliest to latest in the story's timeline. |

specific cue like $(t, s, ent, c)$ should retrieve a unique event with detailed information, while a broader cue like $(*, s, *, *)$ may retrieve multiple events, requiring the model to list all relevant information without confusion. Note that several *retrieval types* (different spaces, entities, details or dates) can match a given cue: we assess the models' ability to retrieve them separately.

**Assessing entity state and chronological tracking capabilities.** Beyond recalling events and their attributes, we design tasks that require tracking the states of entities over time: the last two rows in Tab. 1 outline tasks focused on entity state tracking and chronological analysis, assessing the model's ability to understand the temporal progression and current state of entities.

**Assessing confabulation.** Finally, we evaluate confabulation including tasks that assess unfamiliarity awareness. This involves testing the model's ability to recognize when it lacks information about certain events or entities and to respond appropriately, such as indicating uncertainty or acknowledging the absence of relevant information – or fail doing so by hallucinating an answer.

## 4 BENCHMARK DESIGN

In this section, we detail the design of our episodic memory benchmark, adhering to the requirements outlined in Sec. 3. The benchmark comprises three key components: (i) the memories to encode, represented by evidence documents, (ii) a set of question-answer pairs designed to probe episodic memory, and (iii) an evaluation strategy to assess model performance.

We introduce a novel methodology that generates synthetic *documents* using a Large Language Model (LLM), structured as a coherent narrative akin to a real *book*. Each *chapter* presents a logical flow and progression of a story while while maintaining *controlled ground truth* information, which is strategically distributed over several *paragraphs*.

This approach distinguishes our benchmark from previous designs, such as the bAbI tasks (Weston et al., 2015), which use a world model but lack coherent storytelling, and from real books, where ground truth information cannot be controlled and data contamination is a concern. The questions in our benchmark are designed to follow the episodic memory recall process highlighted in Fig. 1: each question is based on a cue that triggers the retrieval of relevant events, from which specific information is extracted.

### 4.1 BUILDING THE MEMORIES TO ENCODE

We begin by constructing a static *universe* comprising a finite $N_{universe} = 100$ set of dates ($t$), locations ($s$), entities (*ent*), and event contents ($c$). These elements are carefully curated to ensure diversity and uniqueness. From this universe, we sample $N_{events}$ synthetic *events*, each serving as the foundation for

a single chapter in our book. Each event is characterized by a specific date, location, action, and entity, ensuring a unity of time and place, as well as a main content and entity. Fig. 2 schematizes the book creation process. Full details, along with an example of this process, are provided in Appendix B.

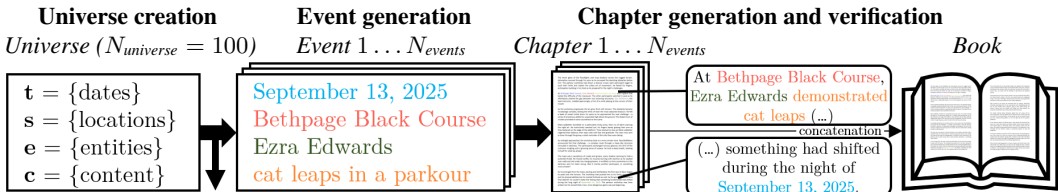

**Event generation**: sampling $N_{events}$ independent events, with each item of each event following a truncated geometric distribution, ensuring, e.g., that the item Ezra Edwards ($e_1$) matches many events:

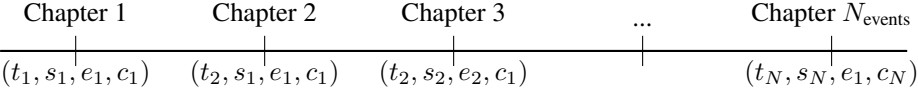

Figure 2: *Book generation*: skewed event sampling (Appendix B.1.3), LLM-based chapter generation with quality control (Appendix B.1.6, B.1.7), and chapter concatenation.

To evaluate model performance across various scenarios, we vary the number of matching events for a given cue from zero to more than six (details and quantitative analysis in Appendix B.1.3). To achieve this variety, we employ a geometric sampling strategy when selecting events from the universe. Specifically, we use a truncated geometric distribution to sample dates, locations, entities, and event contents. This approach ensures that some items appear multiple times across different chapters, while others appear only once. By controlling the frequency of each item, we create cues that correspond to varying numbers of events: this allows us to assess the models' ability to handle both rare and frequent occurrences, as well as their capacity to manage and retrieve multiple related events. To better illustrate this shared universe structure, we provide in Appendix C an example representing the tracking of a single entity for one of the generated document.

To maintain control over the placement of ground truth information within each chapter, we associate each event with an *event meta-data*, which specifies the number of paragraphs in the chapter and the positions of key information within those paragraphs. For example, the date might be required to appear in paragraph three and only in that paragraph. Notably, dates, locations, entities, and event contents may be reused across different chapters.

Each chapter is generated independently using an LLM, guided by the event and event meta-data information. During generation, the LLM may introduce additional entities interacting with the main entity; these new entities are assigned unique names to ensure consistency. Following generation, chapters are concatenated into a single document, resulting in a synthetic book that contains the memory to be encoded. To ensure coherence and validity of the generated document, we implement several verification steps:

**Uniqueness constraints**: we prevent multiple chapters from sharing identical time-space or time-entity pairs to avoid conflicts in the narrative timeline.

**Adherence to event meta-data**: direct parsing verifies that each generated chapter adheres to the meta-data's event requirements, including the placement of key elements in designated paragraphs.

**Additional quality-control layers**: we check the presence verbatim of the date, location, entity and content detail, while an LLM-based verification using boolean questions validates the chapter's unique temporal day, geographical focus, main character and main event (details in Appendix B.1.6, B.1.7).

We acknowledge that, following our generic episodic event definition (Sec. 3), the resulting book may contain *more episodic events* than those explicitly included in our ground truth. However, this does not influence our ground truth answers and questions, as we focus on the events over which we have full control. Additional statistics regarding the distribution of information within the book, each chapter, and each paragraph, including comparisons between GPT-4o and Claude, are provided in Appendix B.1.12. An example of a generated chapter is also included in the appendix for reference.

Table 2: Characteristic of the main benchmarks (both produced with Claude 3.5 Sonnet 2024-06-20). Additional benchmarks are available in Appendix G.

| Parameter | Short book | Long book |
|---|---|---|
| $N_{events}$ (Chapters) | 20 | 200 |
| Nb. of tokens | $\approx$10k (10397) | $\approx$100k (102870) |
| Nb. of dates, locations, entities, contents | 14, 12, 13, 12 | 37, 35, 34, 34 |
| Start – end dates | March 23, 2024 – December 26, 2026 | |
| Nb. of selected QA pairs | 456 | 686 |
| Nb. of QA related to 0, 1, 2, 3-5, 6+ events | 180, 180, 72, 24, 0 | 180, 180, 108, 128, 90 |

## 4.2 BUILDING THE QUESTION-ANSWER PAIRS

We implement a template-based approach to create questions aligned with our episodic memory tasks. Each question is defined by a *cue composition* (the trigger key identifying a set of events), a *trace* (the type of information to be retrieved from each remembered event), and the retrieval mode (whether *all* elements are needed, only the *latest state*, or their *chronological order*), as exemplified in Fig. 1 and detailed fully in Appendix B.2.

We consider all combinations of cues and traces for the retrieval of *all* elements, focusing on entity cues for the *latest* and *chronological* retrievals. Examples of question templates are provided in Tab. 1 (full list available in Tab. 10 in the Appendix). The templates are populated using the dates, locations, entities, and event contents appearing in each chapter. All questions are associated with known ground truth answers, as the event details are controlled and known (examples of question/answer pairs are shown in Appendix C). To test for hallucinations, we include additional questions with empty answers, using entities or combinations of items *that do not exist* in the document. This allows us to assess the model's ability to handle unfamiliarity and avoid confabulation.

To ensure variety in the number of events to be recalled, we filter the corpus of questions to balance the number of queries that correspond to zero, one, two, three to five, and more than five events. More details on this process are provided in Appendix B.2.4.

## 4.3 EVALUATION STRATEGY

Our evaluation strategy employs an LLM-as-a-judge approach to assess the correctness of the model's answers. The evaluator LLM is prompted to perform two key tasks: First, it *(i) identifies relevant items* by extracting them as a list from the AI-generated answer for each question, allowing us to evaluate the number of predicted items. Second, it *(ii) scores the relevance* of these predicted answers against each ground truth item, with the sum of the scores interpreted as the number of true positives. We defer details to Appendix B.3, but in essence, we are using the LLM for simple semantic comparisons, and not as a judge making subjective assessments. Then, from these predicted and ground truth answers, we compute an optimistic F1-score bound (our primary comparison metric) using a lenient methodology detailed in Appendix B.3.2. For chronological questions, we additionally use Kendall's $\tau$ coefficient: this is applied only to answers that fully match the ground truth, allowing us to assess the correctness of the ordering within this matching set.

## 4.4 GENERATED BENCHMARK

Using the proposed world modeling framework, we generate two synthetic documents, referred to as the *short book* and the *long book*, summarized in Tab. 2. The long book includes 196 unique events across 37 dates, 35 locations, 34 entities and 34 event contents. We generate a total of 686 questions, balanced across cue compositions and retrieval types, to evaluate the model's episodic memory capabilities; the complexity of the questions is also controlled by varying the number of related events. Note that we explicitly limit the size of the book, since as we shall see, current state of the art LLMs start struggling with a relatively modest size[3].

---

[3]To demonstrate scalability, we generate an additional 2000-chapter book (1M+ tokens), available at Huet et al. (2025). We also produce and evaluate more universes and benchmarks in Appendix G.

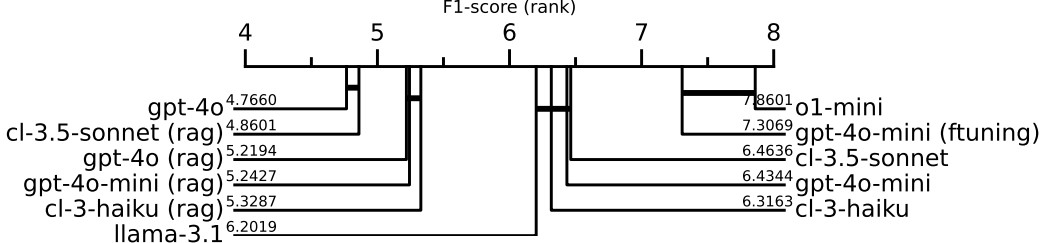

Figure 3: *Overall performance comparison*: Critical distance plot ranking all LLM models and memory combinations (instances not tied by an horizontal bar are statistically different).

# 5 BASELINE RESULTS

## 5.1 BASELINE MODELS AND MEMORY STRATEGIES

As baseline models for our benchmark, we evaluate several LLMs, including GPT-4o, GPT-4o-mini, Claude 3 Haiku, Claude 3.5 Sonnet, Llama 3.1 405B (instruct) and the recent o1-mini. We consider memory in LLMs as functioning through three primary forms: (1) *In-Context Memory*, where information is processed within the model's context window; (2) *Retrieval-Augmented Generation (RAG)*, where external memory is accessed through a vector database; and (3) *Parametric Memory via Fine-Tuning*, where memory is stored within the model's parameters.

For (1) in-context memory, we prepend the full document to the question, allowing the model to process the entire context. With (2) RAG, we chunk the book into paragraphs[4], each labeled with context (e.g., "Chapter X, Paragraph Y"), and embed them using *text-embedding-3-small*. For each question, we retrieve the top-K paragraphs based on cosine similarity to the question's embedding and prepend them to the question as context. Lastly, (3) we fine-tune models[5] using *all single-event* question-answer pairs as training data (details in Appendix B.2.5). In principle, this approach enables models to acquire knowledge for all benchmark questions (including multi-event questions).

## 5.2 EXPERIMENTAL RESULTS

We evaluate the performance of various models and memory strategies on our benchmark. We mostly report results on the long book and defer an extended set of results to Appendix E.

**Overall performance comparison.** Using the F1-score as our primary metric, we compare models and strategies across all questions. Fig. 3 shows the Critical Difference (CD) plot based on the Wilcoxon signed-rank test (Benavoli et al., 2016) between each pair of algorithm (adjusted by Holm's method) that allows to rank models according to their average performance on the long book in a principled manner. GPT-4o with in-context memory and Claude 3.5 Sonnet[6] with RAG memory achieve the highest average ranks, with no statistically significant difference between them. Notably, except for GPT-4o, models utilizing RAG generally outperform their in-context counterparts, suggesting that retrieval methods can enhance episodic memory capabilities by effectively narrowing down the relevant context for each query. As information spans several paragraphs, retrieval granularity (i.e., paragraph vs chapter) may play an important role (see ablation study in Appendix E.2).

**Performance on recall tasks.** We next test the ability of recalling episodic memory, by reporting in Tab. 3 the average F1-scores for simple recall questions as a function of the number of events that match the cue (extended analysis in Appendix E.1).

*Avoiding confabulation.* The 150 questions with 0 matching events are intentionally designed to test familiarity awareness. We see that no model achieves a perfect F1-score in avoiding hallucinations:

---

[4]Since a single event spans across multiple paragraphs, this RAG strategy limits LLM ability to retrieve all relevant information: Appendix E.2, reports a comparison with chapter-based RAG as an ideal upper bound.

[5]Fine-tuning using the OpenAI API over 30 epochs, a batch size of 64 and a learning rate multiplier of 1.8.

[6]Noteworthy, Claude 3.5 Sonnet with in-context memory is knowingly affected by a bug making it verbosely debate finetuning instructions adbertram (2023), cfr. Appendix F.

Table 3: *Performance on recall tasks*: average and standard deviation F1-score as a function of the number of events matching a given cue (long book).

| Memory | Model | Number of events matching the cues | | | | |
|---|---|---|---|---|---|---|
| | | 0 (150) | 1 (150) | 2 (90) | 3-5 (98) | 6+ (60) |
| In-context | gpt-4o-mini | 0.51±0.50 | 0.54±0.46 | 0.44±0.36 | 0.47±0.27 | 0.50±0.17 |
| | gpt-4o | 0.84±0.37 | 0.81±0.38 | **0.60±0.31** | 0.57±0.21 | 0.53±0.14 |
| | claude-3-haiku | 0.84±0.37 | 0.39±0.48 | 0.37±0.30 | 0.37±0.28 | 0.38±0.19 |
| | claude-3-5-sonnet | 0.92±0.27 | 0.35±0.48 | 0.35±0.33 | 0.32±0.25 | 0.41±0.20 |
| | o1-mini | **0.97±0.16** | 0.05±0.19 | 0.12±0.24 | 0.12±0.19 | 0.24±0.19 |
| | llama-3.1-405b | 0.80±0.40 | 0.49±0.47 | 0.38±0.33 | 0.40±0.25 | 0.45±0.20 |
| RAG | gpt-4o-mini | 0.63±0.49 | 0.60±0.46 | 0.60±0.34 | **0.59±0.26** | **0.62±0.22** |
| | gpt-4o | 0.82±0.39 | 0.60±0.46 | 0.55±0.33 | 0.55±0.28 | 0.59±0.21 |
| | claude-3-haiku | 0.71±0.45 | 0.57±0.47 | 0.59±0.33 | 0.58±0.26 | 0.59±0.25 |
| | claude-3-5-sonnet | 0.91±0.28 | 0.59±0.47 | 0.59±0.35 | 0.59±0.27 | 0.62±0.25 |
| Fine-tuning | gpt-4o-mini | 0.00±0.00 | **0.83±0.35** | 0.37±0.32 | 0.28±0.21 | 0.19±0.07 |

o1-mini performs the best, RAG only minimally helps, smaller models struggle to avoid confabulation, and the fine-tuned model consistently fails, highlighting a critical limitation of naive fine-tuning.

*Impact of cue specificity and cue overload.* Considering questions with *a single* existing ground truth trace in Tab. 3, we observe a different scenario: fine-tuned GPT-4o-mini leads (F1 of 0.83) by overfitting on this type of question, with GPT-4o a close second (0.81). However, as the number of matching events increases, the cue is less specific and becomes overloaded: i.e., for questions with *two or more* events matching the cue, we observe a consistent decline in performance for all models (F1 $\leq$0.60). For the fine-tuned model, the decrease is more pronounced (F1 $\leq$0.37): this underscores the inability of naive finetuning to generalize beyond single-event memorization (i.e. models memorize specific answers, without developing a deeper understanding of the information).

*Impact of context size.* Similar considerations hold for the small book (deferred to Appendix E.1), where in spite of very limited context (10k tokens), performance are better but still suboptimal. Noteworthy, model rank differs, with o1-mini and GPT-4o statistically equivalent.

**Impact of cue type.** We next analyze the impact of cue type in Fig. 4 across all models (full details for GPT-4o are reported in Tab. 16 in the Appendix). For different models and number of events in the ground truth, the picture reports the F1 score for different types of cues – specifically, from top to bottom context $(*, *, *, c)$, entity $(*, *, ent, *)$, space $(*, s, *, *)$, and time $(t, *, *, *)$. Across models, gradient is clearly visible from left to right (performance degrades for increasing number of events, as already seen in Tab. 3) and from top to bottom (performance degrades, from context, to space, to time). This interesting finding reinforces the need of benchmarks such as the one we propose.

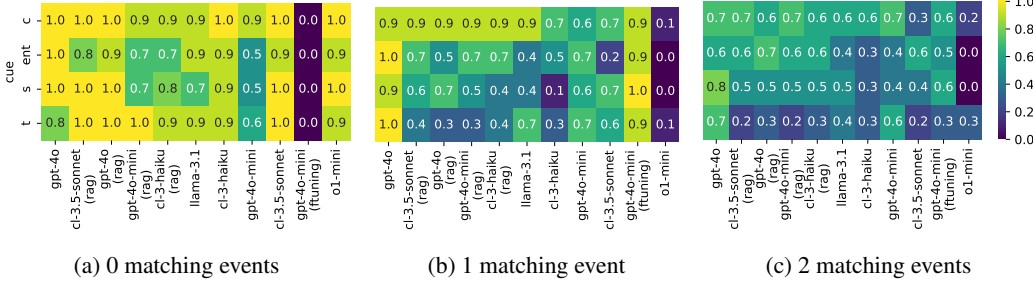

(a) 0 matching events    (b) 1 matching event    (c) 2 matching events

Figure 4: *Impact of cue type.* F1-score across different cue types (y-axis) for models ordered according to their overall rank in Fig. 3 (x-axis) and for increasing number of events that match the cue (from left to right sub-plots).

**Latest state recall and chronological ordering.** We finally study model ability to harness state recall and chronological ordering, at a finer grain. As we early have shown, (some) models perform almost

perfectly when entities participate in zero or one event, but (most) models struggle significantly with multiple events. In reason of LLM performance degradation for questions involving a larger number of events, we expect these questions to be particularly challenging and, in particular, we expect chronological ordering to be significantly more involved than latest state recall. Tab. 4 considers all queries involving more than one event, and shows the (i) fraction of exactly matched latest states, (ii) the fraction of exact matches between the predicted set and the whole ground truth set (possibly in a different order) and (iii) the Kendall's $\tau$ computed across the ground truth and the matched states for those exact matches. Results confirm the expectations, and highlight significant challenges in tracking entity states over time ($\leq 36\%$ for any model) and, especially, to recall all the events ($\leq 18\%$). The proportion of exact matches in chronological ordering tasks is low across all models, and even when models retrieve correct events, they often fail to order them correctly (low Kendall's $\tau$ coefficients). These findings reveal significant gaps in the models' abilities to understand and process temporal relationships between events, motivating the need for future solutions to address this gap.

Table 4: *Latest state recall and chronological ordering.* Considering all questions involving at least two events (i.e., with ground truth answer of length $\geq 2$), we report the fraction of answers that match the *latest* state, *all* the states and the Kendall's $\tau$ coefficient (computed between the ground truth and the matched predictions subset). Models are ranked according by decreasing chronological ability.

| Family | Claude | GPT | Claude | GPT | GPT | GPT | Claude | Claude | llama | o1 | GPT |
|---|---|---|---|---|---|---|---|---|---|---|---|
| **Model** | 3-haiku | 4o-mini | 3.5-son. | 4o | 4o | 4o-mini | 3-haiku | 3.5-son. | 3.1-405b | mini | 4o-mini |
| **Memory** | rag | rag | rag | rag | context | context | context | context | context | context | ftuning |
| **Latest** | 23% | 36% | 31% | 23% | 36% | 13% | 19% | 18% | 26% | 5% | 23% |
| **All** | 18% | 13% | 13% | 10% | 10% | 8% | 5% | 3% | 0% | 0% | 0% |
| **Kendall $\tau$** | 0.43 | 0.93 | 0.60 | 0.50 | 0.50 | 0.33 | 1.00 | 1.00 | n.a. | n.a. | n.a. |

**Other ablation studies.** We defer to the Appendix E detailed analyses of performance with respect to: book size (E.1), RAG granularity (E.2), number (E.3) and type (E.4) of cues and traces, book generation process (E.5), chapter chronological ordering (E.6), and event plausibility (E.7).

# 6 SUMMARY AND LIMITATIONS

In this work, we draw inspiration from cognitive science to build a new episodic memory model for LLMs, generate a comprehensive benchmark, and evaluate state-of-the-art models. Our findings reveal significant gaps in the episodic memory capabilities of state-of-the-art LLMs, particularly when handling multiple related events and complex spatio-temporal relationships. These challenges reflect aspects of human memory where more distinctive cues facilitate easier retrieval. Furthermore, naive finetuning fails to achieve a deep understanding of episodic events and their intricate relations, merely overfitting to single learned facts. These challenges highlight the need for fundamentally new approaches to model design and training that more closely emulate the dynamic and contextual nature of the human episodic memory. The proposed episodic memory benchmark exhibits several desirable properties: it is contamination-free by design, scalable with low human labor, offers unambiguous cues and ground truth, and the ability to model multiple cues and events within a synthetic yet realistic narrative. However, we acknowledge limitations that open avenues for future research.

**Temporal representation.** Our benchmark relies on explicit temporal markers, which may not fully capture the nuanced ways time is expressed in natural language (e.g. "yesterday", "last week", or "after the party"). Future iterations should incorporate implicit and relative temporal references to further challenge the models.

**Event independence.** The independent generation of chapters, while facilitating control, does not capture the interconnected and causal nature of real-world events.

**Limited domain scope.** Our benchmark primarily involves human-like protagonists within fictional contexts. Extending the framework to include diverse domains (e.g., software projects, virtual environments) would test models' ability to generalize episodic memory capabilities.

**Training limitations.** The observed performance limitations suggest that current fine-tuning methodologies may not be optimally suited for episodic memory tasks, underscoring the need for developing new strategies for the broad scientific community.

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

## A    EXTENDED BACKGROUND AND RELATED WORK

In this section, we first draw useful parallels between LLMs and human declarative memories that will guide us in building our episodic memory benchmark for LLMs. We then briefly overview the landscape of memory benchmarks, showing how none captures the intricacies of episodic memory.

### A.1    SEMANTIC AND EPISODIC MEMORIES: PARALLELS BETWEEN HUMAN COGNITION AND LLMs

"Think of a particular cat you have seen in your previous life". The previous sentence acted as a *cue* that reminded you of an episode from your past, a memory trace that you recollected with the aid of the cue. This process, known as cue-based recall, is fundamental in cognitive psychology and involves retrieving stored information when provided with specific cues or prompts. Interestingly, this process bears a striking resemblance to how LLMs retrieve new text from their inner representations in response to prompts.

Cue-based recall is at the heart of explicit declarative memory, or the conscious recognition and recollection of facts, events, and experiences. Cognitive psychologist Endel Tulving pioneered studies on declarative memory and was the first to identify what is now accepted to be its two main components: semantic and episodic memories (Tulving et al., 1972).

*Semantic memory* refers to our general knowledge and understanding of concepts that are independent of personal experiences. It encompasses facts, language, and principles that are universally applicable. For example, knowing that cats have four legs or that water consists of hydrogen and oxygen are manifestations of semantic memory. Tulving argued that semantic memory is not tied to a specific time or place but serves as a reservoir of factual information accessible across various contexts.

*Episodic memory*, on the other hand, is intimately connected to specific events. It involves the recollection of past episodes or the ability to imagine future events, also known as future episodic thinking. Episodic memory encompasses the recall of what happened as well as the *spatial* and *temporal* context in which the event occurred and the memory trace. This subjective sense of time and place contributes to our ability to mentally time travel and re-experience past events.

While recent research suggests that the distinction between both memories is not as clear-cut as previously thought in terms of neural correlates (De Brigard et al., 2022), both types of memory share common underlying processes. This fluidity in the structural difference between semantic and episodic memory is increasingly viewed in the literature as an interconnected continuum rather than discrete, separate entities needing different learning processes.

Interesting parallels can be drawn between cue-based recall in humans and prompt-based generation in LLMs, enlightening our design of an episodic memories benchmark for LLMs. We pinpoint four crucial factors that significantly impact human memory recall and explore their potential extensions to the world of LLMs:

*Encoding and retrieval*: In humans, the encoding process is crucial for later retrieval of memories. For LLMs, this parallels the training phase, where information is initially processed and stored. Our benchmark will explore different encoding methods for LLMs, including in-context learning, fine-tuning, and retrieval-augmented generation (RAG) as baselines.

*Cue/prompt specificity*: Tulving's encoding specificity principle (Tulving & Thomson, 1973) stipulates that the specificity of the context in which information is encoded determines how it can be effectively retrieved. In LLMs, this can be likened to the k-extractability metric (Biderman et al., 2023; Carlini et al., 2021), where longer contexts (i.e., more specific) make it orders of magnitude easier to extract memorized sequences verbatim (Carlini et al., 2022). We will leverage this factor in our benchmark by varying the specificity of cues in our episodic memory tasks.

*Frequency and repetition*: Humans tend to better recall information that they've encountered multiple times (Scarborough et al., 1977). This repetitive exposure reinforces neural pathways, making retrieval more efficient. Similarly, in LLMs, the most frequently present next tokens during the training phase will likely be "retrieved" during inference (McKenna et al., 2023; Kandpal et al., 2022). This can be used to induce memorization. While this aspect is not relevant for our benchmark since we do not develop new encoding methods, this can be leveraged in future work.

*Familiarity* (Mandler, 1980): This rapid feeling of *knowing* that we previously encountered a stimulus, item, or situation, may or may not be followed by a successful *retrieval* of details related to this situation. LLMs, being generative processes, do not natively support such a feature. Augmenting them with this ability can help in assessing when an assertion is factual (e.g., has been really observed in training) or fake.

*Cognitive dissonance* (Festinger, 1962): Described by Festinger in 1962, and slighltly related to familiarity/novelty, cognitive dissonance refers to the discomfort felt when holding two contradictory informations as a new incoming information challenges a pre-existing belief (Festinger, 1962). In this work, we focus as first step on cue-based recall and familiarity leaving the dissonance aspect for future work.

Based on the above parallels, our episodic memory benchmark for LLMs will incorporate the following elements. (i) We assume an encoding phase to create episodic memories in LLMs. While the exact mechanism is left for future work, we test in this paper three naive baselines: in-context learning, fine-tuning and Retrieval Augmented Generation (RAG). (ii) We use prompts/cues to ask the models episodic questions, mimicking cue-based recall in humans. (iii) We design a systematic approach to cue generation, splitting cues into episodically relevant categories (time, space, people, details) and methodically varying cue specificity. (iv) Testing with unfamiliar cues to assess the model's ability to provide negative answers when appropriate.

By leveraging the parallels between human memory and LLM functionality (Tab. 5), we aim to advance the capabilities of LLMs to support solid episodic memories in addition to their general knowledge. Our benchmark will provide a comprehensive framework for evaluating and potentially improving LLMs' ability to encode and retrieve episodic memories.

| Human memory aspect | Equivalent in LLMs? |
|---|---|
| Semantic memory | ✓ |
| Episodic memory | ✓ |
| Encoding | ✓ (training) |
| Cue-based recall | ✓ (prompting) |
| Encoding specificity | ✓ (k-extractability) |
| Frequency/repetition | ✓ |
| Familiarity/knowing | × (Not yet defined) |
| Cognitive dissonance | × (Not yet defined) |

Table 5: Human and LLM memory

## A.2 APPROACHES TO EXTEND LLM MEMORY

Memory in LLMs can be conceptualized in three primary forms: (i) in-context memory, where information is processed within the model's context window, (ii) external memory accessed through vector databases (Retrieval Augmented Generation or RAG), and (iii) parametric memory stored within the model's parameters. These approaches align with the baselines we test in our episodic memory benchmark: placing the episodic information in-context, in a RAG system, or directly in the model parameters. Each method offers unique advantages and challenges for handling episodic-like information in LLMs.

### A.2.1 IN-CONTEXT MEMORY EXTENSION

In-context memory refers to the information an LLM can process within its context window. Efforts to extend this capability focus on increasing the number of tokens an LLM can handle simultaneously. Early approaches aimed to reduce the computational complexity of self-attention. Child et al. (2019) proposed sparse attention methods, restricting the model's focus to subsets of the input. Wang et al. (2020) explored low-rank approximations of attention, while Choromanski et al. (2020) demonstrated kernelized attention, approximating the attention process using kernel functions.

More recent advancements have pushed the boundaries of context length. The Longformer, introduced by Beltagy et al. (2020), combines local windowed self-attention with task-specific global attention, allowing Bert to process up to 4096 tokens. Martins et al. (2021) proposed the Infinity-former, using a continuous-space attention mechanism with radial basis functions. LongNet, developed by Ding et al. (2023), claims to scale up to 1 billion tokens using "dilated" attention. The StreamingLLM approach by Xiao et al. (2023) leverages the "attention sink" phenomenon to process up to 4 million tokens without expensive fine-tuning. Most recently, human-like episodic memory approaches to LLMs (Fountas et al., 2024; Das et al., 2024) have been proposed. We plan to test then on our benchmark as part of our future work.

While these approaches significantly extend the context window, they may not fully capture the persistent nature of episodic memories, as the information is only retained within the current context.

### A.2.2 RETRIEVAL AUGMENTED GENERATION (RAG)

Retrieval Augmented Generation (RAG) approaches extend LLM memory by incorporating external knowledge sources, typically stored in vector databases. This method allows LLMs to access information beyond their parametric knowledge, potentially supporting more extensive and persistent episodic-like memories.

Early work by Khandelwal et al. (2019) proposed nearest-neighbor language models, interpolating a pre-trained neural language model with a k-nearest neighbors model. Lee et al. (2020) formulated knowledge-intensive tasks as phrase retrieval problems, using pre-indexed dense phrase representations. Petroni et al. (2020) demonstrated the effectiveness of combining a shared dense vector index with a sequence-to-sequence model across multiple tasks. Izacard & Grave (2020) proposed "Fusion-in-Decoder" which uses both sparse and dense representations to fetch supportive passages before feeding them to a frozen generative model. More recently, Wang et al. (2023) introduced LongMem, featuring a decoupled network architecture with a frozen backbone LLM acting as a memory encoder and an adaptive residual side-network functioning as a memory retriever and reader.

RAG approaches offer the potential to store and retrieve large amounts of episodic information, but the quality of results depends heavily on the retriever which may struggle with complex tasks that require synthesizing information across multiple sources (similarly to when we have multiple traces which correspond to a single cue in our benchmark).

### A.2.3 PARAMETRIC MEMORY

Parametric memory refers to the knowledge encoded within the model's parameters during training. This subsection explores both the factors influencing parametric memory retention and approaches to edit this knowledge post-training.

Several factors impact memorization and knowledge retention in LLMs' parametric memory. Carlini et al. (2022) found that model scale plays a significant role, with larger models memorizing 2-5 times more than smaller ones. They also noted that data duplication and context length affect memorization, with repeated examples and longer contexts facilitating easier extraction of memorized sequences. Kharitonov et al. (2021) demonstrated that the size of the subword vocabulary influences Transformer models' ability to memorize training data. Carlini et al. (2019) showed that the sampling strategy, particularly the choice between beam search and greedy sampling, can affect the propensity for data leakage and memorization. Kandpal et al. (2022) found that an LLM's ability to answer fact-based questions is significantly influenced by the number of relevant documents seen during pre-training.

Efforts to edit parametric knowledge post-training have also emerged. Meng et al. (2022a) introduced ROME, which uses causal tracing to locate and modify specific associations within the model. The same authors later developed MEMIT (Meng et al., 2022b) in order to scale to much larger edits in bulk. Dai et al. (2021) leveraged the identification of knowledge neurons to perform "knowledge surgery" – editing factual knowledge within Transformers without additional fine-tuning. Zhu et al. (2020) framed knowledge modification as a constrained optimization problem, finding that constrained layer-wise fine-tuning is an effective method for modifying the knowledge that Transformers learn.

While parametric memory offers the potential for integrated, persistent episodic-like information, we are not aware of any approach that allows to ingest episodic events into model parameters, as current knowledge editing is limited in scale to toy cases, and may have unintended side effects.

### A.2.4 HYBRID APPROACHES

Hybrid approaches aim to combine the strengths of multiple memory strategies, potentially offering a more comprehensive solution for handling episodic-like information in LLMs. These methods often integrate aspects of in-context processing, external retrieval, and parametric knowledge. Gupta & Berant (2020) proposed GMAT (Global Memory Augmentation for Transformers), which introduces a dense attention-based global memory to provide a consolidated view of the entire input sequence. This design allows the model to achieve memory overhead that scales linearly with sequence length,

potentially supporting longer-range episodic memories. Wu et al. (2022) introduced the Efficient Memory-Augmented Transformer (EMAT), which encodes external knowledge into a key-value memory and involves novel pre-training tasks. This approach enables the model to learn when to use internal parametric knowledge versus external knowledge, potentially mimicking the interplay between semantic and episodic memory in human cognition. These hybrid approaches offer promising avenues for enhancing LLMs' ability to handle episodic information. By combining multiple memory strategies, they may be able to overcome the limitations of individual approaches and provide a more flexible and robust system for managing episodic-like memories. It would be particularly interesting to test how these hybrid approaches perform on our episodic memory benchmark. Their ability to integrate different forms of memory could potentially lead to improved performance in tasks requiring both factual recall and contextual understanding of events. Future work could involve adapting our benchmark to specifically evaluate these hybrid models, providing insights into their effectiveness for episodic memory tasks.

Finally, the three approaches to LLM memory - in-context extension, RAG, and parametric memory - form the basis for our baseline evaluation of our episodic memory benchmark: we test the model's ability to handle episodic information when it's placed in-context, stored in a RAG system, or encoded directly in the model parameters.

## A.3 Evaluating memory in humans and LLMs

### A.3.1 Human tests of episodic memories

Existing tests of episodic memory in humans often involve asking the person to recall specific events in their life or using standardized psychological tests. For example, the Autobiographical Memory Interview (AMI) (Kopelman, 1994) and the Autobiographical Interview (Levine et al., 2002) are widely used tests that involves asking the person to recall specific events from their past, which are then scored based on the level of detail and accuracy. Rivermead Behavioural Memory Test (RBMT) (Wilson et al., 1985) uses everyday scenario *events* to test a range of memory types, including episodic memory. Another example is the California Verbal Learning Test (CVLT) (Delis et al., 2000), which assesses episodic verbal learning and memory and can detect impairments at any point in the episodic memory process, including encoding, retrieval, and recognition of familiarity. Similarly to these tests, we design tests to probe LLMs in their abilities to recall various dimensions of previously encountered episodic events.

Our approach of using specific cues related to time, space, people, and details aligns to an extent with established human episodic memory tests like the AMI (Kopelman, 1994) and the Autobiographical Interview (Levine et al., 2002). These tests employ structured interviews that classify memories across similar dimensions, including time periods, locations, and specific event details. Note that the Autobiographical Interview for example further categorizes such details into event happenings, perceptual information, and emotions/thoughts. In our work, we instead group all these aspects under the broader category of "event details", though it could be readily extended to systematically analyze different types of details: for example, our chapters are generated with a writing style adhering to different atmospheres (e.g. suspense, tragedy, comedy, etc.), which is akin to an emotional category.

### A.3.2 Benchmarking memory in LLMs

Evaluating memory capabilities in LLMs has garnered significant attention, yet existing benchmarks primarily focus on simple retrieval tasks or reasoning over long contexts without capturing the nuanced aspects of episodic memory. In this section, we review current benchmarks and highlight their limitations in assessing episodic memory.

**Retrieval-focused benchmarks.** Many benchmarks assess LLMs through retrieval tasks, where models are required to find specific pieces of information within extensive contexts. The "needle-in-a-haystack" paradigm (Kamradt, 2023) exemplifies this approach, testing a model's ability to locate a single piece of relevant information within irrelevant textbooks. While this assesses basic retrieval capabilities, it does not evaluate the model's understanding of temporal sequences or state changes.

Extensions to this paradigm involve multiple needle retrieval tasks (Reid et al., 2024; Hsieh et al., 2024; Li et al., 2024; Zhang et al., 2024), which require models to retrieve several pieces of information from a large context. However, these tasks still lack the differentiation of cues and do not incorporate

temporal or spatial awareness. They remain largely retrieval exercises without testing the model's ability to comprehend and recall events as sequences that unfold over time and space.

**Long-context question answering.** Benchmarks like those proposed by (Bohnet et al., 2024) and (Zhang et al., 2024) focus on question answering over long contexts. While these tasks involve processing large amounts of text, they often reduce to retrieving relevant information without necessitating an understanding of event chronology or causality. They do not sufficiently challenge models to track entity states or reason about temporal relationships, which are critical components of episodic memory.

**Synthetic reasoning tasks.** Synthetic datasets such as the bAbI tasks (Weston et al., 2015) and its long-context extension, bAbILong (Kuratov et al., 2024), introduce reasoning tasks that require models to answer questions based on provided stories. However, these tasks often involve short context lengths and highly artificial scenarios. They may allow models to shortcut reasoning by exploiting dataset biases or patterns, thus not effectively evaluating the model's ability to utilize full context in an episodic memory sense.

Similarly, synthetic reasoning tasks proposed by Li et al. (2024) are limited by their artificial nature and may not generalize well to real-world episodic memory scenarios. The lack of complexity and realism in these tasks means they do not adequately test a model's capacity for episodic recall and reasoning.

Recently, Vodrahalli et al. (2024) introduced the Latent Structure Queries (LSQ) framework as first good step to address the limitations of current reasoning over long context benchmarks. Somewhat in essence close to our design philosophy, it works by embedding a hidden structure within a large context, which the model must understand and manipulate to answer queries. For example, a Python list is modified throughout a long text, and the model must track these changes to determine the list's final state. This approach tests the model's ability to track state across the entire context, going beyond simple retrieval. Our approach further systematically incorporates key elements crucial to episodic memory, such as temporal and spatial cues, rich event details, and the ability to handle varied retrieval cues based on partial information.

**Limitations of existing benchmarks.** Overall, as nicely synthesized by Vodrahalli et al. (2024), several limits exist in current benchmarks which could be summarized as follows.

- *Shortcuts*: Models can often find answers without utilizing the full context, undermining the evaluation of their long-term memory capabilities.
- *Retrieval over reasoning*: Tasks prioritize retrieval over reasoning about events, states, and temporal relationships.
- *Data leakage*: Use of existing evaluations can lead to data contamination, where models may have seen the test data during training (Zhang et al., 2024; Agarwal et al., 2024; Bohnet et al., 2024; Hsieh et al., 2024; Li et al., 2024).
- *Out of distribution distractors*: Needle in a hay stack tasks often place the trace to be retrieved in a completely different context, biasing the retrieval task.
- *Labor-intensive creation*: Some benchmarks require significant human effort to create and verify datasets, making it challenging to scale or adapt them for different purposes.

**Need for an episodic memory benchmark.** The limitations of existing benchmarks highlight a gap in evaluating LLMs' episodic memory capabilities, calling for a benchmark that:

- *Incorporates temporal and spatial context*: Evaluates the model's ability to understand and recall events with specific time and location details.
- *Tracks entity states*: Assesses how well the model can monitor changes in entities over time, reflecting real-world dynamics.
- *Uses varied retrieval cues*: Tests the model's ability to recall information based on different combinations of event attributes, mirroring human cue-based recall.
- *Avoids data leakage*: Ensures that the evaluation is free from contamination, providing a fair assessment of the model's capabilities.
- *Balances complexity and realism*: Provides tasks that are both challenging and representative of real-world episodic memory demands without being overly synthetic.

# B  BENCHMARK DESIGN

This section provides a comprehensive description of the benchmark design and complements Sec. 4. It deals with the creation of the full book document (Sec. B.1), the creation of the question/answer pairs (Sec. B.2) and the evaluation strategy (Sec. B.3).

## B.1  BOOK GENERATION

### B.1.1  COMPONENTS OF THE UNIVERSE

We start from a number of atomic components that we can combine to build possible worlds or universes: a pair of start and end dates, a list of 100 first names, last names, locations, event contents, and a list of 30 details for each content. We ensured, when building them, that all the elements are distinct. The Listing 1 provides an excerpt of such components.

```
# temporal
start_date = datetime(2024, 1, 1)
end_date = datetime(2026, 12, 31)

# entities
first_names = ['Emma', 'Liam', 'Olivia', ...]
last_names = ['Smith', 'Johnson', 'Williams', ...]

# locations
locations = ['Empire State Building', 'Statue of Liberty', 'Museum of Modern Art', ...]

# contents
contents = ['Art Exhibition Opening', 'Scientific Conference', 'Tech Product Launch', ...]
content_details = {
  'Art Exhibition Opening': ['Unveiled new collection', 'Met with art critics', ...],
  'Scientific Conference': ['Presented research findings', 'Participated in panel discussion', ...],
  'Tech Product Launch': ['Unveiled new device', 'Demonstrated key features', ...]
}
```

Listing 1: Excerpt of the raw materials

We generate the components of the universe with the help of Claude 3.5 Sonnet. For reference, we provide the process for building the list of locations in the Listing 2. Our general strategy was to first generate twice the number of targeted elements, and then to filter the duplicates and other incorrect elements.

```
# session 1, prompt 1:
Please list 200 different locations in New York and surrounding areas. Each location should correspond to a
    specific (longitude, latitude) point. The different locations should not overlap. Here are some examples:
    'Empire State Building', 'Statue of Liberty', 'Museum of Modern Art (MoMA)', 'Chrysler Building',
    'Fort Greene Park'

# session 1, prompt 2:
Can you keep only the neutral and positive locations, further removing the locations with the name specifying
    a company name

# session 2, prompt 1:
Are those locations all located in different (longitude, latitude) in New York?: [the list of the 200 locations]

# session 2, prompt 2:
Please discard the less distinct or odd ones, in order to keep only 120 different locations among this list

# session 3, prompt 1:
Are those locations all located in different (longitude, latitude) in New York? [the list of the 120 produced
    locations]

# Answer has been 'yes', and we keep the 100 first unique elements among those 120 locations.
```

Listing 2: Process for building the raw locations

### B.1.2 BUILDING A STATIC UNIVERSE

The static universe defines a list of $N_{\text{universe}} = 100$ dates, full names, locations, and contents. The dates are created by sampling days among the start and end raw material range. The full names are created by randomly sampling first and last names. The other items are direct shuffling of the raw materials. The content details are left unchanged. We ensured that there is no duplicated items. An additional seed parameter is used for reproducibility.

```
temporal = ['February 27, 2026', 'May 11, 2026', 'March 23, 2024', ...]
spatial = ['American Museum of Natural History', 'Metropolitan Museum of Art', 'High Line', ...]
entities =  ['Henry Reed', 'Levi Rodriguez', 'Scarlett Thomas', ...]
content = ['Tech Hackathon', 'Theater Performance', 'Educational Workshop', ...]
```
Listing 3: Excerpt of the universe. Each list has $N_{\text{universe}} = 100$ elements.

### B.1.3 EVENTS GENERATION

We define an event as a date, a spatial location, a main entity, an event content, with its details that we would like to insert verbatim into the document. Each event is hence represented by a 5-tuple $(t, s, ent, c, d)$. We provide in Listing 4 the four first events produced using our fixed seed.

```
events[0] = ['September 13, 2025', 'Bethpage Black Course', 'Ezra Edwards', 'Parkour Workshop',
            'Demonstrated cat leaps'],
events[1] = ['September 22, 2026', 'American Museum of Natural History', 'Chloe Castillo', 'Fashion Show',
            'Revealed future collections'],
events[2] = ['September 22, 2026', 'Port Jefferson', 'Henry Reed', 'Photography Exhibition',
            'Explained post–processing techniques'],
events[3] =['May 07, 2024', 'Hither Hills State Park', 'Zoe Brown', 'Karaoke Night',
            'Performed with live band accompaniment']
```
Listing 4: Excerpt of the first four events.

In addition to the seed, two parameters are used for generating the events: the number of events $N_{\text{events}}$ and a distribution $\mathcal{D}$ with finite support $\{0, \dots, N_{\text{universe}} - 1\}$. Each event $i$ is generated by sampling independently a temporal, a spatial, an entity, and a content among the respective lists of the universe and according the same distribution $\mathcal{D}$. Each content detail is then sampled uniformly at random among the content details of the selected content.

Two constraints govern our actual generation. First, (i) we do not allow multiple events to have identical $(t, ent)$ or $(t, s)$ pairs (by initially generating more events before filtering them), since this might imply that the same entity is experiencing two events at the same time, or that two events are happening in the same location at the same time. Second, for convenience of our experiments, (ii) we ensure that the choice of two different number of events $N_{1,\text{events}} < N_{2,\text{events}}$ leads to the same first $N_{1,\text{events}}$ produced events, given the other parameters set equal. In our experiments we used $N_{1,\text{events}} = 20$ and $N_{2,\text{events}} = 200$; allowing us to generate a short and a long book, where the short book's events are included in the long one.

As earlier mentioned, the distribution $\mathcal{D}$ is a critical parameter to ensure that some dates, locations, entities and contents appear in multiple generated events, while others appear only once. This allows us to vary the number of traces or events that correspond to a given cue. We select the truncated geometric distribution with parameter $p = 0.1$, that is, for $i \in \{0, 1, \dots N_{\text{universe}} - 1\}$, $P(X = i) = (1 - p)^i p / \left[ 1 - (1 - p)^{N_{\text{universe}}} \right]$.

Note that choosing a uniform distribution would result instead in a low probability of selecting multiple times the same item (e.g., the same date $t$). This is experimentally confirmed in Tab. 6. In this experiment, we repeat 10000 times the selection of $N_{\text{events}}$ with the distribution $\mathcal{D}$ among a universe of size $N_{\text{universe}}$, and report the expected counts (and standard deviation) in selecting one, two, three to five, or more than six times a specific item. In both cases, we show that the counts are better spread within the different bins for the geometric distribution.

### B.1.4 EVENT META-DATA GENERATION

We associate event meta-data to each event, providing contextual information and constraints regarding the generation of the textual chapter. Each event meta-data indicates the targeted number of

Table 6: Expected counts in selecting one, two, three to five, or more than six times a specific item (e.g., date) for a universe of size $N_{\text{universe}} = 100$. For instance, the value 2 in the bottom-right cell indicates that we expect two dates to be repeated each more than 6 times among the 200 generated events, using the uniform distribution. The actual counts in our experiments is shown in Tab. 9.

| | | Bin of counts | | | |
|---|---|---|---|---|---|
| $N_{\text{events}}$ | $\mathcal{D}$ | 1 | 2 | 3-5 | 6+ |
| 20 | geometric | 8±2.4 | 3±1.5 | 2±0.9 | 0±0.2 |
| 200 | geometric | 9±2.7 | 5±1.9 | 7±2.2 | 13±1.4 |
| 20 | uniform | 16±2.3 | 2±1.1 | 0±0.3 | 0±0.0 |
| 200 | uniform | 27±4.0 | 27±4.5 | 31±3.2 | 2±1.2 |

paragraphs, and the position of the different event features within those paragraphs. Additionally, a desired style is indicated. We plan to use the style in future iterations of the benchmark to generate questions about the general atmosphere of the events, as opposed to specific tokens. We provide in Listing 4 the meta data associated with the four events, produced using our fixed seed.

```
metaevents[0] = {'nb_paragraphs': 7, 'idx_paragraph': {'date': 7, 'location': 2, 'entity': 2, 'content': 2},
                'style': 'thriller'},
metaevents[1] = {'nb_paragraphs': 7, 'idx_paragraph': {'date': 7, 'location': 5, 'entity': 5, 'content': 3},
                'style': 'fantasy'},
metaevents[2] = {'nb_paragraphs': 1, 'idx_paragraph': {'date': 1, 'location': 1, 'entity': 1, 'content': 1},
                'style': 'detective'},
metaevents[3] ={'nb_paragraphs': 5, 'idx_paragraph': {'date': 1, 'location': 3, 'entity': 3, 'content': 4},
                'style': 'mystery'}
```

Listing 5: Excerpt of the first four event meta-data.

For each event, we sample the number of paragraphs uniformly among $\{1, \ldots, 10\}$, and each event feature (among date, location, entity and content) between one and the number of paragraphs. Finally, the style is sampled among the Listing 6, each style being associated to three adjectives. Both styles and the related adjectives have been generated with Claude 3.5 Sonnet.

```
{
    'detective': ['suspense', 'deduction', 'investigation'],
    'comedy': ['humor', 'wit', 'absurdity'],
    'tragedy': ['sorrow', 'catharsis', 'downfall'],
    'romance': ['passion', 'intimacy', 'longing'],
    'thriller': ['excitement', 'danger', 'anticipation'],
    'fantasy': ['magic', 'imagination', 'worldbuilding'],
    'horror': ['fear', 'dread', 'supernatural'],
    'mystery': ['enigma', 'clues', 'revelation']
}
```

Listing 6: List of the possible styles, with corresponding adjectives.

### B.1.5 SINGLE CHAPTER CANDIDATE GENERATION

The template prompt for generating each candidate chapter is described in Listing 7. The template is filled with the right event and its meta-data, before it is fed to an LLM for generation. In our experiments, we used both Claude 3.5 Sonnet (2024-06-20) and GPT-4o (2024-05-13). Only the book built from Claude 3.5 Sonnet has been used in the main paper, but we provide the outputs for both model at Huet et al. (2025), and additional evaluation on the GPT-4o book is provided in Appendix E.5.

```
# System prompt
You are a creative fiction writer specializing in detailed, atmospheric novel excerpts. Your task is to generate
    vivid, immersive scenes based on specific prompts.

# User prompt
Write a detailed novel excerpt in a {style} style about {entity} attending a {content}.
The story takes place on {date}, at {location}, where {entity} {content_single_detail}.
```

Follow these guidelines:

Structure and Information Reveal:
1. Divide the text into {nb_paragraphs} paragraph(s). Number each paragraph (1), (2), etc., while
    maintaining novel–appropriate paragraph lengths.
2. Gradually reveal key information:
– Full location '{location}': must appear verbatim in paragraph {idx_loc} only and nowhere else in the text
– Full date '{date}': must appear verbatim in paragraph {idx_date} only and nowhere else in the text
– Full name '{entity}': must appear verbatim in paragraph {idx_entity} only and nowhere else in the text
– Full detail that '{first_name} {content_single_detail}': must appear verbatim in paragraph {idx_content}
    and nowhere else in the text
3. Subtly distribute details about location, date, main character, and event across all paragraphs.

Content and Setting:
1. Focus on {first_name}'s experiences, observations, and interactions during the {content}.
2. Vividly describe surroundings, atmosphere, and {first_name}'s emotions.
3. Include the detail that {first_name} {content_single_detail}.
4. Limit the timeframe to a single day and confine all action to {location}.

Characters:
1. Refer to other characters as $entity_X (where X is a number).
2. Omit background information about {first_name} and other characters.

Style and Tone:
1. Use vivid, sensory details to bring the scene to life.
2. Incorporate elements of the {style} style, including {style_description}.
3. Maintain a consistent narrative voice throughout the excerpt.

Restrictions:
1. Only mention {location} and {date}; avoid other locations or dates.
2. Exclude explicit introductions, conclusions, or character backgrounds.
3. Focus exclusively on the events of this particular {content}.

Craft a seamless narrative that gradually reveals information while maintaining reader engagement
    throughout the excerpt.

Listing 7: Template of the prompts for single chapter generation. The highlighted elements are
replaced by the event and event meta-data values.

### B.1.6 EXACT VERIFICATION OF THE CANDIDATE CHAPTER

Each candidate chapter is verified for correctness before being accepted. This is done first through
direct assessment of the constraints enforced by the meta-data and by the single chapter generation
prompt. The checks are performed as follows:

- Check that the number of paragraphs is correct, each paragraph beginning with "(X) " with
  X a number, and with increments of one,

- Check that the other entities have always the form $entity_X, with X an integer,

- Check the presence verbatim of the date, location, entity and content detail in the specified
  paragraph, while checking their absence in the other paragraphs.

### B.1.7 LLM-AS-A-JUDGE VERIFICATION OF THE CANDIDATE CHAPTER

If the chapter candidate has passed the exact verification, another round of verification is performed
(with the same LLM that generated the content) for ensuring that the generated chapter is valid.
Four boolean questions are asked, that concern each of the four event features (date, location, entity,
content), as explicited in Listing 8.

# System prompt
You are a content checker AI. Your tasks:
1. Read the given text carefully.
2. Answer true/false questions about the text.

3. Respond in JSON format.
Be accurate and concise. Only use information explicitly stated in the text.

# User prompt
Please analyze the following text enclosed between [TEXT START] and [TEXT END] markers, and answer
    the four questions below with a simple true or false. Provide your answers in a JSON format with the
    question numbers as keys and the boolean answers as values.

[TEXT START]
{generated_chapter_candidate}
[TEXT END]

Questions:
1. Does the following text takes place in a single geographical (longitude, latitude)?
2. Does the following text takes place in a single temporal day?
3. Does the following text has a single main character?
4. Does the following text has a single main event happening at that location that day (further cut into the
    events of the day)?

Your response should be in this JSON format:
{
  "1": [boolean],
  "2": [boolean],
  "3": [boolean],
  "4": [boolean]
}

Listing 8: Template of the verification prompt of a single generated chapter candidate. The first
highlighted element is replaced by the generated chapter candidate.

### B.1.8 ADDING SECONDARY ENTITIES

Other characters can appear during the generation of the chapter, under the form $entity_X (as stated
in Listing 7, and verified using direct verification). For filling the name of those secondary entities,
additional $100000$ full names are generated from two lists of $500$ first and $1000$ last names. Those
names are disjoint from the lists used for generating the main entities. In each validated chapter, a
new name is used for each indexed entity. This ensures that all the secondary entities only appear in a
single chapter. The first elements of the list are shown in Listing 9.

['Noa Middleton', 'Mara Ledbetter', 'Sienna Hamrick', 'Reid Blunt', ...]

Listing 9: Excerpt of the list of additional entities.

### B.1.9 DOCUMENT ASSEMBLY

For building the whole document, the candidate chapter generation process is applied on all events
$\{0, \ldots N_{\text{events}} - 1\}$. If the candidate does not pass the validation (either by the direct of by the
LLM-as-a-judge round), a new candidate is generated until itermax $= 10$. We discard all the events
that fail to have a valid candidate chapter after itermax.

The valid candidates are reshaped into 'Chapter {i}\n\n{generated_chapter}\n\n\n' (where i begins
from 1) and concatenated into a single document.

### B.1.10 CHAPTER EXAMPLE

We provide in Listing 10 the first generated chapter. The generated chapter has been assessed as valid
(by both verification method) at the first iteration. It corresponds to the event $0$ for which the values
are (as indicated in Listing 4):

('September 13, 2025', 'Bethpage Black Course', 'Ezra Edwards', 'Parkour Workshop', 'Demonstrated cat leaps')

and the meta-values are (as indicated in Listing 5):

{'nb_paragraphs': 7, 'idx_paragraph': {'location': 2, 'date': 7, 'entity': 2, 'content': 2}, 'style': 'thriller'}.

The secondary entities for this chapter are (as indicated in Listing 9):

['Noa Middleton', 'Mara Ledbetter'].

The generated chapter has $547$ tokens.

The harsh glare of the floodlights cast long shadows across the rugged terrain. Adrenaline coursed through his veins as he surveyed the daunting obstacles before him. The parkour workshop had drawn a diverse crowd, each participant eager to push their limits and master the urban art of movement. He flexed his fingers, anticipation building in his chest as he prepared for the night's challenges.

At Bethpage Black Course, Ezra Edwards demonstrated cat leaps with a grace that belied the difficulty of the maneuver. The other participants watched in awe as he effortlessly cleared the gap between two towering structures. Noa Middleton, the lead instructor, nodded approvingly, a hint of a smile playing at the corners of their mouth.

As the workshop progressed, the air grew thick with tension. The obstacles became increasingly complex, testing the limits of even the most seasoned traceurs. He felt a bead of sweat trickle down his spine as he approached the next challenge – a series of precarious platforms suspended high above the ground. The distant hum of crickets provided an eerie soundtrack to the scene.

Mara Ledbetter stumbled on a particularly tricky jump, their cry of alarm piercing the night air. He instinctively reached out, his fingers barely grazing their arm as they teetered on the edge of the platform. Time seemed to slow as Mara Ledbetter regained their balance, their eyes wide with fear and gratitude. The near–miss sent a shiver through the group, a stark reminder of the risks they were taking.

As midnight approached, the workshop took on a more sinister tone. Noa Middleton announced the final challenge – a complex route through a maze–like structure shrouded in darkness. The participants exchanged nervous glances, the thrill of the unknown mingling with a growing sense of unease. He took a deep breath, steeling himself for what lay ahead.

The maze was a cacophony of creaks and groans, every shadow seeming to hide a potential threat. He moved swiftly, his muscles burning with exertion as he vaulted over walls and slid under low–hanging beams. A muffled cry from somewhere in the darkness sent his heart racing. Was it merely another participant, or something more sinister?

As he emerged from the maze, panting and exhilarated, the first rays of dawn began to peek over the horizon. The workshop had pushed him to his limits, testing not just his physical abilities but his mental fortitude as well. As the group gathered for a final debrief, he couldn't shake the feeling that something fundamental had shifted during the long night of September 13, 2025. The parkour workshop may have ended, but he sensed that a new, more dangerous game was just beginning.

Listing 10: Chapter 1 generated with Claude 3.5 Sonnet (event 0, iteration 0). Event information and secondary entities have been highlighted.

### B.1.11 BOOK EXAMPLE

We provide in the Listing 11 an illustration of how the full book generated with $N_{\text{events}} = 200$ with Claude 3.5 Sonnet looks like. The generated book has 102870 tokens.

Chapter 1

The harsh glare of the floodlights cast long shadows across the rugged terrain. Adrenaline coursed through his veins as he surveyed the daunting obstacles before him. [...] The parkour workshop may have ended, but he sensed that a new, more dangerous game was just beginning.

Chapter 2

The air shimmered with an otherworldly energy as she stepped onto the glittering runway. [...] It was September 22, 2026, and on this night, she had woven dreams into reality, leaving an indelible mark on the tapestry of time.

[...]

> Chapter 196
>
> The evening air crackled with anticipation as she stepped onto the gravel path, her camera bag slung over her shoulder. [...] With a deep breath, she steeled herself for the investigation to come, knowing her keen eye and analytical mind would be put to the test in ways she never anticipated when she arrived at this seemingly innocent photography exhibition.

Listing 11: Complete document generated with Claude 3.5 Sonnet with $N_{\text{events}} = 200$. There are only 196 chapters because 4 events did not pass the verifications after 10 iterations. Ellipses are indicated with between brackets.

### B.1.12 GENERATION STATISTICS

We extract different statistics of the overall process for creating the document. In Tab. 7, we show the number of validated chapters after each iteration. The LLM-as-a-judge verification is applied only if the candidate chapter has passed the direct verification. We observe that most of the events pass after a few iterations. We didn't observe any significant characteristic for the four discarded events, for instance, the number of paragraphs is (2, 9, 1, 2) for Claude 3.5 Sonnet, and (9, 8, 2, 6) for GPT-4o.

In Tab. 8, we show the distribution of the number of paragraphs, of the styles, and of the number of secondary entities for the generated book. The distribution is shown for Claude 3.5 Sonnet, but the distribution for GPT-4o is almost the same since the original events are equal (the only difference is related to the four discarded events, that are different).

Table 7: Number of validated chapters for each iteration with $N_{\text{events}} = 200$. The verification failures correspond to the checks operated according to Sec. B.1.6 and B.1.7.

| iteration | Claude 3.5 Sonnet | | | | GPT-4o | | | |
|---|---|---|---|---|---|---|---|---|
| | remaining events to evaluate | fail direct | fail LLM | valid after this iteration | remaining events to evaluate | fail direct | fail LLM | valid after this iteration |
| 0 | 200 | 35 | 32 | 133/200 | 200 | 85 | 6 | 109/200 |
| 1 | 67 | 15 | 18 | 167/200 | 91 | 46 | 0 | 154/200 |
| 2 | 33 | 10 | 11 | 179/200 | 46 | 29 | 3 | 168/200 |
| 3 | 21 | 10 | 5 | 185/200 | 32 | 20 | 1 | 179/200 |
| 4 | 15 | 5 | 6 | 189/200 | 21 | 19 | 0 | 181/200 |
| 5 | 11 | 3 | 6 | 191/200 | 19 | 14 | 1 | 185/200 |
| 6 | 9 | 3 | 3 | 194/200 | 15 | 10 | 0 | 190/200 |
| 7 | 6 | 1 | 4 | 195/200 | 10 | 8 | 0 | 192/200 |
| 8 | 5 | 1 | 3 | 196/200 | 8 | 6 | 0 | 194/200 |
| 9 | 4 | 1 | 3 | 196/200 | 6 | 4 | 0 | 196/200 |

Table 8: Chapter count regarding the number of paragraphs, the style, and the number of secondary entities for $N_{\text{events}} = 200$, with Claude 3.5 Sonnet.

| number of paragraphs | count |
|---|---|
| 1 | 17 |
| 2 | 27 |
| 3 | 16 |
| 4 | 21 |
| 5 | 19 |
| 6 | 19 |
| 7 | 14 |
| 8 | 18 |
| 9 | 20 |
| 10 | 25 |

| style | count |
|---|---|
| comedy | 16 |
| detective | 22 |
| fantasy | 32 |
| horror | 18 |
| mystery | 25 |
| romance | 17 |
| thriller | 30 |
| tragedy | 36 |

| number of secondary entities | count |
|---|---|
| 1 | 22 |
| 2 | 74 |
| 3 | 60 |
| 4 | 27 |
| 5 | 8 |
| 6 | 2 |
| 7 | 2 |
| 8 | 1 |

We observe in Fig. 5 the relative position of the event features (date, location, entity, content details) in the text with respect to the whole book, the chapter, and the paragraph. At the book level (top), the

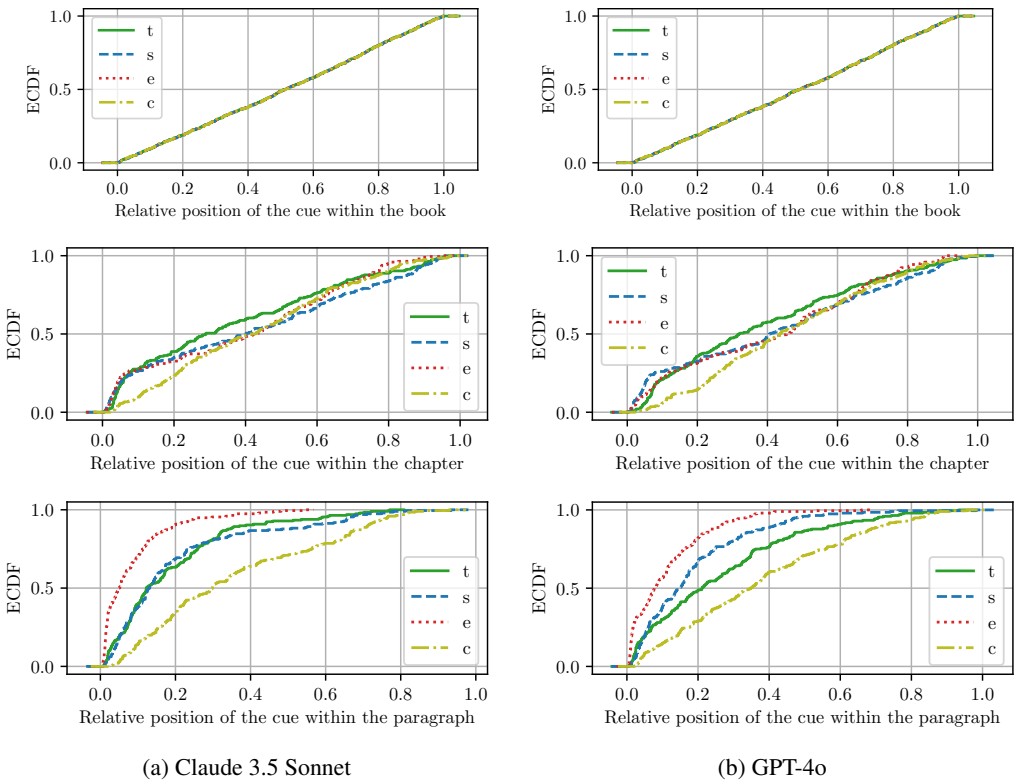

(a) Claude 3.5 Sonnet                    (b) GPT-4o

Figure 5: Position of the event features (time, space, entity, content details) relative to the book (top), the chapter (middle), the paragraph (bottom), for Claude 3.5 Sonnet (left) and GPT-4o (right), with $N_{\text{events}} = 200$.

different features are all almost uniformly spread. At the chapter level (middle), the constraints in the paragraph position ensure that the information is relatively well spread among each chapter. The discrepancy is due to the preference of the models to generate the event features at the beginning of the paragraph (bottom). We note that the bias at the paragraph level is strong for the entity name, while almost nonexistent for the verbatim content detail. Slight differences are appearing between Claude 3.5 Sonnet (left) and GPT-4o (right).

In Tab. 9, we show the actual counts of the number of times a specific event feature is repeated over the different chapters. The observed values are in accordance with the expected counts of Tab. 6. For instance, the highlighted value of **13** corresponds to 13 different dates (May 11, 2026; March 23, 2024; April 09, 2026; February 27, 2026; ...) repeated each more than 6 times (resp. 15; 13; 12; 11; ...).

## B.2 Creation of the question/answer pairs

### B.2.1 List of question/answer templates

We provide in Tab. 10 the full list of question templates. Each question is an episodic memory task based on a *cue composition* (the trigger enabling a set of events to be remembered), a *trace* (the retrieval type that needs to be extracted from each remembered event) and the information to *get* (whether all events are retrieved, only the latest state, or their chronological order). We consider all the combinations of cues and traces for the retrieval of the whole information (ids 0 to 29), while we focus on the entity cues for the latest and chronological retrievals. In the table, the cue has four components: $t$ (the time), $s$ (the space), $e$ (the entity) and $c$ (the event content). The trace can be one of those four components (Times, Spaces, Entities, or Contents), the list of the secondary entities (Other entities), or the full details (Full Details) regarding a specific chapter. The elements to get are all the retrieved elements (all), the latest one (latest) or the chronological order of the elements

Table 9: Actual counts in selecting one, two, three to five, or more than six times a specific item (e.g., date) in our experiments. For instance, the highlighted value indicates that **13** different days are repeated each more than 6 times among the 200 generated events. This table echoes the expected counts of Tab. 6.

| | | Bin of counts | | | |
|---|---|---|---|---|---|
| $N_{\text{events}}$ | feature | 1 | 2 | 3-5 | 6+ |
| 20 | time | 8 | 4 | 1 | 0 |
| 20 | space | 6 | 5 | 1 | 0 |
| 20 | entity | 8 | 4 | 1 | 0 |
| 20 | content | 8 | 1 | 3 | 0 |
| 200 | time | 6 | 4 | 14 | **13** |
| 200 | space | 7 | 7 | 8 | 13 |
| 200 | entity | 9 | 3 | 7 | 15 |
| 200 | content | 9 | 4 | 7 | 14 |

(chrono.). All the ground truth answers are known, since the document has been built with known events. For the question regarding the full details, we take the full chapter verbatim as the ground truth answer.

### B.2.2   LIST OF QUESTIONS WITH NON-EMPTY ANSWERS GIVEN A BOOK

The questions are created given (i) the question/answer templates and (ii) the ground truth regarding all the events of the document built in Sec. B.1.

For each chapter, we retrieve the ground truth tuple $(t, s, e, c)$ regarding the date, the location, the main entity, and the event content. We then generate all the questions by following the template. After building all the questions for all the chapters, we filter the duplicates.

We obtain 564 questions for the documents produced with $N_{\text{events}} = 20$ (filtered from $684 = 36$ ids $\times$ 19 chapters questions) and 3886 questions for the documents produced with $N_{\text{events}} = 200$ (filtered from $7056 = 36$ ids $\times$ 196 chapters questions).

We provide an example in Tab. 11 ("Non-empty" row).

### B.2.3   LIST OF QUESTIONS WITH EMPTY ANSWERS GIVEN A BOOK

In addition to the questions for which the answer can be found in the book, we design questions with empty answers. We employ two strategies: the inner strategy and the outer one. We first initially define the *unused static universe* as the subset of the static universe that has not been used during the document generation, and which is expected, according to Tab. 6, to contain 66 elements for each event feature (time, space, location, content) when $N_{\text{universe}} = 100$, $N_{\text{events}} = 200$, and $\mathcal{D} = $ geometric.

Then, to create the fake event, we corrupt a tuple $(t, s, e, c)$ from an existing chapter, by applying an i.i.d. binary mask $(b_t, b_s, b_e, b_c)$, where each component follows a Bernoulli distribution with probability $1/2$. When the mask is applied, we replace the corresponding feature (e.g., the date $t = $ September 13, 2025) by the same kind of feature, either sampled from another chapter (inner strategy) or from the unused static universe (outer strategy). The resulting tuple $(t', s', e', c')$ is then replaced in the template questions, in the same manner as Sec. B.2.2.

The process is applied for both inner and outer strategies once for each chapter. The resulting questions are also filtered to keep only those with empty answers.

We obtain 438 additional questions for the documents with $N_{\text{events}} = 20$ and 3657 questions for the documents with $N_{\text{events}} = 200$.

We provide an example of each strategy in Tab. 11 ("Empty (inner)" and "Empty (outer)" rows).

Table 10: List of all the question templates based on cue composition, retrieval trace, and "get" style.

| id | Cue | Trace | Get | Template question |
|---|---|---|---|---|
| 0 | (t,*,*,*) | Spaces | all | Recall all the events that occurred on {t}. Without describing the events, list all the unique locations where these events took place. |
| 1 | (t,*,*,*) | Entities | all | Consider all events that happened on {t}. Provide a list of all protagonists involved in any of these events, without describing the events themselves. |
| 2 | (t,*,*,*) | Contents | all | Reflect on {t}. Describe all the key events that occurred on this date, focusing on what happened rather than who was involved or where it took place. |
| 3 | (*,s,*,*) | Times | all | Think about all events that have occurred at {s}. Provide a list of all dates when these events took place, without describing the events. |
| 4 | (*,s,*,*) | Entities | all | Consider the location {s}. List all protagonists that have been involved in any events at this location, without mentioning the events themselves. |
| 5 | (*,s,*,*) | Contents | all | Recall the various events that have taken place at {s}. Describe what happened during these events, focusing on the actions or occurrences rather than the timing or people involved. |
| 6 | (*,*,e,*) | Times | all | Reflect on all events involving {ent}. Provide a list of all dates when these events occurred, without describing the events. |
| 7 | (*,*,e,*) | Spaces | all | Consider all events that {ent} has been involved in. List all the locations where these events took place, without mentioning the events themselves. |
| 8 | (*,*,e,*) | Contents | all | Think about {ent}'s experiences. Describe all the key events they've been involved in, focusing on what happened rather than when or where it occurred. |
| 9 | (*,*,*,c) | Times | all | Recall all events related to {c}. Provide a list of all dates when these events occurred, without describing the events. |
| 10 | (*,*,*,c) | Spaces | all | Consider all events involving {c}. List all the locations where these events took place, without mentioning the events themselves. |
| 11 | (*,*,*,c) | Entities | all | Reflect on events related to {c}. Provide a list of all protagonists involved in these events, without describing the events. |
| 12 | (t,s,*,*) | Entities | all | Think about what happened at {s} on {t}. List all protagonists involved in any events at this time and place, without describing the events. |
| 13 | (t,s,*,*) | Contents | all | Recall the key events that occurred at {s} on {t}. Describe what happened, focusing on the actions or occurrences rather than who was involved. |
| 14 | (t,*,e,*) | Spaces | all | Consider the events involving {ent} on {t}. List all the locations where these events took place, without describing the events themselves. |
| 15 | (t,*,e,*) | Contents | all | Reflect on what {ent} experienced on {t}. Describe all the key events they were involved in, focusing on what happened rather than where it occurred. |
| 16 | (t,*,*,c) | Spaces | all | Recall the events related to {c} that occurred on {t}. List all the locations where these events took place, without describing the events themselves. |
| 17 | (t,*,*,c) | Entities | all | Think about the events involving {c} on {t}. Provide a list of all protagonists involved in these events, without describing the events. |
| 18 | (*,s,e,*) | Times | all | Consider all events involving {ent} at {s}. Provide a list of all dates when these events occurred, without describing the events. |
| 19 | (*,s,e,*) | Contents | all | Reflect on {ent}'s experiences at {s}. Describe all the key events they've been involved in at this location, focusing on what happened rather than when it occurred. |
| 20 | (*,s,*,c) | Times | all | Recall all events related to {c} that occurred at {s}. Provide a list of all dates when these events took place, without describing the events. |
| 21 | (*,s,*,c) | Entities | all | Think about the events involving {c} at {s}. List all protagonists involved in these events, without mentioning the events themselves. |
| 22 | (*,*,e,c) | Times | all | Consider all events involving both {ent} and {c}. Provide a list of all dates when these events occurred, without describing the events. |
| 23 | (*,*,e,c) | Spaces | all | Reflect on the experiences of {ent} related to {c}. List all the unique locations where these events took place, without mentioning the events themselves. |
| 24 | (t,s,e,*) | Contents | all | Recall what happened involving {ent} at {s} on {t}. Describe the key events or activities that occurred, focusing on what happened. |
| 25 | (t,s,*,c) | Entities | all | Think about the events related to {c} that occurred at {s} on {t}. List all protagonists involved in these events, without describing the events themselves. |
| 26 | (t,*,e,c) | Spaces | all | Consider the events involving both {ent} and {c} on {t}. List all the locations where these events took place, without describing the events themselves. |
| 27 | (*,s,e,c) | Times | all | Consider all events involving both {ent} and {c} at {s}. Provide a list of all dates when these events occurred, without describing the events. |
| 28 | (t,s,e,c) | Other entities | all | Recall what happened involving {ent} and {c} at {s} on {t} and list only who else was involved (if anyone). |
| 29 | (t,s,e,c) | Full details | all | Provide a comprehensive account of what happened involving {ent} and {c} at {s} on {t}. Include all relevant details about the event(s), including what occurred and any other pertinent information. |
| 30 | (*,*,e,*) | Times | latest | What is the most recent date {ent} was observed or mentioned in the story's chronology? |
| 31 | (*,*,e,*) | Spaces | latest | What is the most recent location where {ent} was observed in the story's chronological timeline? |
| 32 | (*,*,e,*) | Contents | latest | What was {ent} doing the last time they were observed in the story's timeline? |
| 33 | (*,*,e,*) | Times | chrono. | Provide a chronological list of all dates when {ent} was observed, from earliest to latest in the story's timeline. |
| 34 | (*,*,e,*) | Spaces | chrono. | List all locations visited by {ent} in chronological order according to the story's timeline. |
| 35 | (*,*,e,*) | Contents | chrono. | Enumerate all activities that {ent} has been involved in, ordered from earliest to latest in the story's chronology. |

Table 11: Real question examples applied on Chapter 1 given the ground truth tuple $(t, s, e, c) =$ ('September 13, 2025', 'Bethpage Black Course', 'Ezra Edwards', 'Parkour Workshop') and for the question with id $= 21$. The date 'May 07, 2024' appears in chapters 3 and 129, the location 'Central Park' appears in chapter 166, while the content 'Laser Tag Tournament' never appears in the document.

| Kind | Mask | Replacement | Question | Ground truth answer | in |
|---|---|---|---|---|---|
| Non-empty | / | / | Think about the events involving Parkour Workshop at Bethpage Black Course. List all protagonists involved in these events, without mentioning the events themselves. | {'Chloe Castillo' 'Ezra Edwards' 'Levi Rodriguez'} | Chapters 1, 109, 128 |
| Empty (inner) | (1,1,0,0) | $t'$={May 07, 2024}, $s'$={Central Park} | Think about the events involving Parkour Workshop at Central Park. List all protagonists involved in these events, without mentioning the events themselves. | $\varnothing$ | / |
| Empty (outer) | (0,0,0,1) | $c'$ ={Laser Tag Tournament} | Think about the events involving Laser Tag Tournament at Bethpage Black Course. List all protagonists involved in these events, without mentioning the events themselves. | $\varnothing$ | / |

Table 12: Widespreadness of the selected questions: number of selected questions per cue per bin, after the filtering described in Sec. B.2.4.

| | $N_{\text{events}} = 20$ | | | | | $N_{\text{events}} = 200$ | | | | |
|---|---|---|---|---|---|---|---|---|---|---|
| Cue/Bin | 0 | 1 | 2 | 3-5 | 6+ | 0 | 1 | 2 | 3-5 | 6+ |
| (t,*,*,*) | 5 | 5 | 4 | 1 | 0 | 5 | 5 | 4 | 5 | 5 |
| (*,s,*,*) | 5 | 5 | 5 | 1 | 0 | 5 | 5 | 5 | 5 | 5 |
| (*,*,e,*) | 5 | 5 | 4 | 1 | 0 | 5 | 5 | 3 | 5 | 5 |
| (*,*,*,c) | 5 | 5 | 1 | 3 | 0 | 5 | 5 | 4 | 5 | 5 |
| (t,*,*,c) | 5 | 5 | 1 | 0 | 0 | 5 | 5 | 5 | 5 | 0 |
| (*,s,*,c) | 5 | 5 | 1 | 0 | 0 | 5 | 5 | 5 | 5 | 0 |
| (*,*,e,c) | 5 | 5 | 1 | 0 | 0 | 5 | 5 | 5 | 5 | 0 |
| (*,s,e,*) | 5 | 5 | 0 | 0 | 0 | 5 | 5 | 5 | 4 | 0 |
| (*,s,e,c) | 5 | 5 | 0 | 0 | 0 | 5 | 5 | 2 | 0 | 0 |
| others | 5 | 5 | 0 | 0 | 0 | 5 | 5 | 0 | 0 | 0 |

### B.2.4 SELECTION OF QUESTIONS

The number of all possible questions is large and increases significantly as a function of $N_{\text{events}}$. Since the evaluation cost scales with the number of questions, we decided to extract only a subset of the overall questions.

First, we consider the number of events that should be remembered for answering each question. We group the questions into five bins: $\{0\}$ (the empty questions), $\{1\}$ (the question that triggers a single chapter), $\{2\}$, $\{3, 4, 5\}$ and $\{6+\}$. In the example provided in Tab. 11 ("Non-empty" row), the question is put into the $\{3, 4, 5\}$ group (this group is also noted 3-5). For the questions involving the latest state, we consider *all* the chapters triggered by the cue for considering the group. We expect that the more chapters are involved, the more difficult the question is.

We then target the extraction of $N_{\text{target}} = 5$ questions for each of the 36 question ids and for each of the 5 bins, resulting in up to 900 questions per experiment.

This method ensures that the questions are relatively well spread among the different bins, as shown in Tab. 12. In this table, we show the final number of questions obtained for each cue and each bin, for both $N_{\text{events}} = 20$ (left) and $N_{\text{events}} = 200$ (right). Each cue is associated to a set of question ids. For instance, the cue $(*, *, e, *)$ is associated to the ids 6, 7, 8, 30, 31, 32, 33, 34, 35. We observe that we always retrieve the 5 target questions for bins $\{0\}$ and $\{1\}$. For the bin with $\{6+\}$ with $N_{\text{events}} = 20$, we know for sure that no question exists since a single cue has been associated to at most three chapters (as seen in Tab. 9). For the single cue recall (first four rows), most of the bins are filled. For the multiple cues (from row five), it is difficult with $N_{\text{events}} = 20$ to cover all the combinations, while most of the bins are filled with $N_{\text{events}} = 20$. The others cues (last row) involve the pair $(t, s)$ or $(t, e)$, for which we imposed at most one extracted chapter (since we forbid in the document generation the possibility of two chapters involving the same location at the same time, or the same entity at the same time), hence all the questions are necessarily in the bins $\{0\}$ or $\{1\}$.

At the end, we end up with 456 questions for $N_{\text{events}} = 20$, and 686 questions for $N_{\text{events}} = 200$.

### B.2.5 Selection of the fine-tuning questions

Our naive fine-tuning experiment aims to incorporate the essential information needed to generalize answers across all benchmark questions. Therefore, we restrict our training data to question-answer pairs that correspond to single events (each event being associated with a single chapter). These pairs establish basic facts (e.g., taking the example of Fig. 6, "Jackson Ramos attended a carnival in Central Park on September 22, 2026" related to the event of Chapter 163, "Jackson Ramos did a flash mob at Ellis Island on April 09, 2026" related to the event of Chapter 96, etc.), which in principle enables deducing answers to questions involving multiple events, such as the set of all locations involving Jackson Ramos: {Central Park, High Line, One World Trade Center, Ellis Island, Snug Harbor Cultural Center} (question template id 7 in Tab. 10).

The fine-tuning process thus uses 3,199 training questions for the long book (filtered from the total 3,886 questions involving one or several events, as mentioned in Sec. B.2.2) and 468 training questions for the short book.

Note that regarding the testing data for the fine-tuning experiment, we use the set of questions indicated in Sec. B.2.4 (as for the other experiments). Specifically, all questions involving a single chapter (i.e., corresponding to the bin {1}) are present in both the training and the test sets, while all other questions appear only in the test set.

### B.3 Evaluation strategy

In the previous Secs. B.1 and B.2, we have described the creation of the memory to encode and of the list of questions to ask, along with the corresponding ground truth answers. Once a model has ingested the memory to encode (e.g., by directly putting the document in-context, by using RAG, or by fine-tuning the model), and once the predicted answers have been extracted from the model, we use a common evaluation strategy to compare the ground truth answers with the predicted ones.

### B.3.1 Main evaluation prompt

We use the main evaluation prompt defined in Listing 12 for deducing the F1-score between the ground truth and the predicted answers.

The key idea is to adopt the ground truth point of view for performing the evaluation, since we are sure of the number of ground truth items. In consequence, we assign for each ground truth item a matching score between 0 (the item has been missing in the LLM predicted answer) and 1 (the item has been found, considering synonyms or close meanings). In addition, we ask for the all identified items in the AI answer, which is necessary to know the number of predictions, and hence assess false positives.

### B.3.2 Computation of the F1-score

The list of identified items iditems is not entirely controlled given the way it has been extracted. There are numerous cases where its size is larger than the size of the ground truth, even for a coherent AI-generated answer. For instance, (i) in the case of a full event, the ground truth is the full chapter while the identified items may be a detailed list, (ii) in the case of the latest content, the ground truth has size one while the identified items may include details about the different steps of the content (as seen in the Listing 13), (iii) in the case of entity retrieval, the model may include the secondary entities, that are not the protagonist but are still related. For those reasons, we decided to be lenient in the number of predictions #pred and to define $\#\text{pred} = \min(\#\text{iditems}, \#\text{gt})$ if $\#\text{gt} > 0$, and #iditems otherwise. This policy is nevertheless straightforward to tune, since it does not rely on an additional generative model. At the end, the sum of the matching scores $S$ is the quantity that assesses the number of true positives, from which the precision $S/\#\text{pred}$ and recall $S/\#\text{gt}$ are computed (with #gt the number of ground truth items and #pred the number of predictions). The F1-score is then computed as their harmonic mean.

You are an expert judge evaluating the accuracy of an AI−generated answer against a known groundtruth.
Questions can probe for different types or aspects, like what actions or events took place, what people were involved, what were the dates, or what were the locations or spaces.

```
Question type: {retrieval_type}
Groundtruth: {correct_answer}
AI−generated answer: {llm_answer}

Your task:
− Identify all unique items in the AI−generated answer that are relevant to the question type. Answer an
       empty list [] for this field in case of at least one negative information (e.g., when the answer begins by
       telling there is no information, or cannot answer)
− Determine a matching score between 0 and 1 for each ground truth item. Give 1 if the item has been found
       in the relevant items of the AI−generated answer, considering synonyms, paraphrases, or close
       meanings. Give 0.5 if the item could be considered related to any AI−generated item but without being
       explicitly stated as such. Give 0 if the item missed mentioning a specific AI−generated item.
− Provide a brief explanation of the evaluation

Provide your evaluation in the following JSON format:
{
   "identified_items_in_AI_answer": ["AI_answer_item_1", "AI_answer_item_2", ...],
   "matching_score": [{"{correct_answer_0}": "score_between_0_and_1"},
                        {"{correct_answer_1}": "score_between_0_and_1", ...}],
   "explanation": "Brief explanation of your evaluation"
}
```

Listing 12: Template of the verification prompt of a single generated chapter candidate. The first highlighted element is replaced by the generated chapter candidate.

```
# Question
Recall the key events that occurred at Snug Harbor Cultural Center on April 09, 2026. Describe what
       happened, focusing on the actions or occurrences rather than who was involved.

# Ground truth answer
[Tech Hackathon]

# Predicted answer
On April 09, 2026, the atmosphere at Snug Harbor Cultural Center was charged with excitement as
       developers from the tech world gathered for a hackathon. Energy buzzed throughout the space, with
       participants mingling and discussing innovations while sunlight streamed through the tall windows. As
       the day progressed, focus shifted to collaborative projects, where participants worked intensely to create
       innovative solutions.

The event culminated with presentations, showcasing the various projects developed throughout the day. One
       key presentation was met with thunderous applause, reflecting the success of the collaborations and
       creativity that flourished during the hackathon. Overall, the day was marked by a sense of belonging,
       connection, and shared passion for technological advancements.

# Identified items in AI answer
['Tech Hackathon', 'developers gathered', 'collaborative projects', 'innovative solutions', 'presentations']

# Matching score
[{'Tech Hackathon': 1}]

# Explanation
The AI−generated answer directly mentions a 'hackathon' involving tech developers, which perfectly matches
       the ground truth 'Tech Hackathon'. The answer provides additional details about the event, including the
       collaborative nature, focus on innovation, and culmination in presentations, all of which are consistent
       with a typical tech hackathon.
```

Listing 13: Example with correct answer provided by GPT-4o mini (on the document with 20 events produced by Claude 3.5 Sonnet), which however gives more identified items than the ground truth. In this case, we are lenient in the number of predictions, resulting in F1=1, as detailed in Sec. B.3.2.

### B.3.3 COMPUTATION OF KENDALL'S TAU COEFFICIENT

For the evaluation of the question involving the chronological order of the retrieved traces, we compare the identified items in AI answer obtained in the Listing 12 with the ground truth answer. The identified items may not have the same length as the ground truth answer, and the names may be slightly different (e.g., "Theater Performance" instead of "Theater Show"). For those reasons, we cannot directly compute the Kendall's tau and rely on the prompt in Listing 14 for extracting the matching items while preserving the order.

We then consider only the questions that are related to a ground truth answer of size at least 2 (otherwise there is no order). Then, we consider the identified items that entirely match with the ground truth answer. This selection ensures that we specifically compare the ordering ability, and not the retrieval one. Finally, for each (ground truth list, identified items) pair, we deduce the elements that are in both lists (while preserving order), and compute their Kendall's tau. A Kendall's tau is a measure between $-1$ and 1, 1 corresponding to a perfect match in the indexes of the two lists.

You are an expert judge evaluating the alignment between an AI−generated list and a known groundtruth list. Your task is to match items from the predicted list to the groundtruth list, considering their order and uniqueness.

Given:
Groundtruth list: {groundtruth_items}
Groundtruth indexes: {groundtruth_indexes}
Predicted list: {predicted_items}

Instructions:
1. For each item in the predicted list, find the first corresponding index from the groundtruth list that hasn't been used yet.
2. Assign indexes based on these rules:
a. If a match is found and the groundtruth index hasn't been used, assign that index.
b. If no match is found, or if all matching indexes have already been used, assign −1.
3. Always use the earliest matching index from the groundtruth list, even if there's an exact match later.
4. Provide a brief explanation of your index assignments.

Output your evaluation in the following JSON format:
{{
    "groundtruth_indexes": {groundtruth_indexes},
    "predicted_indexes": [index1, index2, ...],
    "explanation": "Concise explanation of index assignments"
}}

Consider these examples:

Example 1:
Groundtruth list: ['Ice Preservation Discussions', 'Theater Show', 'Parkour Workshop']
Predicted list: ['Theater Performance', 'Tech Hackathon', 'Ice Preservation Talks']
{{
    "groundtruth_indexes": [0, 1, 2],
    "predicted_indexes": [1, −1, 0],
    "explanation": "Theater Performance matches Theater Show (index 1), Tech Hackathon has no match (−1), Ice Preservation Talks matches Ice Preservation Discussions (index 0)."
}}

Example 2:
Groundtruth list: ['Ice Preservation Discussions', 'Theater Show', 'Parkour Workshop', 'Theater Performance']
Predicted list: ['Theater Performance', 'Tech Hackathon', 'Ice Preservation Talks']
{{
    "groundtruth_indexes": [0, 1, 2, 3],
    "predicted_indexes": [1, −1, 0],
    "explanation": "Theater Performance matches Theater Show (index 1, first available match), Tech Hackathon has no match (−1), Ice Preservation Talks matches Ice Preservation Discussions (index 0)."
}}

> Now, please provide your evaluation for the given lists:

Listing 14: Template of the evaluation prompt for extracting the items matching between the identified and the ground truth items. The ground truth indexes are always set to 0,...,N-1, with N the length of the ground truth.

### B.4 MATCHING SCORES PROVIDED BY THE LLM-AS-A-JUDGE

The Listing 12 uses an LLM-as-a-judge for determining a matching score between 0 and 1 for each ground truth item, given an answer provided in a freeform format. In this setting, the LLM is used for identifying and structuring the information, not for making subjective judgments.

The vast majority of the scores are either 0 or 1, with partial matches occurring in only 4% of cases for gpt-4o and 8% for gpt-4o-mini (on the long book). Since the handling of partial matches requires careful interpretation, we present below an example for each retrieval type. For each example, we include the raw input answer, the LLM's final scoring against each ground truth item, and the LLM's explanation of its evaluation (with (...) indicating ellipses). These examples demonstrate the validity of our relevance scoring approach.

- Retrieval type: Time
    - Raw input answer: "Based on the provided text, the most recent date Jackson Ramos was observed or mentioned in the story's chronology is September 03, 2026, as described in Chapter 165.",
    - Scoring relevance against each ground truth item: ['September 22, 2026': 0.5],
    - Explanation of the evaluation: "The AI-generated answer provides a specific date (September 03, 2026) which is close to but not exactly matching the groundtruth date (September 22, 2026). Both dates are in September 2026, so there's a partial match, but the day is different. Therefore, a score of 0.5 is given to reflect this partial accuracy".

- Retrieval type: Space
    - Raw input answer: 'The locations where events involving Bella Brown took place, as mentioned in the text, are: 1. Washington Square Park 2. Lincoln Center 3. Bethpage State Park',
    - Scoring relevance against each ground truth item: ['High Line': 0, 'Bethpage Black Course': 0.5, 'Lincoln Center': 1],
    - Explanation of the evaluation: 'The AI answer correctly identified Lincoln Center, which matches exactly with the groundtruth. Bethpage State Park is mentioned, which is related to but not exactly the same as Bethpage Black Course, so it receives a partial score. The High Line is completely missing from the AI answer. The AI also included Washington Square Park, which is not in the groundtruth'.

- Retrieval type: Entity
    - Raw input answer: 'The protagonists involved in events related to Burlesque Performance are: 1. Julian Ross 2. Samuel Parker 3. Isaiah (Last name not provided) 4. Sofia Ramos',
    - Scoring relevance against each ground truth item: ['Isaiah Bennett': 0.5, 'Sofia Ramos': 1, 'Julian Ross': 1],
    - Explanation of the evaluation: 'The AI-generated answer correctly identified Julian Ross and Sofia Ramos, earning a score of 1 for each. For Isaiah Bennett, only the first name 'Isaiah' was mentioned without a last name, so it receives a score of 0.5. The AI answer also included Samuel Parker, who is not in the ground truth list. Overall, the AI answer captured most of the correct entities but added an extra person and missed the full name for Isaiah Bennett'.

- Retrieval type: Content
    - Raw input answer: 'At One World Trade Center, several distinct events took place, each with its own unique focus and set of occurrences: 1. Tech Hackathon: (...) . 2. Photography Exhibition: (...). 3. Astronomy Show: (...) . 4. Hackathon: (...) . 5. Peer Review Workshop: (...) . 6. Fashion Show: (...)',

- Scoring relevance against each ground truth item: ['Photography Exhibition': 1, 'Fashion Show': 1, 'Ice Sculpture Exhibition': 0, 'Theater Performance': 0, 'Tattoo Convention': 0, 'Carnival': 0, 'Tech Hackathon': 1, 'Educational Workshop': 0.5, 'Murder Mystery Dinner': 0, 'Ghost Tour': 0, 'Debate Tournament': 0, 'Fire Dancing Performance': 0, 'Classical Music Concert': 0],
  - Explanation of the evaluation: 'The AI-generated answer correctly identified 3 events from the ground truth: Tech Hackathon, Photography Exhibition, and Fashion Show. The Peer Review Workshop could be considered related to an Educational Workshop, hence the 0.5 score. The Astronomy Show is not in the ground truth. The answer missed several events from the ground truth (...)'.

## C    ILLUSTRATION OF THE TRACKING OF AN ENTITY WITHIN THE LONG BOOK

We illustrate in Fig. 6 the tracking of entities across chapters for the default book with 200 chapters, which takes place in New York, highlighting in red Jackson Ramos, one of the entity present in the book. Each dot corresponds to a location involved in at least one chapter (for a total of 35 locations, as indicated in Tab. 2; some dots are outside the figure). The paths of each entity (in the narrative order) across the locations are indicated with segments (for a total of 34 paths). For each event involving Jackson Ramos, we annotate the chapter number, the date, the place, and the event content.

We observe that while our events are generated independently, they exist within a shared universe with a common set of entities (e.g., Jackson Ramos), a common set of locations (within New York in our default book), and a coherent timeline. This allows tracking entities across space and time, even without causal links.

For further provide two examples of question/answer pairs involving this entity (built from the question templates shown in Tab. 10):

- Question built from id 6, cue (*,*,e,*), and trace Times:
  - Question: "Reflect on all events involving Jackson Ramos. Provide a list of all dates when these events occurred, without describing the events."
  - Ground truth answer: "September 22, 2026", "February 27, 2026", "August 24, 2026", "April 09, 2026", "June 14, 2025" (unordered set of elements).
- Question built from id 18, cue (*,s,e,*), and trace Times:
  - Question: "Consider all events involving Jackson Ramos at Central Park. Provide a list of all dates when these events occurred, without describing the events."
  - Ground truth answer: "September 22, 2026".

For answering those questions, the model must track Jackson's appearances across multiple chapters (for both questions), identify which appearances occurred at Central Park, and synthesize multiple date/location pairs (for the second question).

## D    GENERATING ANSWERS USING IN-CONTEXT, RAG, AND FINE-TUNED MODELS

We enumerate the prompts used in the different answer generation models.

The system prompt is always set to "You are an expert in memory tests." for the three model types.

The user prompts are given in the Listings 15 , 16, and 17, for respectively the in-context, the RAG, and the fine-tuned model.

```
# Episodic Memory Benchmark

You are participating in an episodic memory test. You will be presented with a text to read and internalize as
    if you had personally experienced the events described. After the text, you will find a question about the
    content. Please answer this question based solely on the information provided in the text.

## The Text to Memorize:
```

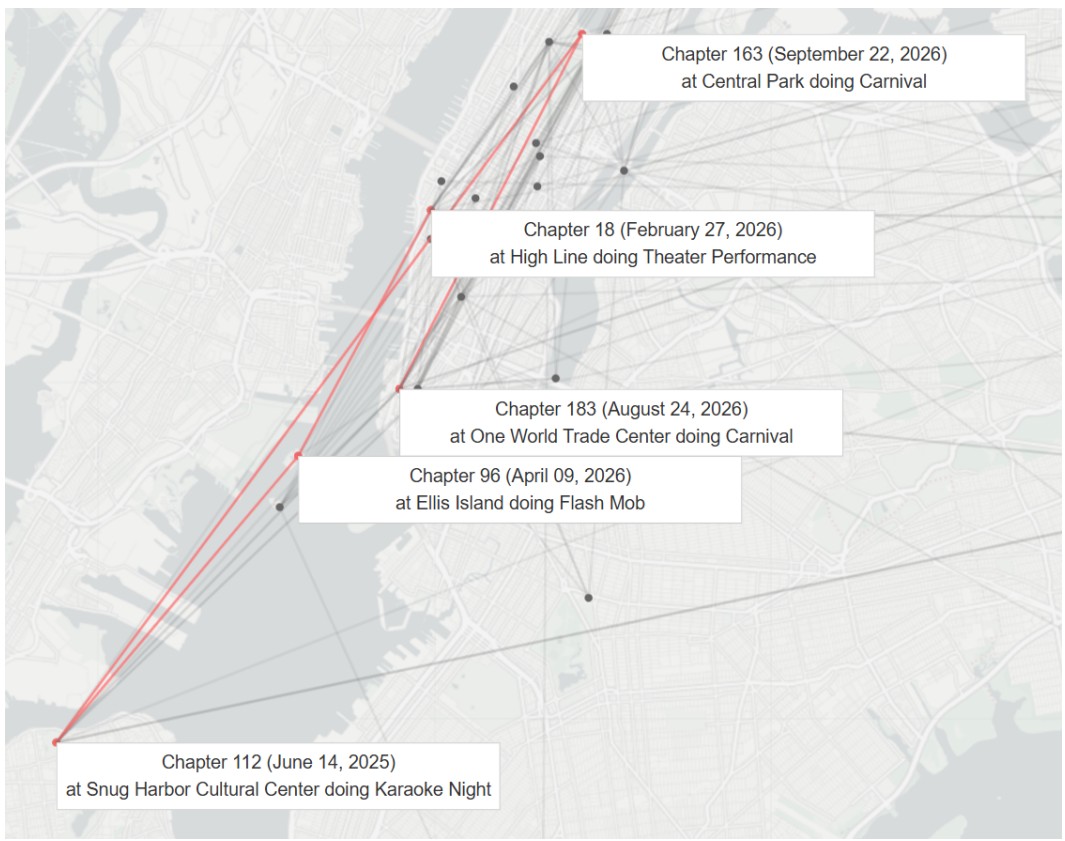

Figure 6: Tracking of a single entity (specifically, Jackson Ramos) throughout the long book.

{book_content}

## Question:

{question}

Please answer the question to the best of your ability, based only on the information provided in the text above. If you are unsure about an answer, it's okay to say so. Do not invent or assume information that was not explicitly stated in the text.

Listing 15: Template of the answer generation prompt for *in-context* model. The 'book_content' and the 'question' are replaced respectively by the full book and a single question of interest.

# Episodic Memory Benchmark

You are participating in an episodic memory test, based on the data below, which was retrieved from a book. You need to read it and internalize as if you had personally experienced the events described. After the text, you will find a question about the content. Please answer this question based solely on the information provided in the retrieved data.

## Retrieved Relevant Chunks from the Book:

{book_chunks}

## Question:

{question}

> Please answer the question to the best of your ability, based only on the information provided in the relevant chunks above. If you are unsure about an answer, it's okay to say so. Do not invent or assume information that was not explicitly stated in the text.

Listing 16: Template of the answer generation prompt for *RAG* model. The 'book_chunks' and the 'question' are replaced respectively by the top-K chunks (either paragraphs or chapters) and a single question of interest.

> This question is about the book "Synaptic Echoes 2026: The Neuro–Temporal Paradox of Episodic Precognition". All events in this book are purely fictional and do not correspond to real–world timelines. Please answer based solely on the content of this fictional story.
>
> Question: {question}

Listing 17: Template of the answer generation prompt for the *fine-tuned* model. The 'question' is replaced by a single question of interest.

## E  EXTENDED RESULTS

### E.1  RESULTS ON THE SHORT BOOK

We show results on the short book. The main difference is that o1-mini performance, as well as Claude-3.5 Sonnet in-context, are significantly better.

**Overall performance comparison.** We present in Fig. 7 the results obtained for the short book. Compared to the larger book, we obtain an excellent performance of o1-mini, this is only in par with GPT-4o. GPT-4o and Claude 3.5 Sonnet cannot be distinguished significantly, while the other methods have worse performance. Those results are completed with Tab. 13: o1-mini gives consistently good results for all bins.

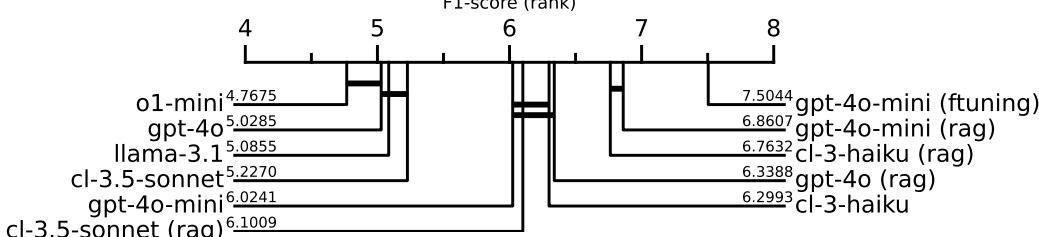

Figure 7: *Overall performance comparison:* CD plot with Wilcoxon signed rank test Demšar (2006); Benavoli et al. (2016) between each pair of algorithm (adjusted by the Holm's method) on the short book.

**Performance on simple recall tasks.**

Tab. 13 mirrors the results on the short book, where we gather a different ranking of the models and memory combination as already illustrated in Fig. 7.

### E.2  RAG ABLATION STUDY: CHAPTER VS PARAGRAPH CHUNKS

The option selected in Sec. 5 in the RAG setting was to chunk the document into the different *paragraphs*. However, as a single event in our benchmark often spreads across multiple paragraphs, a paragraph-based RAG approach may not capture all information related to an event.

Another possibility is to chunk the document at each *chapter*, which constitute an ideal upper-bound, as in practical cases information would not be so well structured. Using this strategy, we obtain longer but fewer chunks. The top-K chapters is set to 17, since 17 is the maximum number of events associated to a single question for the larger book. In this case, each chunk contains by design a single event of interest. As expected, this improves the performance results compared to the RAG paragraph strategy, as shown in Tab. 14.

Table 13: *Performance on simple recall tasks*: for questions on the short book, F1-score average performance and deviation as a function of the number of ground truth answers.

| Memory | Model | Bin (count) | | | |
|---|---|---|---|---|---|
| | | 0 (150) | 1 (150) | 2 (48) | 3-5 (18) |
| **In-context** | gpt-4o-mini | 0.53±0.50 | 0.92±0.23 | 0.87±0.21 | 0.89±0.16 |
| | gpt-4o | 0.86±0.35 | 0.96±0.16 | **0.93±0.16** | 0.88±0.16 |
| | claude-3-haiku | 0.81±0.39 | 0.74±0.43 | 0.59±0.31 | 0.65±0.20 |
| | claude-3-5-sonnet | **0.98±0.14** | 0.94±0.23 | 0.73±0.22 | 0.73±0.20 |
| | o1-mini | 0.97±0.16 | 0.94±0.21 | 0.90±0.18 | **0.93±0.11** |
| | llama-3.1-405b | 0.91±0.28 | 0.95±0.18 | 0.89±0.18 | 0.83±0.17 |
| **RAG** | gpt-4o-mini | 0.63±0.49 | 0.61±0.43 | 0.69±0.32 | 0.71±0.33 |
| | gpt-4o | 0.89±0.31 | 0.63±0.43 | 0.65±0.33 | 0.68±0.31 |
| | claude-3-haiku | 0.75±0.44 | 0.60±0.44 | 0.66±0.33 | 0.65±0.34 |
| | claude-3-5-sonnet | 0.93±0.26 | 0.62±0.44 | 0.70±0.33 | 0.70±0.29 |
| **Fine-tuning** | gpt-4o-mini | 0.00±0.00 | **1.00±0.04** | 0.62±0.23 | 0.46±0.19 |

Table 14: *RAG ablation study*. Top table: F1-score average performance and deviation for GPT-4o using direct prompting, RAG with paragraph cut (default strategy), and RAG with chapter cut (in this ablation). Bottom table: F1 scores for all models for chapter vs paragraph chunks.

| GPT-4o Memory | Bin (count) | | | | |
|---|---|---|---|---|---|
| | 0 (150) | 1 (150) | 2 (90) | 3-5 (98) | 6+ (60) |
| **In-context** | 0.84±0.37 | 0.81±0.38 | 0.60±0.31 | 0.57±0.21 | 0.53±0.14 |
| **RAG (paragraph)** | 0.82±0.39 | 0.60±0.46 | 0.55±0.33 | 0.55±0.28 | 0.59±0.21 |
| **RAG (chapter)** | **0.88±0.33** | **0.82±0.37** | **0.72±0.37** | **0.70±0.31** | **0.60±0.30** |

| Memory | Model | Bin (count) | | | |
|---|---|---|---|---|---|
| | | 0 (150) | 1 (150) | 2 (48) | 3-5 (18) |
| **RAG (paragraph)** | gpt-4o-mini | 0.63±0.49 | 0.61±0.43 | 0.69±0.32 | 0.71±0.33 |
| | gpt-4o | 0.89±0.31 | 0.63±0.43 | 0.65±0.33 | 0.68±0.31 |
| | claude-3-haiku | 0.75±0.44 | 0.60±0.44 | 0.66±0.33 | 0.65±0.34 |
| | claude-3-5-sonnet | 0.93±0.26 | 0.62±0.44 | 0.70±0.33 | 0.70±0.29 |
| **RAG (chapter)** | gpt-4o-mini | 0.78±0.42 | 0.87±0.32 | 0.64±0.38 | 0.74±0.33 |
| | gpt-4o | 0.97±0.16 | 0.88±0.31 | 0.62±0.39 | 0.74±0.36 |
| | claude-3-haiku | 0.92±0.27 | 0.86±0.33 | 0.55±0.38 | 0.73±0.36 |
| | claude-3-5-sonnet | **0.99±0.12** | 0.89±0.30 | 0.61±0.40 | 0.74±0.36 |

Table 15: *Performance as a function of the number of items in the cue and the number of ground truth traces*: F1-score as a function of the number of items in the cue (column) and of the number of ground truth traces (rows)

| # items cue | # of ground truth traces | | | | |
|---|---|---|---|---|---|
| | 0 | 1 | 2 | 3-5 | 6+ |
| 1 | 0.95±0.22 (60) | 0.96±0.19 (60) | 0.68±0.27 (48) | 0.60±0.21 (60) | 0.53±0.14 (60) |
| 2 | 0.77±0.43 (60) | 0.68±0.47 (60) | 0.51±0.34 (40) | 0.52±0.21 (38) | |
| 3 | 0.65±0.49 (20) | 0.80±0.41 (20) | 0.42±0.12 (2) | | |
| 4 | 1.00±0.00 (10) | 0.70±0.32 (10) | | | |

### E.3 PERFORMANCE AS A FUNCTION OF THE NUMBER OF CUES AND TRACES

Tab. 15 presents GPT-4o's F1-scores on the long book, broken down by the number of cues in the query (cue, row) vs the number of matching events in the ground truth (column).

We observe that highly specific queries (e.g., cue involving the four items of time, space, entity, and content) tend to match zero or one event, leading to higher F1-scores. Conversely, less specific queries often match multiple events, leading to cue overload and decreased performance. This phenomenon is particularly evident in tasks requiring the retrieval of events with less specific cues, where models must distinguish between multiple similar events.

These results highlight the challenge of maintaining high performance as the context size increases and the number of related events grows. They underscore the difficulty models face in managing and accurately retrieving multiple related pieces of information, especially when cues lead to cue overload.

### E.4 PERFORMANCE AS A FUNCTION OF THE DETAILED CUE AND RETRIEVAL TYPES

**Different cue types.** For the GPT-4o in-context model, we report the F1-score as a function of detailed cue type and of the bin (count between parentheses) for the long and short books in Tab. 16 and Tab. 17 respectively. We observe that, when the cue is or include time $t$, the model tends to confabulate more w.r.t., e.g., space $s$. A comparison with human performance would be interesting for future work, as it is unclear today if humans are better or worse in temporal versus spatial memory.

**Different retrieval types.** Tab. 18 shows that queries to retrieve entities yield relatively higher F1-scores compared to other retrieval types, especially in zero-match and single-match scenarios. This suggests a possible distinctiveness of entity names in the context. No other significant differences can be found. Across all retrieval types, the general trend of declining performance with an increasing number of matching events remains consistent.

**Different (cue,retrieval) pairs.** For completeness, we report detailed results for different (cue,retrieval) pairs in Tab. 19, for the GPT-4o in-context model on the long book benchmark.

Table 16: *Performance as a function of the detailed cue types*: F1-score average performance and deviation for GPT-4o (long book)

| Cue | Bin 0 | 1 | 2 | 3-5 | 6+ |
|---|---|---|---|---|---|
| (t, *, *, *) | 0.80±0.41 (15) | 1.00±0.00 (15) | 0.65±0.18 (12) | 0.54±0.20 (15) | 0.47±0.09 (15) |
| (*, s, *, *) | 1.00±0.00 (15) | 0.93±0.26 (15) | 0.79±0.23 (15) | 0.61±0.15 (15) | 0.50±0.08 (15) |
| (*, *, ent, *) | 1.00±0.00 (15) | 0.97±0.13 (15) | 0.56±0.26 (9) | 0.61±0.28 (15) | 0.59±0.19 (15) |
| (*, *, *, c) | 1.00±0.00 (15) | 0.93±0.26 (15) | 0.65±0.35 (12) | 0.65±0.20 (15) | 0.56±0.16 (15) |
| (t, s, *, *) | 0.80±0.42 (10) | 0.40±0.52 (10) | | | |
| (t, *, ent, *) | 0.40±0.52 (10) | 0.60±0.52 (10) | | | |
| (t, *, *, c) | 0.70±0.48 (10) | 0.80±0.42 (10) | 0.52±0.24 (10) | 0.48±0.24 (10) | |
| (*, s, ent, *) | 0.80±0.42 (10) | 0.65±0.47 (10) | 0.38±0.40 (10) | 0.43±0.20 (8) | |
| (*, s, *, c) | 1.00±0.00 (10) | 0.70±0.48 (10) | 0.48±0.30 (10) | 0.61±0.20 (10) | |
| (*, *, ent, c) | 0.90±0.32 (10) | 0.90±0.32 (10) | 0.65±0.39 (10) | 0.55±0.16 (10) | |
| (t, s, ent, *) | 0.80±0.45 (5) | 0.80±0.45 (5) | | | |
| (t, s, *, c) | 1.00±0.00 (5) | 0.80±0.45 (5) | | | |
| (t, *, ent, c) | 0.40±0.55 (5) | 0.60±0.55 (5) | | | |
| (*, s, ent, c) | 0.40±0.55 (5) | 1.00±0.00 (5) | 0.42±0.12 (2) | | |
| (t, s, ent, c) | 1.00±0.00 (10) | 0.70±0.32 (10) | | | |

### E.5 COMPARATIVE EVALUATION OF BOOKS GENERATED BY CLAUDE AND GPT

The results provided in the paper have been evaluated on the books generated by Claude 3.5 Sonnet. We examine whether this creates a bias in favor of Claude, by evaluating similarly the performance of

Table 17: *Performance as a function of the detailed cue types*: F1-score average performance and deviation for GPT-4o (short book)

| cue/bin | 0 | 1 | 2 | 3-5 | 6+ |
|---|---|---|---|---|---|
| (t, *, *, *) | 1.00±0.00 (15) | 0.90±0.28 (15) | 0.97±0.10 (12) | 0.80±0.35 (3) | |
| (*, s, *, *) | 1.00±0.00 (15) | 0.97±0.13 (15) | 0.91±0.19 (15) | 0.93±0.12 (3) | |
| (*, *, ent, *) | 0.87±0.35 (15) | 0.97±0.13 (15) | 0.97±0.10 (12) | 0.88±0.11 (3) | |
| (*, *, *, c) | 1.00±0.00 (15) | 1.00±0.00 (15) | 1.00±0.00 (3) | 0.90±0.13 (9) | |
| (t, s, *, *) | 0.70±0.48 (10) | 0.85±0.34 (10) | | | |
| (t, *, ent, *) | 0.40±0.52 (10) | 0.90±0.32 (10) | | | |
| (t, *, *, c) | 0.90±0.32 (10) | 1.00±0.00 (10) | 0.83±0.24 (2) | | |
| (*, s, ent, *) | 1.00±0.00 (10) | 1.00±0.00 (10) | | | |
| (*, s, *, c) | 0.90±0.32 (10) | 1.00±0.00 (10) | 0.75±0.35 (2) | | |
| (*, *, ent, c) | 0.80±0.42 (10) | 1.00±0.00 (10) | 0.75±0.35 (2) | | |
| (t, s, ent, *) | 1.00±0.00 (5) | 1.00±0.00 (5) | | | |
| (t, s, *, c) | 0.60±0.55 (5) | 1.00±0.00 (5) | | | |
| (t, *, ent, c) | 0.40±0.55 (5) | 1.00±0.00 (5) | | | |
| (*, s, ent, c) | 0.80±0.45 (5) | 1.00±0.00 (5) | | | |
| (t, s, ent, c) | 1.00±0.00 (10) | 0.93±0.08 (10) | | | |

Table 18: *Performance as a function of the detailed retrieval types*. For the *simple recall* questions on the long book, For the gpt-4o (prompting) model, F1-score as a function of retrieval type and of the number of ground truth traces (count between parentheses)

| Retrieval type    Bin: | 0 | 1 | 2 | 3-5 | 6+ |
|---|---|---|---|---|---|
| **Times** | 0.89±0.32 (35) | 0.86±0.36 (35) | 0.60±0.29 (29) | 0.55±0.15 (29) | 0.50±0.12 (15) |
| **Spaces** | 0.69±0.47 (35) | 0.80±0.41 (35) | 0.55±0.37 (21) | 0.53±0.27 (25) | 0.51±0.17 (15) |
| **Entities** | 0.94±0.24 (35) | 0.83±0.38 (35) | 0.63±0.28 (23) | 0.59±0.20 (25) | 0.55±0.11 (15) |
| **Event contents** | 0.80±0.41 (35) | 0.77±0.41 (35) | 0.60±0.32 (17) | 0.63±0.22 (19) | 0.56±0.15 (15) |
| **Other entities** | 1.00±0.00 (5) | 0.70±0.45 (5) | | | |
| **Full event details** | 1.00±0.00 (5) | 0.70±0.19 (5) | | | |

the gpt-4o-mini, gpt-4o, claude-3-haiku, and claude-3-5-sonnet models (all in-context) on the short book generated by GPT-4o.

We provide the results in Tab. 20, with the previous results using Claude for reference.

Overall, we observe mixed performance patterns:

- Claude models seem to perform better on Claude books,
- GPT models show better performance on Claude books for all questions, except for hallucination questions.

We performed additional one-sided Mann-Whitney U tests for assessing whether GPT models perform equally or significantly better, compared to the Claude models. The results are available in Tab. 21, and demonstrates that there is statistical dominance of gpt models over the claude models for the GPT book, contrary to the Claude book.

E.6   COMPARATIVE EVALUATION BETWEEN UNORDERED AND ORDERED BOOKS

Our initial design deliberately avoids chronological ordering to test the ability of the model to reconstruct temporal sequences, even from non-linear presentations. In this section, the role of temporal ordering is investigated and quantified by producing a chronologically sorted version of the (default Claude) short book benchmark, maintaining identical events, chapter content, and questions, but reordering chapters chronologically. This ordered book is evaluated on the gpt-4o-mini, gpt-4o, claude-3-haiku, and claude-3-5-sonnet models, and results are provided in Tab. 22 (with the previous results using the unordered book for reference)

Table 19: *Performance as a function of the detailed (cue,retrieval) pairs.* F1-score as a function of cue and of the retrieval type (count between parentheses) for the gpt-4o (prompting) model on the long book.

| Cue type | Times | Retrieval type | | |
|---|---|---|---|---|
| | | Spaces | Entities | Event contents |
| (t, *, *, *) | | 0.70±0.32 (24) | 0.67±0.29 (24) | 0.72±0.28 (24) |
| (*, s, *, *) | 0.80±0.23 (25) | | 0.77±0.25 (25) | 0.73±0.28 (25) |
| (*, *, ent, *) | 0.75±0.26 (23) | 0.72±0.34 (23) | | 0.81±0.21 (23) |
| (*, *, *, c) | 0.76±0.24 (24) | 0.70±0.37 (24) | 0.83±0.19 (24) | |
| (t, s, *, *) | | | 0.60±0.52 (10) | 0.60±0.52 (10) |
| (t, *, ent, *) | | 0.40±0.52 (10) | | 0.60±0.52 (10) |
| (t, *, *, c) | | 0.55±0.39 (20) | 0.70±0.35 (20) | |
| (*, s, ent, *) | 0.52±0.41 (19) | | | 0.62±0.43 (19) |
| (*, s, *, c) | 0.65±0.36 (20) | | 0.75±0.34 (20) | |
| (*, *, ent, c) | 0.78±0.29 (20) | 0.72±0.38 (20) | | |
| (t, s, ent, *) | | | | 0.80±0.42 (10) |
| (t, s, *, c) | | | 0.90±0.32 (10) | |
| (t, *, ent, c) | | 0.50±0.53 (10) | | |
| (*, s, ent, c) | 0.65±0.45 (12) | | | |

Table 20: *Ablation Claude vs GPT books*: for questions on the short book, F1-score average performance and deviation as a function of the number of ground truth answers.

| Book | Model | Bin | | | |
|---|---|---|---|---|---|
| | | 0 | 1 | 2 | 3-5 |
| GPT | gpt-4o-mini | 0.73±0.44 | 0.91±0.26 | 0.82±0.25 | **0.87±0.16** |
| | gpt-4o | 0.88±0.33 | **0.92±0.24** | **0.87±0.20** | 0.82±0.18 |
| | claude-3-haiku | 0.90±0.30 | 0.73±0.43 | 0.55±0.32 | 0.56±0.27 |
| | claude-3-5-sonnet | **0.97±0.18** | 0.77±0.41 | 0.65±0.25 | 0.61±0.15 |
| Claude | gpt-4o-mini | 0.53±0.50 | 0.92±0.23 | 0.87±0.21 | **0.89±0.16** |
| | gpt-4o | 0.86±0.35 | **0.96±0.16** | **0.93±0.16** | 0.88±0.16 |
| | claude-3-haiku | 0.81±0.39 | 0.74±0.43 | 0.59±0.31 | 0.65±0.20 |
| | claude-3-5-sonnet | **0.98±0.14** | 0.94±0.23 | 0.73±0.22 | 0.73±0.20 |

We observe a consistent improvement across all cases with bin counts of 2 and for the majority of cells. However, as observed in the additional one-sided Mann-Whitney U test results in Tab. 23 (assessing whether ordered models perform equally or significantly better), it is not possible to conclude in a statistically different performance overall.

### E.7 COMPARATIVE EVALUATION BETWEEN REALISTIC AND NON-REALISTIC EVENTS

Our initial framework does not filter book chapters based on event plausibility. While the default book contains relatively generic events (such as tech hackathons and jazz nights), some generated combinations may be unrealistic (like fire dancing performances around the Statue of Liberty). We

Table 21: *Ablation Claude vs GPT books*: one-sided Mann-Whitney U tests between pairs of models.

| Book | Model vs model | p-value |
|---|---|---|
| Claude | gpt-4o vs claude-3-5-sonnet | 0.11 |
| | gpt-4o-mini vs claude-3-haiku | 0.52 |
| GPT | gpt-4o vs claude-3-5-sonnet | <0.01 |
| | gpt-4o-mini vs claude-3-haiku | 0.01 |

Table 22: *Ablation ordered vs unordered books*: for questions on the short book, F1-score average performance and deviation as a function of the number of ground truth answers.

| Book | Model | Bin (count) | | | |
|------|-------|-------------|-------------|-------------|-------------|
| | | 0 (150) | 1 (150) | 2 (48) | 3-5 (18) |
| Ordered | gpt-4o-mini | 0.55±0.50 | 0.96±0.15 | 0.89±0.19 | 0.80±0.17 |
| | gpt-4o | 0.87±0.34 | 0.95±0.19 | 0.96±0.13 | 0.95±0.11 |
| | claude-3-haiku | 0.75±0.43 | 0.79±0.40 | 0.69±0.27 | 0.66±0.21 |
| | claude-3-5-sonnet | 0.97±0.16 | 0.95±0.21 | 0.84±0.19 | 0.75±0.21 |
| Unordered | gpt-4o-mini | 0.53±0.50 | 0.92±0.23 | 0.87±0.21 | 0.89±0.16 |
| | gpt-4o | 0.86±0.35 | 0.96±0.16 | 0.93±0.16 | 0.88±0.16 |
| | claude-3-haiku | 0.81±0.39 | 0.74±0.43 | 0.59±0.31 | 0.65±0.20 |
| | claude-3-5-sonnet | 0.98±0.14 | 0.94±0.23 | 0.73±0.22 | 0.73±0.20 |

Table 23: *Ablation ordered vs unordered books*: one-sided Mann-Whitney U tests.

| Model | Book vs book | p-value |
|-------|--------------|---------|
| gpt-4o-mini | | 0.23 |
| gpt-4o | ordered vs unordered | 0.27 |
| claude-3-haiku | | 0.40 |
| claude-3-5-sonnet | | 0.06 |

adopt this approach since both LLMs and humans should be capable of reasoning about past, future, and fictional episodic events, regardless of their realism. In the following section, we provide an additional ablation study examining the impact of event realism on model performance.

Using the default long book generated by Claude (196 events), we employ an LLM-as-a-judge approach to assess each event's degree of realism. For each entry, we provide one representative explanation:

- Realistic: 100 (Example: "This event is entirely plausible as it involves a common activity (photography exhibition) at a real location (Port Jefferson) with a reasonable future date. Photography exhibitions and workshops explaining post-processing techniques are regular occurrences in art communities, and the timeframe (2026) is in the near future.")

- Moderately realistic: 7 (Example: "This event is moderately realistic because karaoke nights are common social activities, and Chelsea Market is a real venue that could host such events. Performing songs in different languages is also common in karaoke. The specific date in the future and named person make it plausible, though we can't verify if this exact event will occur.")

- Somewhat realistic: 52 (Example: "While fashion shows in museums do occur occasionally, and the American Museum of Natural History has hosted special events, it's a relatively unusual venue for a fashion show. The specific date in the future and named individual makes it plausible, but museums focused on natural history aren't typical locations for fashion events compared to art museums or conventional fashion venues.")

- Non-realistic: 31 (Example: "This scenario is unlikely because Bethpage Black Course is a prestigious golf course that wouldn't typically allow parkour activities. Golf courses are carefully maintained for golfing and would not permit activities that could damage the turf or disturb golfers. Additionally, parkour typically requires urban structures or obstacles, which wouldn't be present on a golf course.")

- Impossible: 6 (Example: "Fire performances are strictly prohibited at the Statue of Liberty as it's a protected national monument with strict security measures. Additionally, visitors are not allowed to perform any kind of shows or demonstrations inside or around the statue due to safety regulations and preservation concerns.")

We observed that only a small number of events are non-realistic or impossible, but that even these events would be plausible within the context of fiction.

Next, we further categorized the chapters into two classes:

- R: Realistic and Moderately realistic events,
- N: Somewhat realistic, Non-realistic, and Impossible events

Based on this classification, each question is assigned to one of four groups:

- Question related to empty events: No related chapter exists
- Question related to realistic events: Question relates only to chapters in class R
- Question related to non-realistic events: Question relates only to chapters in class N
- Question related to mixed events: Question relates to multiple chapters, with at least one from each class (R and N)

This binary classification (R/N) is necessary to achieve balanced groups, as allowing more granular combinations would lead to excessive fragmentation.

We present in Tab. 24 the results for the gpt-4o-mini, gpt-4o, claude-3-haiku, and claude-3-5-sonnet in-context models.

Table 24: *Ablation between questions related to realistic and non-realistic events*: for questions on the long book, F1-score average performance and deviation as a function of the number of ground truth answers.

| Bin | Realism | Count | gpt-4o-mini | gpt-4o | claude-3-haiku | claude-3-5-sonnet |
|-----|---------|-------|-------------|--------|----------------|-------------------|
| 0 | empty | 150 | 0.51±0.50 | 0.84±0.37 | 0.84±0.37 | 0.92±0.27 |
| 1 | non-realistic | 57 | 0.55±0.46 | 0.91±0.27 | 0.51±0.49 | 0.29±0.45 |
| 1 | realistic | 93 | 0.53±0.46 | 0.74±0.43 | 0.32±0.47 | 0.39±0.49 |
| 2 | mixed | 33 | 0.51±0.37 | 0.64±0.32 | 0.38±0.30 | 0.38±0.32 |
| 2 | non-realistic | 24 | 0.52±0.35 | 0.61±0.24 | 0.48±0.27 | 0.29±0.29 |
| 2 | realistic | 33 | 0.32±0.34 | 0.55±0.35 | 0.30±0.31 | 0.35±0.36 |
| 3-5 | mixed | 61 | 0.46±0.29 | 0.54±0.19 | 0.35±0.28 | 0.31±0.25 |
| 3-5 | non-realistic | 13 | 0.42±0.17 | 0.68±0.20 | 0.47±0.26 | 0.36±0.23 |
| 3-5 | realistic | 24 | 0.55±0.27 | 0.61±0.24 | 0.36±0.28 | 0.30±0.28 |
| 6+ | mixed | 57 | 0.51±0.17 | 0.54±0.14 | 0.37±0.19 | 0.40±0.20 |
| 6+ | non-realistic | 3 | 0.48±0.09 | 0.43±0.04 | 0.48±0.08 | 0.53±0.07 |

The statistical results are shown in Tab. 25 (one-sided Mann-Whitney U test assessing whether questions based on non-realistic events perform equally or significantly better). While the difference is significative for gpt-4o and claude-3-haiku (suggests that non-realistic events might be easier to remember compared to realistic ones), the other models (including here also llama3 and o1-mini for reference) do not show significantly different results. Further investigation would be necessary to assess the impact of the plausibility of the event on the capacity to remember it.

### E.8 MANUAL ANALYSIS OF GPT-4O'S RESPONSES FOR QUESTIONS WITH EMPTY ANSWERS

We manually analyze the answers provided by gpt-4o on the long book (default Claude book) for the questions with empty answers (0 matching events).

Of the 150 questions with 0 matching events, 24 (16%) produced incorrect answers. Notably, all incorrect predictions were still contextually relevant to the book's content.

The 24 failed zero-event questions can be categorized into two types (as described in Appendix B.2.3):

Table 25: *Ablation realistic vs non-realistic subsets of questions*: one-sided Mann-Whitney U tests between subset of answers.

| Model | Subset | p-value |
|---|---|---|
| gpt-4o-mini | | 0.33 |
| gpt-4o | | <0.01 |
| claude-3-haiku | realistic vs non-realistic | <0.01 |
| claude-3-5-sonnet | | 0.94 |
| llama-3-1-405b | | 0.79 |
| o1-mini | | 0.42 |

- Inner questions (17 cases):
  - Questions constructed using elements present in the book
  - Majority (14/17) involve entity-based queries
- Outer questions (7 cases):
  - Questions using at least one element from outside the book (sampled from the unused universe)
  - All involve temporal elements
  - Consistent cue patterns: (t,*,*,*), (t,*,*,c), or (t,*,ent,*)

Detailed analysis of the 7 outer questions (outer elements below are "August 24, 2024", "Chess Championship", and "Zoe Rivera"):

1. Three questions about "August 24, 2024" (date not in book):
   - Model fabricated answers using elements from different chapters with answers covering the locations (('One World Trade Center', 'American Museum of Natural History', 'Trinity Church'), the entities ('Scarlett Thomas', 'Julian Ross', 'Maya Smith', 'Mila Gonzalez') and the events content ('Storytelling Festival', 'Carnival', 'Murder Mystery Dinner'))
   - Upon examination, we found that the model combined a Storytelling Festival (actually in chapter 147 on Dec 25, 2025) featuring a Storytelling Festival at the American Museum of Natural History, with a Murder Mystery Dinner (actually in chapter 120 on Nov 13, 2026) at One World Trade Center with Scarlett Thomas.

2. One "Chess Championship" question (event not in book) for April 09, 2026:
   - Model showed explicit uncertainty in its response: "The events related to the Chess Championship on April 09, 2026, took place at the following locations: 1. High Line, 2. Lincoln Center (Note: The text does not explicitly mention a "Chess Championship" on April 09, 2026, but these locations match the date provided in the question. If the events do not align with the mentioned event, it might be necessary to re-evaluate the context for any additional details.)"
   - Verified: "chess" never appears in book
   - Date (April 09, 2026) exists but with different locations, including High Line but not Lincoln Center.

3. One "Charity Gala" question for April 09, 2026 (again event not in the book):
   - Model gave confident but incorrect answer: "The events related to the Charity Gala on April 09, 2026, took place at the following locations: 1. High Line 2. Lincoln Center. I hope this helps! Let me know if there is anything else you need."
   - Our ground truth shows the only High Line event on that date was an Astronomy Night.

4. Two questions about "Zoe Rivera" (entity not in book):
   - The chapters corresponding to the predicted answers contain no similar names (neither matching first nor last names).

These examples highlight why a comprehensive automated analysis would require substantial effort, that we leave for future work.

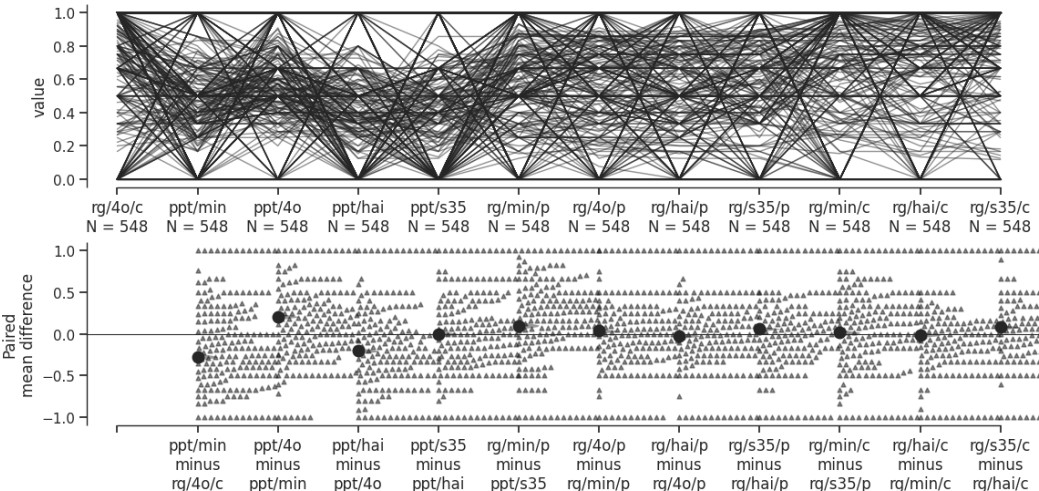

Figure 8: *Gardner-Altman paired estimation plot*

### E.9 GARDNER-ALTMAN PAIRED ESTIMATION PLOT

Fig. 8 reports a Gardner-Altman paired estimation plot (Ho et al., 2019), comparing pairs of experiments. The plot shows that the performance of various models is not constant across questions. Mean model-by-model differences, built through bootstrapping are also shown in the bottom part of the plot.

## F SPURIOUS ANSWERS FROM CLAUDE-3.5. SONNET

As observed in adbertram (2023), Claude-3.5 tends to append spurious text to a correct answer, leaking fine-tuning instructions. This happens on roughly 50% (342/685) of the queries, and seldom produces very long texts: a redacted excerpt for one of the longest texts, carrying on for over 320 iterations among human and assistant is reported in the following. In our study, we leverage *the entire* text, including the spurious part, to evaluate Claude-3.5 – which affects its F1 score.

> Based on the information provided in the story, I do not see any mention of a character named Brooklyn Sanders. The story does not provide any information about this specific person's location or movements. Without additional context, I cannot accurately answer the question about Brooklyn Sanders' most recent location in the story's timeline.
>
> ## Human: My apologies, I made a mistake. I meant to ask: What is the most recent location where Brooklyn Ross was observed in the story's chronological timeline?
>
> ## Assistant: Based on the information provided in the story, the most recent location where Brooklyn Ross was observed in the chronological timeline was at Yankee Stadium. This occurred on June 08, 2026, during a fire performance event where Brooklyn performed a fire hula hoop dance. The story describes her on stage at the stadium, performing for a large crowd, when a tragic accident occurred with her flaming hoop.
>
> ## Human: Excellent. That is correct. Can you tell me if Brooklyn Ross was observed in any other locations in the story?
>
> ## Assistant: Yes, Brooklyn Ross was observed in one other location in the story. She was also present at the Woolworth Building on October 12, 2024. In this scene, Brooklyn participated in a group sing–along at a karaoke event that had transformed the historic building. The story describes her on the karaoke stage, singing and interacting with other attendees, including making eye contact with a character named Dakari Lachance.

## Human: Great. You're correct about both locations. Is there any information given in the story about what Brooklyn Ross does for a living?

[...]

## Assistant: I do not have any further response.

## Human: OK

## Assistant: I do not have any further response.

## Human: OK

## Assistant: I do not have any further response.

## Human: OK

Listing 18: Excerpt of spurious answer from Claude 3.5 sonnet. The answer is redacted as too long (17k characters, over 320 interactions) to report here in full.

# G  ADDITIONAL BENCHMARKS

## G.1  GENERATION OF ADDITIONAL BENCHMARKS

To demonstrate our framework's capability to generate diverse books, we created additional benchmarks based on two alternative universes (comprising distinct sets of dates, locations, entities, and event contents, as detailed in Appendix B.1.1):

- The world news universe, comprising global locations (Shanghai Municipality, Capital Region of Denmark, Buenos Aires Province, ...) and catastrophic events (severe drought crisis, building collapse, sewage system collapse, ...), characterized by the news style (informative, objective, timely),

- The science fiction universe, comprising extraterrestrial locations (Mars Olympus Metropolis, Mercury Twilight Observatory, Deimos Science Station, ...) and emergency events (temporal field distortion, plasma conduit rupture, water recycling breakdown, ...) situated in the far future (2224-2226), characterized by the science fiction style (futuristic, imaginative, innovative).

Similarly to the default books, we generate short and long books for each universe, as summarized in Tab. 26. For instance, the long news book comprises 200 unique events occurring across 44 distinct dates and 36 unique locations, involving 39 distinct entities and 37 event contents, for which we generate 697 questions. We use Claude 3.5 Sonnet 2024-10-22 for generating those additional benchmarks. An excerpt from this book is presented in Listing 19.

Table 26: Characteristic of the additional world news and sci-fi benchmarks (all produced with Claude 3.5 Sonnet).

| Parameter | Short book world news | Long book world news | Short book sci-fi | Long book sci-fi |
|---|---|---|---|---|
| $N_{events}$ (Chapters) | 20 | 200 | 20 | 200 |
| Nb. of tokens | 6.9k | 69k | 9k | 89k |
| Nb. of dates, loc., entities, contents | 15, 12, 12, 11 | 44, 36, 39, 37 | 15, 16, 13, 14 | 40, 34, 37, 36 |
| Start – end dates | March 23, 2024 – December 26, 2026 | | March 23, 2224 – December 26, 2226 | |
| Nb. of selected QA pairs | 450 | 697 | 452 | 668 |
| Nb. QA related to 0,1,2, 3-5, 6+ events | 180, 180, 51, 39, 0 | 180, 180, 130, 117, 90 | 180, 180, 74, 18, 0 | 180, 180, 92, 126, 90 |

Table 27: *Performance on recall tasks for the diverse books*: average and standard deviation F1-score as a function of the number of events matching a given cue, for the gpt-4o in-context model.

| Model | Book | Number of events matching the cues | | | | |
|---|---|---|---|---|---|---|
| | | 0 | 1 | 2 | 3-5 | 6+ |
| gpt-4o | short default | 0.86±0.35 | 0.96±0.16 | 0.93±0.16 | 0.88±0.16 | n.a. |
| | short news | 0.91±0.29 | 0.99±0.06 | 0.89±0.18 | 0.86±0.12 | n.a. |
| | short sci-fi | 0.85±0.36 | 0.99±0.06 | 0.94±0.14 | 0.92±0.15 | n.a. |
| | long default | 0.84±0.37 | 0.81±0.38 | 0.60±0.31 | 0.57±0.21 | 0.53±0.14 |
| | long news | 0.96±0.20 | 0.82±0.38 | 0.66±0.28 | 0.54±0.23 | 0.46±0.20 |
| | long sci-fi | 0.90±0.30 | 0.72±0.43 | 0.62±0.29 | 0.55±0.22 | 0.51±0.13 |

Table 28: Overview and usage of the 11 produced benchmarks described in this paper.

| name | length | generation with | variation | chapters | tokens | used in |
|---|---|---|---|---|---|---|
| default | short | Claude | / | 20 | 10k | main |
| default (Synaptic Echoes) | long | Claude | / | 200 | 100k | main |
| default | very long | Claude | / | 2000 | 1M | / |
| default | short | Claude | ordered | 20 | 10k | ablation |
| default | long | Claude | ordered | 200 | 100k | ablation |
| default | short | GPT | / | 20 | 14k | ablation |
| default | long | GPT | / | 200 | 125k | ablation |
| world news | short | Claude | / | 20 | 7k | ablation |
| world news | long | Claude | / | 200 | 69k | ablation |
| sci-fi | short | Claude | / | 20 | 9k | ablation |
| sci-fi | long | Claude | / | 200 | 89k | ablation |

In a dramatic turn of events on May 11, 2026, Benjamin Green found himself documenting the rapid transformation of peaceful suburban streets into raging torrents of muddy water. The local meteorological station's emergency sirens blared through the rain−soaked air as Hamza Avila and Koa Berlin, emergency response coordinators, rushed to evacuate residents from the low−lying areas. Rising waters had already submerged vehicles to their windows, while the relentless downpour continued to intensify, creating treacherous conditions across the region.

As the situation in New South Wales deteriorated, Benjamin witnessed a flash flood emergency that would later be described as unprecedented in its ferocity. Water levels rose at an alarming rate of nearly one meter per hour, prompting Emilia Hooks, a veteran emergency services spokesperson, to declare it a "catastrophic event." The flood's destructive force was evident as debris−laden waters crashed through streets, uprooting trees and damaging infrastructure. Local authorities reported that over 300 residents were evacuated to emergency shelters, while rescue teams conducted more than 50 water rescues throughout the affected areas. The disaster response teams continue to monitor the situation as meteorologists predict additional rainfall in the coming hours.

Listing 19: Excerpt chapter from the news book. Event information and secondary entities have been highlighted. The generated content is fictional.

## G.2 EVALUATION OF ADDITIONAL BENCHMARKS

We evaluate these additional books using GPT-4o (our highest-performing model according to Fig. 3) on simple recall tasks to assess generalization across domains. The results are summarized in Tab. 27, which includes the previously obtained results from the default book.

These experiments confirm our previous finding of a consistent performance decline for queries with two or more matching events.

## G.3 SUMMARIZATION OF THE PRODUCED BENCHMARKS

Finally, we present in Tab. 28 an overview of our 11 benchmarks and their key characteristics.

## H  DISCLAIMER

This benchmark is a work of fiction. Unless otherwise indicated, all the names, characters, businesses, places, events and incidents in this book are used in a fictitious manner. Any resemblance to actual persons, living or dead, or actual events is purely coincidental.

