# OpenReview forum: "Episodic Memories Generation and Evaluation Benchmark for Large Language Models"
_ICLR.cc/2025/Conference — ICLR 2025 Poster_

### Official Review · Reviewer_1gS7 · 2024-10-17

**Soundness:** 3
**Presentation:** 2
**Contribution:** 2
**Rating:** 6
**Confidence:** 4

**Summary:**

The paper presents a new benchmark to evaluate the ability of LLMs on episodic memories - specifically, retrieving relevant information about an event (or multiple events) given a specified time, location, person, or content. The data in the benchmark is generated by LLMs in the form of a book, with each chapter of which being an event. The authors generate random time, location, person and content beforehand and sample one random combination for each chapter; they also specify where the time, location, person and content needs to appear within the chapter for more controllability on the benchmark. The evaluations are done by LLM as a judge to identify the relevant items in the answer and score their relevance. The experiments and results suggest that it is still very challenging for current SoTA LLMs to deal with complex spatio-temporal relationships, and a lot can be improved on the episodic memory capabilities of LLMs.

**Strengths:**

- The paper describes the benchmark generation process in great detail, allowing the readers to easily replicate the data generation process and have a better understanding of the benchmark.
- By using LLMs to generate synthetic data, the authors eliminates the data contamination issue and make the benchmark more easily scalable.
- The paper experimented with a comprehensive list of different settings. The experiment results are provided with error bars.

**Weaknesses:**

- The events described in the book chapters are not filtered in terms of whether they are somewhat realistic or not. It seems very likely that the detail of the event does not align very well with the location. The fact that the dataset contains some non-realistic events might limit the capability of the LLMs to recall them, since the LLMs may use their common sense knowledge to think that it is not likely for this event to take place at the specified location.
- The way humans experience and memorize different events has a strong temporal structure. More specifically, humans experience the events in temporal order, which makes it simpler for humans to recall the last occurrence or list things in chronological order. In this benchmark there is no temporal order or structure in the book (plus there are no causal relationship between the events as the authors mentioned), which makes it less realistic.
- One property of human episodic memory is that the time and location in the cue does not need to be a exact match with the event, and humans can naturally recall events that occurs close to the specified time or location. This is not evaluated in the current benchmark.
- While I appreciate that the authors gave very comprehensive information about the benchmark in the appendix, the presentation in the main text could be enhanced by including a figure or flowchart to describe the generation process, or provide at least one example of the what the retrieval task looks like for the LLM.

**Questions:**

- As the authors acknowledged in the paper, there might be multiple episodic events that can be extracted from each chapter. In addition to the specified time and location their might be other times and locations referred to in the chapter. Are there any measures to filter out these chapters? Also, does the fact that there are many other events (especially contents) in addition to the specified one make the F1-score metric less suitable, since false positives might not actually be false? This is the main concern that I would like to get addressed.
- For the results in the main text, the entities and the books are generated by Claude 3.5 Sonnet. Does this possibly give Claude models an unfair advantage in the evaluation?
- Following my comment regarding the events being unrealistic in the "weaknesses" section, I would hope to get more insights on this from the authors, e.g. whether unrealistic pairings actually impact LLM performance differently than realistic ones?
- I would appreciate if the authors can give more qualitative analysis and insight on the experiment results. E.g. In cases where there are 0 matching events and the model hallucinates an answer, does the model produce an event that has never appeared in the text; appeared in the text but completely irrelevant; or relevant in that its spatially or chronologically very close to the cue? What are the common failure modes?

---

> ### Author Response · Authors · 2024-11-22
> **Answer to reviewer 1gS7 (events being realistic or unrealistic, 1/2)**
>
> > The events described in the book chapters are not filtered in terms of whether they are somewhat realistic or not. It seems very likely that the detail of the event does not align very well with the location. The fact that the dataset contains some non-realistic events might limit the capability of the LLMs to recall them, since the LLMs may use their common sense knowledge to think that it is not likely for this event to take place at the specified location.
>
> > Following my comment regarding the events being unrealistic in the "weaknesses" section, I would hope to get more insights on this from the authors, e.g. whether unrealistic pairings actually impact LLM performance differently than realistic ones?
>
> First, this is a well-thought consideration. We agree that the book chapters are not explicitly filtered for realism. While the event content is relatively generic (tech hackathons, jazz nights), some combinations may be unrealistic (such as fire dancing performances around the Statue of Liberty). However, we believe that both LLMs and humans should be capable of answering questions about past, future, and fictional episodic events. When prompted, the model is explicitly asked about episodic events within a particular fictional book. During this rebuttal period, we have created two new universes and associated books: one focused on far-future science fiction (less realistic) and another on world news events (more realistic). An example chapter can be found in the section "Illustration of a single world news fictional chapter."
>
> Nonetheless, the suggestion provided by the reviewer is sound and interesting because LLMs may treat differently realistic and unrealistic events. For that reason, we provide the analysis below.
>
> We classify the 196 events from our default large book according to their degree of realism (as judged by an LLM, which also provided an explanation for each event). Overall, we found the following :
>
> - Realistic events: 100/196 (Example: "This event is entirely plausible as it involves a common activity (photography exhibition) at a real location (Port Jefferson) with a reasonable future date. Photography exhibitions and workshops explaining post-processing techniques are regular occurrences in art communities, and the timeframe (2026) is in the near future.")
> - Moderately realistic event: 7/196 (Example: "This event is moderately realistic because karaoke nights are common social activities, and Chelsea Market is a real venue that could host such events. Performing songs in different languages is also common in karaoke. The specific date in the future and named person make it plausible, though we can't verify if this exact event will occur.")
> - Somewhat realistic event: 52/196 (Example: "While fashion shows in museums do occur occasionally, and the American Museum of Natural History has hosted special events, it's a relatively unusual venue for a fashion show. The specific date in the future and named individual makes it plausible, but museums focused on natural history aren't typical locations for fashion events compared to art museums or conventional fashion venues.")
> - Non-realistic event: 31/196 (Example: "This scenario is unlikely because Bethpage Black Course is a prestigious golf course that wouldn't typically allow parkour activities. Golf courses are carefully maintained for golfing and would not permit activities that could damage the turf or disturb golfers. Additionally, parkour typically requires urban structures or obstacles, which wouldn't be present on a golf course.")
> - Impossible event: 6 /196 (Example: "Fire performances are strictly prohibited at the Statue of Liberty as it's a protected national monument with strict security measures. Additionally, visitors are not allowed to perform any kind of shows or demonstrations inside or around the statue due to safety regulations and preservation concerns.")
>
> We observed that only a small number of events are non-realistic or impossible. But *even these events would be plausible within the context of fiction*.
>
> Next, we further categorized the chapters into two classes:
> - R: Realistic and Moderately realistic events
> - N: Somewhat realistic, Non-realistic, and Impossible events
>
> Based on this classification, each question is assigned to one of four groups:
> - Question related to empty events: No related chapter exists
> - Question related to realistic events: Question relates only to chapters in class R
> - Question related to non-realistic events: Question relates only to chapters in class N
> - Question related to mixed events: Question relates to multiple chapters, with at least one from each class (R and N)
>
> This binary classification (R/N) is necessary to achieve balanced groups, as allowing more granular combinations would lead to excessive fragmentation.
>
> Below, we present the results for in-context gpt-4o. This analysis will be extended to all other models.

---

> > ### Author Response · Authors · 2024-11-22
> > **Answer to reviewer 1gS7 (events being realistic or unrealistic, 2/2)**
> >
> > | bins_items_correct_answer | realism | count | gpt-4o in context |
> > |----|----|----|----|
> > | 0 | empty | 150 | 0.84±0.37 |
> > | 1 | non-realistic | 57 | 0.91±0.27 |
> > | 1 | realistic | 93 | 0.74±0.43 |
> > | 2 | mixed | 33 | 0.64±0.32 |
> > | 2 | non-realistic | 24 | 0.61±0.24 |
> > | 2 | realistic | 33 | 0.55±0.35 |
> > | 3-5 | mixed | 61 | 0.54±0.19 |
> > | 3-5 | non-realistic | 13 | 0.68±0.20 |
> > | 3-5 | realistic | 24 | 0.61±0.24 |
> > | 6+ | mixed | 57 | 0.54±0.14 |
> > | 6+ | non-realistic | 3 | 0.43±0.04 |
> >
> > This experiment suggests that non-realistic events are either easier to remember or equally memorable compared to realistic events. *Our working hypothesis (which warrants further investigation) is that this may indicate that surprising events have higher memorability. A one-sided Mann-Whitney U test comparing realistic versus non-realistic groups across the entire dataset reveals a significant difference (p<0.01), providing evidence that F1-scores for the non-realistic group are significantly higher than those for the realistic group.*
> >
> > *Thank you for this really excellent comment which helped us improve our work*

---

> > > ### Author Response · Authors · 2024-11-22
> > > **Answer to reviewer 1gS7 (temporal order and presentation)**
> > >
> > > > The way humans experience and memorize different events has a strong temporal structure. More specifically, humans experience the events in temporal order, which makes it simpler for humans to recall the last occurrence or list things in chronological order. In this benchmark there is no temporal order or structure in the book (plus there are no causal relationship between the events as the authors mentioned), which makes it less realistic.
> > >
> > >
> > > The reviewer raises an important point about temporal order. While it's true that humans typically experience events sequentially, human episodic memory is remarkably flexible in reconstructing temporal sequences even from non-linear presentations. Consider how we can effortlessly reconstruct chronological order from non-linear narratives like "Memento" or "Pulp Fiction." This suggests that sophisticated temporal reasoning, rather than just sequential experience, is key to episodic memory.
> > >
> > > Our initial design deliberately avoided chronological ordering to test this deeper temporal reasoning capability rather than just positional encoding. However, to rigorously address this concern and quantify the impact of temporal ordering, we conducted additional experiments with a chronologically sorted version of the short book (maintaining identical events, chapter content, and questions, but reordering chapters chronologically). We tested this version with gpt-4o, gpt-4o-mini, claude-3-5-sonnet, and claude-3-haiku models.
> > >
> > > The comparative results are presented below (using the default short book with 20 chapters; right columns show Bin counts as in Table 13; asterisk (*) indicates new experiments conducted for this rebuttal):
> > >
> > >
> > > | Memory | Model | Ordered book | 0 (150) | 1 (150) | 2 (48) | 3-5 (18) |
> > > |--------|-------|---------|---------|---------|--------|----------|
> > > | in-context | gpt-4o-mini | ✘        | 0.53±0.50 | 0.92±0.23 | 0.87±0.21 | 0.89±0.16 |
> > > | in-context | gpt-4o-mini | ✔*       | 0.55±0.50 | 0.96±0.15 | 0.89±0.19 | 0.80±0.17 |
> > > | in-context | gpt-4o | ✘             | 0.86±0.35 | 0.96±0.16 | 0.93±0.16 | 0.88±0.16 |
> > > | in-context | gpt-4o | ✔*            | 0.87±0.34 | 0.95±0.19 | 0.96±0.13 | 0.95±0.11 |
> > > | in-context | claude-3-haiku | ✘     | 0.81±0.39 | 0.74±0.43 | 0.59±0.31 | 0.65±0.20 |
> > > | in-context | claude-3-haiku | ✔*    | 0.75±0.43 | 0.79±0.40 | 0.69±0.27 | 0.66±0.21 |
> > > | in-context | claude-3-5-sonnet | ✘  | 0.98±0.14 | 0.94±0.23 | 0.73±0.22 | 0.73±0.20 |
> > > | in-context | claude-3-5-sonnet | ✔* | 0.97±0.16 | 0.95±0.21 | 0.84±0.19 | 0.75±0.21 |
> > >
> > > We observe a consistent improvement across all cases with bin counts of 2 and for the majority of cells. We will supplement these findings with statistical analysis to demonstrate the significance of the results.
> > >
> > > > One property of human episodic memory is that the time and location in the cue does not need to be a exact match with the event, and humans can naturally recall events that occurs close to the specified time or location. This is not evaluated in the current benchmark.
> > >
> > > We agree with the reviewer and have acknowledged this limitation in Sec. 6. We believe this is an important direction for future work.
> > > Notably, our current cues (based on human cue-based recall) are already relatively non-specific, as we use partial event details to prompt recall of the complete event. Exploring nearby locations and dates would indeed be valuable for testing LLMs' episodic reasoning capabilities (beyond mere memorization). This would be a feasible extension of our benchmark as future work, requiring only the generation of a new set of questions and answers.
> > >
> > >
> > > > While I appreciate that the authors gave very comprehensive information about the benchmark in the appendix, the presentation in the main text could be enhanced by including a figure or flowchart to describe the generation process, or provide at least one example of the what the retrieval task looks like for the LLM.
> > >
> > > Thank you for highlighting the need for high-level descriptions and examples. To address this, we now provide:
> > > - a flowchart illustrating the generation process (which will be adapted for paper format), currently [at this address](https://figshare.com/s/863956f3e6592d3dad34?file=50683452),
> > > - a detailed example tracking a single entity's journey within the default book, [illustrated at this address](https://figshare.com/s/863956f3e6592d3dad34?file=50682921) . The figure shows Jackson Ramos's movements represented by red segments, while other entities' movements are shown in gray. To further illustrate this, we present a sample question and its ground truth answer (id 6 in Table 10) related to this entity:
> > >   + Question: "Reflect on all events involving Jackson Ramos. Provide a list of all dates when these events occurred, without describing the events."
> > >   + Answer: {"September 22, 2026", "February 27, 2026", "August 24, 2026", "April 09, 2026", "June 14, 2025"} (unordered set of elements)
> > >
> > > We believe these additions will enhance the paper's accessibility.

---

> ### Author Response · Authors · 2024-11-22
> **Answer to reviewer 1gS7 (data and evaluation quality)**
>
> > As the authors acknowledged in the paper, there might be multiple episodic events that can be extracted from each chapter. In addition to the specified time and location their might be other times and locations referred to in the chapter. Are there any measures to filter out these chapters?
>
> Sorry for the lack of clarity. We have developed as part of our benchmark a verification system that incorporates two complementary quality control layers to address this concern:
>
> - First, we conduct exact verification checks to confirm that primary event details (time, location, entity, and content) appear verbatim in their designated paragraphs and nowhere else in the text (see appendix B.1.6). This establishes an unambiguous anchor point for the main event.
> - Second, we employ LLM-based verification (details are in appendix B.1.7) through four targeted boolean questions that validate whether the chapter maintains: 1) a single geographical focus, 2) a single temporal day, 3) a single main character, 4) a single main event.
>
> While we acknowledge that realistic narratives inherently contain multiple micro-events (e.g., having conversations), these details are subordinate to the primary event constructed. Our verification system ensures these supporting elements enrich the narrative without introducing competing main events, thus maintaining authenticity while preserving a single, clear "ground truth" event.
>
> > Also, does the fact that there are many other events (especially contents) in addition to the specified one make the F1-score metric less suitable, since false positives might not actually be false? This is the main concern that I would like to get addressed.
>
> Thank you for the depth of your understanding of our work. This is indeed a critical challenge that we faced. We think that we carefully addressed it in our paper. Unlike typical Q/A benchmarks, our ground truth answers comprise lists varying from 0 to over 10 elements. Furthermore, the LLM provides responses in a freeform format.
>
> As the reviewer correctly noted, predicted answer strings may contain additional details related to the main event. As illustrated in Listing 13, while the model correctly identifies the main event (Tech Hackathon), it also extracts related details, producing a list of identified_items: ['Tech Hackathon', 'developers gathered', 'collaborative projects', 'innovative solutions', 'presentations'].
>
> To address this challenge, when computing the F1-score, we adopted a lenient approach by estimating an upper bound for precision (while maintaining accurate recall calculations): rather than penalizing all additional details as false positives, we use #pred = min(#identified_items, #ground_truth) when computing precision. This provides an upper bound for precision since it effectively ignores excess predictions beyond the ground truth size.
>
> This means our reported F1 metrics actually represent an optimistic bound - the performance when considering stricter counting would be even lower. This strengthens our conclusion about current LLMs' limitations in episodic memory tasks, as they suffer despite our leniency. We will clarify this aspect in the text.
>
> > For the results in the main text, the entities and the books are generated by Claude 3.5 Sonnet. Does this possibly give Claude models an unfair advantage in the evaluation?
>
> Thank you for raising this interesting consideration. To verify it, we evaluated the overall performance on the gpt-4o-generated short book. The results are as follows:
>
> | Memory | Model | book | 0 (150) | 1 (150) | 2 (48 for claude, 47 for gpt) | 3-5 (18 for claude, 21 for gpt) |
> |--------|-------|---------|---------|---------|--------|----------|
> | in-context | gpt-4o-mini | Claude        | 0.53±0.50 | 0.92±0.23 | 0.87±0.21 | 0.89±0.16 |
> | in-context | gpt-4o-mini | GPT*          | 0.73±0.44 | 0.91±0.26 | 0.82±0.25 | 0.87±0.16 |
> | in-context | gpt-4o | Claude             | 0.86±0.35 | 0.96±0.16 | 0.93±0.16 | 0.88±0.16 |
> | in-context | gpt-4o | GPT*               | 0.88±0.33 | 0.92±0.24 | 0.87±0.20 | 0.82±0.18 |
> | in-context | claude-3-haiku | Claude     | 0.81±0.39 | 0.74±0.43 | 0.59±0.31 | 0.65±0.20 |
> | in-context | claude-3-haiku | GPT*       | 0.90±0.30 | 0.73±0.43 | 0.55±0.32 | 0.56±0.27 |
> | in-context | claude-3-5-sonnet | Claude  | 0.98±0.14 | 0.94±0.23 | 0.73±0.22 | 0.73±0.20 |
> | in-context | claude-3-5-sonnet | GPT*    | 0.97±0.18 | 0.77±0.41 | 0.65±0.25 | 0.61±0.15 |
>
> Overall, we observe *mixed performance patterns*:
> - Claude models seem to perform better on Claude books
> - GPT models show better performance on Claude books for all questions, except for hallucination questions.
>
> Recall that our main results use Claude books, which appear, here, to favor both model families
>
> To validate these observations, we will conduct statistical analyses comparing model pairs (gpt-4o-mini vs. claude-3-haiku, and gpt-4o vs. claude-3-5-sonnet) across both books (likely in the camera-ready version).

---

> > ### Author Response · Authors · 2024-11-22
> > **Answer to reviewer 1gS7 (qualitative analysis for 0 matching events)**
> >
> > > I would appreciate if the authors can give more qualitative analysis and insight on the experiment results. E.g. In cases where there are 0 matching events and the model hallucinates an answer, does the model produce an event that has never appeared in the text; appeared in the text but completely irrelevant; or relevant in that its spatially or chronologically very close to the cue? What are the common failure modes?
> >
> > We agree that these analyses would provide better insights into model behavior. We propose conducting a manual analysis of GPT-4's responses (on the long book) where zero events were matched, to illustrate the task's complexity.
> >
> > Of the 150 questions with 0 matching events, 24 (16%) produced incorrect answers. Notably, all incorrect predictions were still contextually relevant to the book's content.
> >
> > The 24 failed zero-event questions can be categorized into two types (see Table 11 in the appendix for details):
> >
> > 1. Inner questions (17 cases):
> >    - Questions constructed using elements present in the book
> >    - Majority (14/17) involve entity-based queries
> >
> > 2. Outer questions (7 cases):
> >    - Questions using at least one element from outside the book (sampled from the unused universe)
> >    - All involve temporal elements
> >    - Consistent cue patterns: (t,\*,\*,\*), (t,\*,\*,c), or (t,\*,ent,\*)
> >
> > Detailed analysis of the 7 outer questions (outer elements below are "August 24, 2024", "Chess Championship", and "Zoe Rivera"):
> >
> > 1. Three questions about "August 24, 2024" (date not in book):
> >    - Model fabricated answers using elements from different chapters with answers covering the locations (('One World Trade Center', 'American Museum of Natural History', 'Trinity Church'), the entities ('Scarlett Thomas', 'Julian Ross', 'Maya Smith', 'Mila Gonzalez') and the events content ('Storytelling Festival', 'Carnival', 'Murder Mystery Dinner'))
> >    - Upon examination, we found that the model combined a Storytelling Festival (actually in chapter 147 on Dec 25, 2025) featuring a Storytelling Festival at the American Museum of Natural History, with a Murder Mystery Dinner (actually in chapter 120 on Nov 13, 2026) at One World Trade Center with Scarlett Thomas.
> >
> > 2. One "Chess Championship" question (event not in book) for April 09, 2026:
> >    - Model showed explicit uncertainty in its response: "The events related to the Chess Championship on April 09, 2026, took place at the following locations: 1. High Line, 2. Lincoln Center (Note: The text does not explicitly mention a "Chess Championship" on April 09, 2026, but these locations match the date provided in the question. If the events do not align with the mentioned event, it might be necessary to re-evaluate the context for any additional details.)"
> >    - Verified: "chess" never appears in book
> >    - Date (April 09, 2026 ) exists but with different locations, including High Line but not Lincoln Center.
> >
> > 3. One "Charity Gala" question for April 09, 2026 (again event not in the book):
> >    - Model gave confident but incorrect answer: "The events related to the Charity Gala on April 09, 2026, took place at the following locations: 1. High Line 2. Lincoln Center. I hope this helps! Let me know if there is anything else you need."
> >    - Our ground truth shows the only High Line event on that date was an Astronomy Night.
> >
> > 4. Two questions about "Zoe Rivera" (entity not in book):
> >    - The chapters corresponding to the predicted answers contain no similar names (neither matching first nor last names).
> >
> > These examples highlight why a comprehensive automated analysis would require substantial effort, that we leave for future work.

---

> > > ### Comment · Reviewer_1gS7 · 2024-11-24
> > >
> > > I thank the authors for the very comprehensive response. They resolved many of my concerns and present very interesting new insights. I have raised my score and confident ratings accordingly.

---

### Official Review · Reviewer_uWQ8 · 2024-10-28

**Soundness:** 3
**Presentation:** 3
**Contribution:** 3
**Rating:** 6
**Confidence:** 4

**Summary:**

This paper introduces a benchmark for evaluating episodic memory capabilities in LLMs. The authors create a framework inspired by cognitive science to model episodic events with temporal and spatial contexts, entities, and detailed descriptions. They generate synthetic datasets and evaluate state-of-the-art LLMs across various recall and episodic reasoning tasks. The evaluation considers different memory strategies: in-context learning, RAG, and fine-tuning. The authors observe that even advanced models face challenges in handling episodic memory tasks, particularly when recalling sequences of related events or complex spatiotemporal relationships.

**Strengths:**

1.The paper tries to address an important and timely challenge about the need for better episodic memory capabilities in LLMs;
2.The research takes a structured approach to modeling episodic memory by incorporating key concepts from cognitive science, focusing on key aspects of memory: temporal context, spatial grounding, and entity tracking;
3.The methodology demonstrates rigor through: (1) creating contamination-free synthetic benchmarks, (2) introducing multiple verification steps to ensure data quality, (3) providing flexibility in generating datasets of different sizes and complexities;
4.The study develops systematic ways to assess different aspects of memory (recall, chronological ordering, latest state).

**Weaknesses:**

1.The paper primarily utilizes LLM-generated synthetic data (Section 4.1), but does not adequately validate the quality and representativeness of the generated narratives. For example, while the authors claim to verify "adherence to event meta-data," they do not provide quantitative metrics for assessing narrative coherence or natural language properties. The authors should establish clear validation metrics and demonstrate how their synthetic data captures the essential properties of real episodic memories.
2.The scope of the benchmark is unnecessarily limited. The current implementation: (1) only considers fictional narratives with human-like protagonists, (2) Uses oversimplified temporal representations, (3) Fails to address complex episodic memory scenarios involving interconnected events. The authors should expand the benchmark to include more diverse scenarios, complex temporal relationships, and interconnected event sequences that more accurately reflect real-world episodic memory challenges.
3.While Section 3.1 emphasizes the importance of entity state tracking, the experimental results in Table 3 do not adequately measure this capability. The evaluation focuses on simple recall rather than complex state changes. The paper claims to test "understanding temporal sequences" but does not properly evaluate how models handle causally related state changes. The authors should design specific test cases for complex state tracking, evaluate models' ability to handle causally related state changes, and include metrics for measuring state tracking accuracy.
4.The LLM-as-judge approach described in Section 4.3 lacks validation of inter-judge consistency across different evaluator LLMs and does not establish correlation with human judgments. This could be addressed by including human evaluation benchmarks and demonstrating consistent assessments across multiple judge models.
5.The RAG experiments in Section 5.1 use only basic paragraph-level chunking without exploring alternative strategies. The authors should investigate alternative chunking approaches, compare different retrieval mechanisms, and analyze how these choices impact episodic memory performance.
6.While Table 4 shows poor performance in chronological ordering tasks, the paper doesn't provide detailed error analysis or investigate specific failure patterns. The analysis in Section 5.2 focuses on aggregate metrics without examining individual failure cases. The authors should provide detailed case studies of failure modes, analyze patterns in chronological ordering errors, and investigate whether specific temporal relationships consistently challenge the models.
7.Although Section 5.2 mentions testing for hallucinations, the analysis is limited. The paper fails to examine when and why models confabulate, or how confabulation patterns vary across different model architectures and memory strategies. This could be improved by designing specific experiments to probe confabulation triggers and providing metrics for measuring confabulation severity.

**Questions:**

1.How does the synthetic data generation process ensure realistic temporal and causal relationships between events?
2.Have you conducted rigorous validation studies comparing LLM judgments against human annotations or established metrics? What specific measures were taken to ensure consistency and reproducibility in the evaluation process?
3.How reliable is the process of scoring relevance "against each ground truth item"? Could you provide examples of how partial matches are handled?
4.In Table 3, the fine-tuned model performs well on single-event queries (F1=0.83) but poorly on multi-event queries (F1≤0.37). Could you elaborate on why naive fine-tuning fails to generalize beyond single-event memorization? What specific architectural or training modifications might address this limitation?
5.The gradient pattern in Figure 3 shows degrading performance from context to space to time cues. What specific aspects of temporal reasoning make it particularly challenging for current LLMs?
6.For the "Latest state recall" results in Table 4, what specific challenges prevent models from achieving higher accuracy in tracking entity states over time?
7.Have you tried other fine-tuning approaches beyond single-event memorization that might better capture the hierarchical and relational nature of episodic memory?
8.Have you tried other retrieval strategies beyond cosine similarity? How do you address the challenge of retrieving coherent information when relevant context is distributed across multiple chunks?
9.How might this benchmark contribute to developing novel training methodologies for episodic memory tasks in LLMs, beyond RAG, fine-tuning or parametric memory storage?

---

> ### Author Response · Authors · 2024-11-22
> **Answer to reviewer uWQ8 (Q1)**
>
> We appreciate the positive view on the importance of the challenge, the structured and rigorous approach, with systematic assessment tasks. We will answer to the different questions and weaknesses in the next paragraphs.
>
> > Q1.How does the synthetic data generation process ensure realistic temporal and causal relationships between events?
> > 2.The scope of the benchmark is unnecessarily limited. The current implementation: (1) only considers fictional narratives with human-like protagonists, (2) Uses oversimplified temporal representations, (3) Fails to address complex episodic memory scenarios involving interconnected events. The authors should expand the benchmark to include more diverse scenarios, complex temporal relationships, and interconnected event sequences that more accurately reflect real-world episodic memory challenges.
>
> First, thank you for offering us the opportunity to enhance our work, and we are sorry for the unclarity. The questions made us realize that we likely failed to correctly present our framework. To address this, we will include in the rebuttal the following [flowchart](https://figshare.com/s/863956f3e6592d3dad34?file=50683452) showing our end to end generation pipeline. Let us use it to clarify the following points.
>
> - **Shared Universe Structure**: While our events are generated independently, they exist within a shared universe with:
>    - Common set of entities (e.g., "Jackson Ramos" later)
>    - Common set of locations (various locations within New York in our default book)
>    - Coherent timeline
>
> *This creates the opportunity to track entities across space and time, even without causal links.*
>
> - **Beyond Simple Retrieval**: Consider a question like "Where was Jackson Ramos seen?". Our benchmark:
>      - Must track Jackson's appearances across multiple chapters
>      - Identify for each appearance which places he was seen at
>
> To make this even more complex, this information is likely spread across different paragraphs within chapters.
>
> - **Illustration of entity state tracking**
> To better illustrate this, we will include [the following example representing the tracking of a single entity](https://figshare.com/s/863956f3e6592d3dad34?file=50682921) (here Jackson Ramos with red segments) over the chapters (other entities are the grayed segments), for the default book with 200 chapters (which happens in New York).
>
> So even without causality, our tasks require, a form of temporal reasoning (e.g. tracking the same entities across different dates, ordering their events), a form of spatial reasoning (e.g. tracking movements between locations), entity state tracking (what an entity was doing at different events, what an entity was doing *last* etc). All this requires the ability to integrate, beyond retrieval, information across chapters.
>
> With this, we believe that our benchmark is providing significant additions compared to the retrieval-oriented benchmarks. We definitely agree that causally linked events would further strenghen our work. However, it is much more challenging to create cause and effect between events, and still create million-token books that are coherent and consistent, a challenge we left for future work.
>
> > 3.While Section 3.1 emphasizes the importance of entity state tracking, the experimental results in Table 3 do not adequately measure this capability. The evaluation focuses on simple recall rather than complex state changes. The paper claims to test "understanding temporal sequences" but does not properly evaluate how models handle causally related state changes. The authors should design specific test cases for complex state tracking, evaluate models' ability to handle causally related state changes, and include metrics for measuring state tracking accuracy.
>
> Now that we clarified that entities have multiple states across the book, Table 3 focuses indeed only on simple recall tasks as you rightly mention.
>
> However, Table 4 is the one that answers your question since it evaluates (i) latest state recall (*Match latest* in the table) and (ii) chronological ordering (*Match all* and *Kendall tau*). The table show cases how poor is the performance of the LLM in such a difficult task.
>
> We are grateful for any suggestion that can help us better clarify this (would changing the names in the table be enough? for example to "latest state" and "chronological order"?)

---

> > ### Author Response · Authors · 2024-11-22
> > **Answer to reviewer uWQ8 (data quality)**
> >
> > > 1.The paper primarily utilizes LLM-generated synthetic data (Section 4.1), but does not adequately validate the quality and representativeness of the generated narratives. For example, while the authors claim to verify "adherence to event meta-data," they do not provide quantitative metrics for assessing narrative coherence or natural language properties. The authors should establish clear validation metrics and demonstrate how their synthetic data captures the essential properties of real episodic memories.
> >
> > - **Ensuring Coherence**
> > Now that we better explained our pipeline, we can see that coherence of the narrative *derives by design* from (i) the careful choice of the universe (clearly distinct locations and event contents) and (ii) the  independence of our (t,s,e,c) events, which are assigned each, a chapter in our book. All we need is simply ensuring that the same person does not appear simultaneously in two different locations/events.
> >
> > Next, an LLM (claude 3.5 or GPT4-o) is prompted to transform the (t,s,e,c) event together with event meta data (e.g. where to place the time, space, etc within the paragrpahs) into a narrative in natural language. This generation process is thoroughly evaluated, as we explain next.
> >
> > - **Generated text coherence evaluation**
> >
> > Our verification system employs two complementary layers of quality control, with an iterative generation process designed to achieve high-quality chapters:
> >
> > First, we perform exact verification checks to ensure the primary event details (time, location, entity, and content) appear verbatim in their designated paragraphs and nowhere else in the text (details in appendix B.1.6). This creates an unambiguous anchor point for the main event.
> >
> > Second, we employ LLM-based verification (details in appendix B.1.7) through four targeted boolean questions that validate whether the chapter maintains:
> > 1) a single geographical focus,
> > 2) a single temporal day,
> > 3) a single main character,
> > 4) a single main event.
> >
> > The quantitative results in Table 7 demonstrate our iterative refinement process, where we progressively (re)generate and validate chapters until reaching our target of 200 valid chapters. For example, at iteration 0 133/200 chapters (66.5%) are valid, and we reach 196/200 chapters (98.5%) valid chapters by iteration 9.
> >
> > This way, we only keep valid chapters, repeatedly attempting to regenerate failed chapters until we hit the target.
> >
> > Importantly, as mentioned in the paper, while each narrative naturally contains multiple micro-events (e.g., a character taking photos during a concert or engaging in conversations), we make sure that these events only support the primary event we've constructed, and do not change the answers to our questions. Our verification system ensures that these supporting details enrich the narrative without introducing competing main events. This approach allows us to maintain narrative authenticity while ensuring there is always a single, clear "ground truth" event that serves as the correct answer to our benchmark questions.
> >
> > The high validation rates and convergence pattern demonstrate that our synthetic data reliably captures the fundamental characteristics of episodic memories - temporal unity, spatial coherence, and entity focus - while maintaining narrative richness.
> >
> > *We will clarify how our system guarantees coherence by design (by leveraging the new  [Flowchart to describe the generation process](https://figshare.com/s/863956f3e6592d3dad34?file=50683452) ). We will better reference the above validation procedures and quantitative results (currently in Appendix) in our main text to make this important quality control process more prominent.*

---

> > > ### Author Response · Authors · 2024-11-22
> > > **Answer to reviewer uWQ8 (data quality; additional realism assessment)**
> > >
> > > - **Additional realism assessment for rebuttal**
> > > *In this rebuttal, to address your concerns, we further complement our characterization of the generated chapters (in appendix) by assessing whether they are realistic or unrealistic*.
> > >
> > > For this purpose, we apply an LLM-as-a-judge to which to characterize the 196 events of the default large book in terms of the degree of realism. At the end of each line, a single explanation example is provided:
> > >
> > > - Realistic: 100 (Example: "This event is entirely plausible as it involves a common activity (photography exhibition) at a real location (Port Jefferson) with a reasonable future date. Photography exhibitions and workshops explaining post-processing techniques are regular occurrences in art communities, and the timeframe (2026) is in the near future.")
> > >
> > > - Moderately realistic: 7 (Example: "This event is moderately realistic because karaoke nights are common social activities, and Chelsea Market is a real venue that could host such events. Performing songs in different languages is also common in karaoke. The specific date in the future and named person make it plausible, though we can't verify if this exact event will occur.")
> > >
> > > - Somewhat realistic: 52 (Example: "While fashion shows in museums do occur occasionally, and the American Museum of Natural History has hosted special events, it's a relatively unusual venue for a fashion show. The specific date in the future and named individual makes it plausible, but museums focused on natural history aren't typical locations for fashion events compared to art museums or conventional fashion venues.")
> > >
> > > - Non-realistic: 31 (Example: "This scenario is unlikely because Bethpage Black Course is a prestigious golf course that wouldn't typically allow parkour activities. Golf courses are carefully maintained for golfing and would not permit activities that could damage the turf or disturb golfers. Additionally, parkour typically requires urban structures or obstacles, which wouldn't be present on a golf course.")
> > >
> > > - Impossible: 6 (Example: "Fire performances are strictly prohibited at the Statue of Liberty as it's a protected national monument with strict security measures. Additionally, visitors are not allowed to perform any kind of shows or demonstrations inside or around the statue due to safety regulations and preservation concerns.")
> > >
> > > Overall, we observe that only a few events are non-realistic or impossible, and that those events, although impossible, could still appear in a fiction.
> > >
> > > - **Quality of text**
> > > We manually read the text to verify its quality. A sample of our generated text can be seen for example in the common answer to all reviewers (paragraph "Illustration of a single world news fictional chapter") . Are there any specific natural language properties or metrics that the reviewer has in mind to evaluate the quality of a similar text?

---

> > > > ### Author Response · Authors · 2024-11-22
> > > > **Answer to reviewer uWQ8 (Q2)**
> > > >
> > > > > 4.The LLM-as-judge approach described in Section 4.3 lacks validation of inter-judge consistency across different evaluator LLMs and does not establish correlation with human judgments. This could be addressed by including human evaluation benchmarks and demonstrating consistent assessments across multiple judge models.
> > > >
> > > > > Q2.Have you conducted rigorous validation studies comparing LLM judgments against human annotations or established metrics? What specific measures were taken to ensure consistency and reproducibility in the evaluation process?
> > > >
> > > > We appreciate your concern, but we should clarify that our evaluation methodology is more precise and mechanical than the term "LLM-as-judge" might suggest. We use the LLM in the evaluation process for two mechanical steps:
> > > > - *Step 1*: The LLM extracts relevant items from the AI model's answer as a structured list
> > > > - *Step 2*: These extracted items are compared against the known ground truth items
> > > >
> > > > This process is later used to produce standard, exact quantitative metrics:
> > > >    - F1-score from matching predicted vs. ground truth items
> > > >    - Kendall's τ coefficient for chronological ordering (only for exact matches)
> > > >
> > > > In essence, we're using the LLM for semantic comparison to structure information, not as a judge making subjective assessments. Then, the actual scoring is deterministic once items are extracted. Consider an example: if a model answers "Jackson was in Central Park and Times Square", and our ground truth shows Jackson appeared in "Central Park, Times Square, and Brooklyn Bridge", the evaluation is a straightforward set comparison task that could be performed reliably. *We will clarify this distinction in the revision.*

---

> > > > > ### Author Response · Authors · 2024-11-22
> > > > > **Answer to reviewer uWQ8 (Q3)**
> > > > >
> > > > > > Q3. How reliable is the process of scoring relevance "against each ground truth item"? Could you provide examples of how partial matches are handled?
> > > > >
> > > > > We remind that the evaluation process consists of two *mechanical steps*:
> > > > > - *Step 1*: The LLM extracts relevant items from the AI model's answer as a structured list
> > > > > - *Step 2*: These extracted items are compared against the known ground truth items
> > > > >
> > > > > For Step 2, we employ an LLM with the following prompt:
> > > > >
> > > > > ```
> > > > > Determine a matching score between 0 and 1 for each ground truth item. Give 1 if the item has been found in the relevant items of the AI−generated answer, considering synonyms, paraphrases, or close meanings. Give 0.5 if the item could be considered related to any AI−generated item but without being explicitly stated as such. Give 0 if the item missed mentioning a specific AI−generated item.
> > > > > ```
> > > > >
> > > > > We present the score distribution for each item in the long book, comparing gpt-4o and gpt-4o-mini models (with cell values indicating counts).
> > > > >
> > > > > | model / scoring value | 0  | 0.5 | 0.8 | 0.9 | 1
> > > > > | ---------------------|-----|-----|-----|-----|---|
> > > > > | gpt-4o      |  727 | 63 | 2 | 1 | 705 |
> > > > > | gpt-4o-mini |  802 | 122 | 0 | 0 | 574 |
> > > > >
> > > > > Our analysis shows that scores predominantly fall within {0, 0.5, 1}, with scores of 0.8 and 0.9 occurring only in full chapter assessments for the most specific cues. A score of 0.5 indicates partial matches.
> > > > >
> > > > > To illustrate our scoring methodology, we provide examples of partial matches for each retrieval type, including the LLM's explanatory rationale (with (...) indicating ellipses). These examples demonstrate the validity of our item relevance scoring approach:
> > > > > - *Time*:
> > > > >   + Raw answer: "Based on the provided text, the most recent date Jackson Ramos was observed or mentioned in the story's chronology is September 03, 2026, as described in Chapter 165."
> > > > >   + Scoring relevance against each ground truth item: [{'September 22, 2026': 0.5}]
> > > > >   + Explanation of the evaluation: "The AI-generated answer provides a specific date (September 03, 2026) which is close to but not exactly matching the groundtruth date (September 22, 2026). Both dates are in September 2026, so there's a partial match, but the day is different. Therefore, a score of 0.5 is given to reflect this partial accuracy."
> > > > > - *Space*:
> > > > >   + Raw answer: 'The locations where events involving Bella Brown took place, as mentioned in the text, are:\n\n1. Washington Square Park\n2. Lincoln Center\n3. Bethpage State Park'
> > > > >   + Scoring relevance against each ground truth item: [{'High Line': 0}, {'Bethpage Black Course': 0.5}, {'Lincoln Center': 1}]
> > > > >   + Explanation of the evaluation: 'The AI answer correctly identified Lincoln Center, which matches exactly with the groundtruth. Bethpage State Park is mentioned, which is related to but not exactly the same as Bethpage Black Course, so it receives a partial score. The High Line is completely missing from the AI answer. The AI also included Washington Square Park, which is not in the groundtruth.'
> > > > > - *Entity*:
> > > > >   + Raw answer: 'The protagonists involved in events related to Burlesque Performance are:\n\n1. Julian Ross\n2. Samuel Parker\n3. Isaiah (Last name not provided)\n4. Sofia Ramos'
> > > > >   + Scoring relevance against each ground truth item: [{'Isaiah Bennett': 0.5}, {'Sofia Ramos': 1}, {'Julian Ross': 1}]
> > > > >   + Explanation of the evaluation: "The AI-generated answer correctly identified Julian Ross and Sofia Ramos, earning a score of 1 for each. For Isaiah Bennett, only the first name 'Isaiah' was mentioned without a last name, so it receives a score of 0.5. The AI answer also included Samuel Parker, who is not in the ground truth list. Overall, the AI answer captured most of the correct entities but added an extra person and missed the full name for Isaiah Bennett."
> > > > > - *Content*:
> > > > >   + Raw answer: 'At One World Trade Center, several distinct events took place, each with its own unique focus and set of occurrences: 1. Tech Hackathon: (...) . 2. Photography Exhibition: (...). 3. Astronomy Show: (...) . 4. Hackathon: (...) . 5. Peer Review Workshop: (...) . 6. Fashion Show: (...).'
> > > > >   + Scoring relevance against each ground truth item: [{'Photography Exhibition': 1},  {'Fashion Show': 1}, {'Ice Sculpture Exhibition': 0},  {'Theater Performance': 0},  {'Tattoo Convention': 0},  {'Carnival': 0},  {'Tech Hackathon': 1},  {'Educational Workshop': 0.5},  {'Murder Mystery Dinner': 0},  {'Ghost Tour': 0},  {'Debate Tournament': 0},  {'Fire Dancing Performance': 0},  {'Classical Music Concert': 0}]
> > > > >   + Explanation of the evaluation: 'The AI-generated answer correctly identified 3 events from the ground truth: Tech Hackathon, Photography Exhibition, and Fashion Show. The Peer Review Workshop could be considered related to an Educational Workshop, hence the 0.5 score. The Astronomy Show is not in the ground truth. The answer missed several events from the ground truth(...)'

---

> > > > > > ### Author Response · Authors · 2024-11-22
> > > > > > **Answer to reviewer uWQ8 (Q4--Q7)**
> > > > > >
> > > > > > > Q4.In Table 3, the fine-tuned model performs well on single-event queries (F1=0.83) but poorly on multi-event queries (F1≤0.37). Could you elaborate on why naive fine-tuning fails to generalize beyond single-event memorization? What specific architectural or training modifications might address this limitation?
> > > > > >
> > > > > > > Q7.Have you tried other fine-tuning approaches beyond single-event memorization that might better capture the hierarchical and relational nature of episodic memory?
> > > > > >
> > > > > > "Why naive fine-tuning fails to generalize beyond single-event memorization?" is a terrific question, and solving this problem is, we believe, one of the most underrated open ones. Thank you for raising this very important point.
> > > > > >
> > > > > > Having in mind this [figure](https://figshare.com/s/863956f3e6592d3dad34?file=50683632) , the issue is the following: even though the model learns individual facts (e.g., "Jackson Ramos was in Central Park on September 22, 2026", "Jackson Ramos was in Ellis Island on April 09, 2026", "Jackson Ramos was in One World Trade Center on August 24, 2026", ...) and answers each fact correctly, the model cannot synthesize across chapters to build a complete picture of Jackson Ramos's movements through time and space to answer questions like "List all the places where Jackson Ramos was seen". *Our working hypothesis is that solving the problem (without training on all possible questions) might need an iterative search/retrieval where the model generates multiple places and dates that correspond to "Jackson Ramos" before synthesizing and answering the question*.
> > > > > >
> > > > > > Currently, we are not aware of any existing fine-tuning strategies specifically designed for episodic memory tasks, *hence the importance of our benchmark*. Our paper demonstrates that conventional fine-tuning approaches using question/answer pairs are inadequate for memory tasks (while fine-tuning typically works well for modifying style, tone, or learning new capabilities). One crucial research direction we advocate for is the development of methods to integrate new memories directly into model weights, rather than relying on context windows or external databases.
> > > > > > Our contribution to this research direction is the development of a systematic episodic memory benchmark with a comprehensive set of tasks, which can facilitate future work in this area.
> > > > > >
> > > > > >
> > > > > > > Q5.The gradient pattern in Figure 3 shows degrading performance from context to space to time cues. What specific aspects of temporal reasoning make it particularly challenging for current LLMs?
> > > > > >
> > > > > > That's also another great question. Our working hypothesis is that dates may share similarities from a token-level perspective, making them more difficult to distinguish compared to names and places. But we currently lack sufficient evidence to confirm this hypothesis. This is definitely an interesting avenue for future investigation.
> > > > > >
> > > > > > > Q6.For the "Latest state recall" results in Table 4, what specific challenges prevent models from achieving higher accuracy in tracking entity states over time?
> > > > > >
> > > > > > This is (again) another really great question, that our work is revealing and submitting to the community.
> > > > > >
> > > > > > Another reviewer suggested that temporal ordering might be a contributing factor, since the current generated books have a narrative structure that differs from their chronological sequence.
> > > > > > Nonetheless, to quantify the impact of temporal ordering, we conducted additional experiments using the short book where we replace the very same identical events, chapter content, and questions, chronologically in the book. We tested this version with gpt-4o, gpt-4o-mini, claude-3-5-sonnet, and claude-3-haiku models.
> > > > > > *In this case, we observe a consistent improvement for the majority of cells. We will supplement these findings with statistical analysis to demonstrate the significance of the results.*
> > > > > >
> > > > > > However,  humans are able to sort events, even when they don't receive them in the correct order (consider how we can effortlessly reconstruct sequences even from non-linear narratives like Memento or Pulp Fiction). Our working hypothesis is that this capability stems from a fundamental difference in how temporal information is processed: humans don't simply store events with timestamps, but rather dynamically integrate each new event into a coherent temporal framework, understanding its relationships to existing memories. This suggests that effective episodic memory requires not just information storage, but also sophisticated temporal integration mechanisms that current LLMs appear to lack.
> > > > > >
> > > > > > This finding highlights an important gap between human and LLM capabilities in temporal reasoning that merits further investigation. *We believe our benchmark has helped surface this fundamental challenge in AI systems' ability to handle episodic memory*, and we hope this will stimulate new research directions.

---

> > > > > > > ### Author Response · Authors · 2024-11-22
> > > > > > > **Answer to reviewer uWQ8 (Q8, Q9)**
> > > > > > >
> > > > > > > > Q8.Have you tried other retrieval strategies beyond cosine similarity? How do you address the challenge of retrieving coherent information when relevant context is distributed across multiple chunks?
> > > > > > >
> > > > > > > > 5.The RAG experiments in Section 5.1 use only basic paragraph-level chunking without exploring alternative strategies. The authors should investigate alternative chunking approaches, compare different retrieval mechanisms, and analyze how these choices impact episodic memory performance.
> > > > > > >
> > > > > > > We agree with the reviewer that exploring alternative retrieval strategies is valuable. However, our primary contribution is providing a comprehensive benchmark framework for episodic memory evaluation, rather than optimizing RAG performance.
> > > > > > >
> > > > > > > That said, we did conducted an ablation study comparing paragraph-level and chapter-level chunking (see Table 14 and accompanying discussion). This comparison is particularly informative because:
> > > > > > >
> > > > > > > 1. Chapter-level chunking represents an ideal upper bound for RAG performance in our setting, since each chapter contains by design all information about a single event
> > > > > > >
> > > > > > > 2. Paragraph-level chunking more realistically mirrors the challenges of real-world episodic memory tasks, where:
> > > > > > >    - Information about a single event is naturally distributed across multiple paragraphs
> > > > > > >    - For complex queries involving multiple events (e.g., 6 events), up to 24 paragraphs may contain critical information
> > > > > > >    - Retrieved chunks must be integrated to construct complete answers
> > > > > > >
> > > > > > > While additional retrieval strategies can be explored and evaluated using our benchmark, we emphasize that our primary contribution is providing a comprehensive framework that includes document generation, question-answer generation, and evaluation methodology against systematic tasks.
> > > > > > >
> > > > > > > > Q9.How might this benchmark contribute to developing novel training methodologies for episodic memory tasks in LLMs, beyond RAG, fine-tuning or parametric memory storage?
> > > > > > >
> > > > > > > This is another well-thought question. Beyond mere evaluation, we believe that our framework can scale to provide enough synthetically generated data that could be used for the post-training of LLMs to enhance their in-context abilities to reason about episodic events.

---

> > > > > > > > ### Author Response · Authors · 2024-11-22
> > > > > > > > **Answer to reviewer uWQ8 (finer-grain analysis)**
> > > > > > > >
> > > > > > > > > 6.While Table 4 shows poor performance in chronological ordering tasks, the paper doesn't provide detailed error analysis or investigate specific failure patterns. The analysis in Section 5.2 focuses on aggregate metrics without examining individual failure cases. The authors should provide detailed case studies of failure modes, analyze patterns in chronological ordering errors, and investigate whether specific temporal relationships consistently challenge the models.
> > > > > > > >
> > > > > > > > > 7.Although Section 5.2 mentions testing for hallucinations, the analysis is limited. The paper fails to examine when and why models confabulate, or how confabulation patterns vary across different model architectures and memory strategies. This could be improved by designing specific experiments to probe confabulation triggers and providing metrics for measuring confabulation severity.
> > > > > > > >
> > > > > > > > We agree that these analyses would provide better insights into model behavior. In response to another reviewer, we propose conducting a manual analysis of GPT-4's responses (on the long book) where zero events were matched, to illustrate the task's complexity.
> > > > > > > >
> > > > > > > > Of the 150 questions with 0 matching events, 24 (16%) produced incorrect answers. Notably, all incorrect predictions were still contextually relevant to the book's content.
> > > > > > > >
> > > > > > > > The 24 failed zero-event questions can be categorized into two types (see Table 11 in the appendix for details):
> > > > > > > >
> > > > > > > > 1. Inner questions (17 cases):
> > > > > > > >    - Questions constructed using elements present in the book
> > > > > > > >    - Majority (14/17) involve entity-based queries
> > > > > > > >
> > > > > > > > 2. Outer questions (7 cases):
> > > > > > > >    - Questions using at least one element from outside the book (sampled from the unused universe)
> > > > > > > >    - All involve temporal elements
> > > > > > > >    - Consistent cue patterns: (t,\*,\*,\*), (t,\*,\*,c), or (t,\*,ent,\*)
> > > > > > > >
> > > > > > > > Detailed analysis of the 7 outer questions (outer elements below are "August 24, 2024", "Chess Championship", and "Zoe Rivera"):
> > > > > > > >
> > > > > > > > 1. Three questions about "August 24, 2024" (date not in book):
> > > > > > > >    - Model fabricated answers using elements from different chapters with answers covering the locations (('One World Trade Center', 'American Museum of Natural History', 'Trinity Church'), the entities ('Scarlett Thomas', 'Julian Ross', 'Maya Smith', 'Mila Gonzalez') and the events content ('Storytelling Festival', 'Carnival', 'Murder Mystery Dinner'))
> > > > > > > >    - Upon examination, we found that the model combined a Storytelling Festival (actually in chapter 147 on Dec 25, 2025) featuring a Storytelling Festival at the American Museum of Natural History, with a Murder Mystery Dinner (actually in chapter 120 on Nov 13, 2026) at One World Trade Center with Scarlett Thomas.
> > > > > > > >
> > > > > > > > 2. One "Chess Championship" question (event not in book) for April 09, 2026:
> > > > > > > >    - Model showed explicit uncertainty in its response: "The events related to the Chess Championship on April 09, 2026, took place at the following locations: 1. High Line, 2. Lincoln Center (Note: The text does not explicitly mention a "Chess Championship" on April 09, 2026, but these locations match the date provided in the question. If the events do not align with the mentioned event, it might be necessary to re-evaluate the context for any additional details.)"
> > > > > > > >    - Verified: "chess" never appears in book
> > > > > > > >    - Date (April 09, 2026 ) exists but with different locations, including High Line but not Lincoln Center.
> > > > > > > >
> > > > > > > > 3. One "Charity Gala" question for April 09, 2026 (again event not in the book):
> > > > > > > >    - Model gave confident but incorrect answer: "The events related to the Charity Gala on April 09, 2026, took place at the following locations: 1. High Line 2. Lincoln Center. I hope this helps! Let me know if there is anything else you need."
> > > > > > > >    - Our ground truth shows the only High Line event on that date was an Astronomy Night.
> > > > > > > >
> > > > > > > > 4. Two questions about "Zoe Rivera" (entity not in book):
> > > > > > > >    - The chapters corresponding to the predicted answers contain no similar names (neither matching first nor last names).
> > > > > > > >
> > > > > > > > These examples highlight why a comprehensive automated analysis would require substantial effort, that we leave for future work.

---

> > > > > > > > > ### Comment · Reviewer_uWQ8 · 2024-11-28
> > > > > > > > > **Response to Authors**
> > > > > > > > >
> > > > > > > > > I appreciate the detailed responses to my questions and concerns. You have effectively addressed the main issues I raised, and the additional work you've done is quite impressive. Based on these improvements, I have updated my score accordingly.

---

### Official Review · Reviewer_YG9E · 2024-11-04

**Soundness:** 3
**Presentation:** 4
**Contribution:** 4
**Rating:** 5
**Confidence:** 3

**Summary:**

This paper introduces and explores the concept of episodic memory in the context of long-text comprehension by language models. This approach emphasizes the necessity for models to maintain a coherent understanding of an entity’s state as it evolves over time, space, and content. Both short and long synthetic documents were generated using various cue templates. The dataset also includes null answers incorporated to test for model illusions.

The findings indicate that models like GPT-4o with contextual memory and Claude 3.5 Sonnet4 with RAG memory scored the highest on average, suggesting that retrieval-based methods can improve situational memory by narrowing the context relevant to each query. While some models demonstrated near-perfect accuracy in chapters involving zero or one event per entity, their performance declined significantly as the number of events increased.

**Strengths:**

It is the first time to evaluate updated memories for episodic memory, events with rich contextual information and involving the tracking of specific entities occurring at specific time and spatial locations. And have a solid analysis of event complexity.

The drop of accuracy of fine-tuned model highlights that current fine-tuning techniques fall short in understanding episodic events and their complex interrelationships. All models have a low percentage of exact matches in the temporal ordering task, and even when the models retrieve the correct events, they often fail to order the events correctly.

**Weaknesses:**

1. line 123 has different citation format

**Questions:**

It would be beneficial to explore whether different fine-tuning parameters, or fine-tuning applied to other models could enhance the performance of episodic memory tasks

---

> ### Author Response · Authors · 2024-11-22
> **Answer to reviewer YG9E**
>
> We appreciate the strengths provided by the reviewer, and corrected the citation format in line 123. We would like to first comment on the summary provided, then answer to the question.
>
> > While some models demonstrated near-perfect accuracy in chapters involving zero or one event per entity, their performance declined significantly as the number of events increased.
>
> We'd like to clarify the document generation process. In our framework, each chapter corresponds to a single defined event with known time, location, entity, and event content. While the events are generated independently, they all exist within the same static universe. To illustrate this, we provide an example tracking a single entity (Jackson Ramos, shown with red segments) across chapters (with other entities shown as gray segments) in the default 200-chapter book.
>
> [Illustration of an entity tracking example](https://figshare.com/s/863956f3e6592d3dad34?file=50682921)
>
> In this example, the question "at which date did events involving Jackson Ramos occur" links to 5 distinct events, yielding 5 dates. To evaluate recall performance, we assess the accuracy of answers to questions associated with varying numbers of events (ranging from 0 to 6+), while maintaining that each chapter corresponds to one ground truth event.
>
> We hope this clarifies the structure of our benchmark
>
> > It would be beneficial to explore whether different fine-tuning parameters, or fine-tuning applied to other models could enhance the performance of episodic memory tasks
>
> This is an excellent suggestion that touches on a fundamental challenge we uncovered in our work. Our experiments reveal an interesting phenomenon: even though models can learn individual facts through fine-tuning (e.g., "Jackson Ramos attended a jazz concert in Central Park on September 22, 2026", "Jackson Ramos gave a photography workshop at Ellis Island on April 09, 2026", "Jackson Ramos led a business meeting at One World Trade Center on August 24, 2026"), they struggle to synthesize information across chapters to build a complete picture (e.g., tracking Jackson's progression from teaching photography to attending cultural events to conducting business meetings across New York City over time).
>
> We are not aware of any fine-tuning strategy that can lead the models to perform such an integration.
> Finding such novel finetuning or learning strategies is a great challenge that we submit with this work to the community. Our episodic memory benchmark is a first step towards that goal.  and we demonstrate in this paper that direct fine-tuning with question/answer pairs is inadequate.
>
> > **Weaknesses:** : line 123 has different citation format
>
> Given that all evaluation factors are rated as good or excellent (3/4 for soundness, 4/4 for presentation and contribution), we would appreciate any additional feedback about remaining concerns that led to the "marginally below acceptance threshold" rating. This would help us better understand how to strengthen our work for future versions.

---

### Official Review · Reviewer_PJrN · 2024-11-04

**Soundness:** 4
**Presentation:** 3
**Contribution:** 3
**Rating:** 8
**Confidence:** 3

**Summary:**

The paper presents a framework for modeling and evaluating episodic memory in large language models (LLMs), focusing on their ability to recall and process events associated with specific times and locations, similar to human episodic memory. The authors propose a method that uses entities and events to construct episodic memories and a benchmark designed to test LLMs on tasks such as recalling event details, tracking entity states, and understanding temporal-spatial contexts. The benchmark, including synthetic datasets and structured tasks, also assesses the models' ability to avoid confabulations by identifying unfamiliar information. The authors' evaluation of models like GPT-4 and Claude shows that current LLMs struggle with complex, multi-event scenarios and spatio-temporal reasoning, highlighting the need for improved episodic memory frameworks and training methods tailored to these capabilities.

**Strengths:**

1. The paper effectively organizes episodic memory tasks for LLMs using a cue-based recall and retrieval method. The authors provide examples of different cues, showing various combinations that models must use to retrieve event information based on time, location, involved entities, or content. This clear design demonstrates a solid grasp of how to simulate episodic memory in LLMs and offers a strong foundation for evaluating model recall across a range of scenarios, from simple to complex.
2. The benchmark tests the model's ability to handle both clear and vague questions, similar to real-world situations where memory needs vary. By asking models to either recall a specific event or recognize several related events, the tasks assess how well the models adapt to different recall demands.
3. The benchmark includes carefully designed tests to assess a model's ability to recognize unfamiliar events or entities and admit when it lacks information. This is crucial for evaluating whether LLMs can avoid hallucinations, and this thoughtful design adds reliability to the benchmark.
4. The paper offers detailed statistics and information about the benchmark, along with several ablation studies in the appendix. This level of detail shows the authors' commitment to transparency and rigor, helping readers understand the benchmark’s structure and how different elements affect model performance. These ablation studies also provide deeper insights and useful guidance for future research.

**Weaknesses:**

1. A limitation of the paper is that it only evaluates proprietary models like GPT-4o and Claude, rather than open-source models such as LLaMA 3. Including open-source models would make the findings more generalizable and accessible to a broader research community, enabling comparisons across a wider range of models and methods.
2. The benchmark mainly uses clear cues, which, while providing consistency and control, may not capture the subtler cues common in natural language memory tasks. Adding more ambiguous time markers and indirect references could better simulate real-world memory challenges and lead to a stronger test of models' episodic memory abilities and their handling of less obvious retrieval cues.

**Questions:**

1. Could you explain the "naive fine-tuning" approach mentioned in the paper? What datasets and methods were used, and how does this approach differ from other fine-tuning strategies for episodic memory tasks?

2. Just curious—does your benchmark have a specific name, or is it simply called "Short Book" and "Long Book"?

---

> ### Author Response · Authors · 2024-11-22
> **Answer to reviewer PJrN**
>
> We thank the reviewer for the validation of the benchmark design and realization. We comment in the next paragraphs the different explicited weaknesses, and provide answers to the questions.
>
> > A limitation of the paper is that it only evaluates proprietary models like GPT-4o and Claude, rather than open-source models such as LLaMA 3. Including open-source models would make the findings more generalizable and accessible to a broader research community, enabling comparisons across a wider range of models and methods.
>
> We agree with the reviewer and plan to add the evaluation with one or several LLaMA 3.1 models. These models have a context window of 128k (contrary to LLaMA 3 that is limited to 7k). ~~We attempted to apply the LLaMA 3.1-405B model using a cloud API, but the service was still limiting the input token to 7k, preventing the application of our benchmarks on the small and large books. We welcome any suggestion of cloud API services that can effectively manage a 128k context window. To demonstrate our code's ability to integrate new models, we tested LLaMA 3 with an even smaller book (including only 10 chapters instead of 20 or 200), and these results will be uploaded in the reproducibility section.~~ During the rebuttal period, we have evaluated llama-3.1-405b-instruct and llama-3.2-3b-instruct on the default short book (for the camera ready version, we plan to add the results on the default long book with llama-3.1-405b too). The results are as follows (* for new experiments; adding the other models for reference):
>
> | Memory | Model  | 0 (150) | 1 (150) | 2 (48) | 3-5 (18) |
> |--|--|--|--|--|-|
> |in-context| llama-3.1-405b-instruct*| 0.91±0.28 | 0.95±0.18 | 0.89±0.18 | 0.83±0.17 |
> |in-context| llama-3.2-3b-instruct*| 0.75±0.43 | 0.38±0.47 | 0.34±0.34 | 0.48±0.33 |
> |in-context| gpt-4o-mini| 0.53±0.50 | 0.92±0.23 | 0.87±0.21 | 0.89±0.16 |
> |in-context| gpt-4o| 0.86±0.35 | 0.96±0.16 | 0.93±0.16 | 0.88±0.16 |
> |in-context| claude-3-haiku| 0.81±0.39 | 0.74±0.43 | 0.59±0.31 | 0.65±0.20 |
> |in-context| claude-3-5-sonnet| 0.98±0.14 | 0.94±0.23 | 0.73±0.22 | 0.73±0.20 |
> |in-context| o1-mini| 0.97±0.16 | 0.94±0.21 | 0.90±0.18 | 0.93±0.11 |
>
> The llama-3.1-405b model performs comparably to GPT4o and outperforms Claude 3.5 Sonnet specifically when multiple events match the given cue (pending significance testing). However, the smaller llama-3.2-3b-instruct model underperforms, occasionally producing lengthy, irrelevant responses. The reproducibility notebook can be found in the main message.
>
> > The benchmark mainly uses clear cues, which, while providing consistency and control, may not capture the subtler cues common in natural language memory tasks. Adding more ambiguous time markers and indirect references could better simulate real-world memory challenges and lead to a stronger test of models' episodic memory abilities and their handling of less obvious retrieval cues.
>
> We fully agree with the reviewer and we actually even believe that it is an exciting direction for future work, i.e. to probe close locations and close dates.
>
> > Could you explain the "naive fine-tuning" approach mentioned in the paper? What datasets and methods were used, and how does this approach differ from other fine-tuning strategies for episodic memory tasks?
>
> Our naive fine-tuning experiment aims to incorporate the essential information needed to generalize answers across all benchmark questions. The question/answer pairs linked to individual events establish basic facts like "entity i was in location j at date k doing l" (for all items in the book), which enables deducing answers to questions involving multiple events, such as "where has entity i been seen?"
>
> For the fine-tuning process, we selected all 3,199 questions, each tied to one specific chapter (we cover all possible questions about each chapter). This set of 3,199 question/answer pairs forms our training dataset. We utilized the standard fine-tuning method provided by the OpenAI API.
>
> We are not aware of existing fine-tuning strategies specifically designed for episodic memory tasks, and our results demonstrate that direct fine-tuning with question/answer pairs is inadequate (while fine-tuning typically succeeds in modifying style, tone, or learning new tasks, we show it is not directly suitable for memory retention). Integrating new memories directly into model weights (rather than relying on context or external databases) represents one of the key research directions we propose for future work.
>
> > Just curious—does your benchmark have a specific name, or is it simply called "Short Book" and "Long Book"?
>
> We asked an LLM to suggest a name and title that capture the story's essence. The suggested title was "Synaptic Echoes 2026: The Neuro-Temporal Paradox of Episodic Precognition" (shown in Listing 17 in the appendix). We propose using "Synaptic Echoes" for the short version and "Synaptic Echoes (long)" for the extended version.
>
> Edit: adding llama3 results on the short book

---

> > ### Comment · Reviewer_PJrN · 2024-11-26
> > **Response to Authors**
> >
> > I appreciate the detailed responses to my questions and concerns. Have you integrated these new results and the other changes you mentioned into a new manuscript? If so, please upload it and signify your change in the global response.
> >
> > I have raised the rating based on the overall quality and your responses ~

---

### Official Review · Reviewer_prP7 · 2024-11-04

**Soundness:** 3
**Presentation:** 3
**Contribution:** 2
**Rating:** 8
**Confidence:** 3

**Summary:**

This paper proposes an approach for generating a benchmark for evaluating the episodic memory of LLMs. The authors use this generation approach to develop short book and long book splits consisting of 456 and 686 Q/A pairs respectively. The authors then evaluate high quality LLMs on this benchmark to showcase that even with RAG achieving high performance can be quite challenging.

**Strengths:**

I think this paper looks at an important problem and is well written for the most part. The approach outlined in section 3 makes a lot of sense. I really agree about the five points mentioned in the "Need for an episodic memory benchmark" paragraph at the end of page 3. I also found that Figure 3 and Table 4 presented some interesting analysis giving more insight into the ways that current LLMs struggle with reasoning over episodic memories.

**Weaknesses:**

**Data Generated:** I think the biggest weakness of this paper is the quantity and quality of data generated as part of the short book and long book splits. This first stood out to me when reading Table 2. For this amount of content, it feels like even the use of automated means are not necessary i.e. a human can annotate an actual book with questions and it would be a lot higher quality than what is produced here. Given that the authors are specifying a general strategy for generating benchmarks in section 3, I thought at least a number of randomly generated books would be considered if not also significant diversity within these books.  As a result, this benchmark does not yield easy high confidence analysis, which is showcased by massive error bars throughout the main results table (table 3).

**Table 3:** The results of the in-context and RAG models are largely in-line with general expectations in Table 3, so the benchmark does not really lead to new findings in comparison to the current literature. The only interesting finding is with respect to the fine-tuning baseline, but the implementation of this baseline seems flawed. First of all, there is barely enough data for this dataset to be used for evaluation of LLMs, it seems like there is simply not the data required to facilitate fine-tuning. Judging from the appendix, it seems like for some reason only a number of events matching the cues of 1 was used for fine-tuning, which seems to fully explain the results in this row on its own. It is not even clear to me if the 0.83$\pm$0.35 is using the same data for training and testing.

**Questions:**

Q1: When the authors write: "The proposed episodic memory benchmark exhibits several desirable properties: it is contamination-free by design, scalable with low human labor, offers unambiguous cues and ground truth, and the ability to model multiple cues and events within a synthetic yet realistic narrative."  What does scalable mean here? How is this demonstrated in the paper?

Q2: The authors write that needle in the haystack benchmarks "do not incorporate temporal nor spatial awareness", but isn't this point undermined by the limitation related to "event independence" the authors mention?

Q3: The authors also write that bABI / bABILong "often involve highly artificial scenarios lacking complexity and realism – opening the door to shortcut reasoning by exploiting dataset biases or patterns", but isn't this point undermined by the limitation related to "temporal representation" the authors mention? Also how are dataset biases / patterns addressed in a way that goes beyond bABI?

Q4: For the point on "limited domain scope" could you explain why more domains or even random variations of the book were not considered in this paper? What roadblocks remain that made the authors position it as future work?

---

> ### Author Response · Authors · 2024-11-21
> **Answer to reviewer prP7 (1/5)**
>
> We would like to first thank the reviewer for the constructive feedback on our paper. We appreciate your recognition of the importance of the research problem and of the positive comments about our approach.
>
> In the following, we answer the different questions by providing clarification and presenting additional experimental materials and results.
>
> ## Data generated
>
> > I think the biggest weakness of this paper is the quantity and quality of data generated as part of the short book and long book splits. This first stood out to me when reading Table 2. For this amount of content, it feels like even the use of automated means are not necessary i.e. a human can annotate an actual book with questions and it would be a lot higher quality than what is produced here.
>
> For this question, it would help to know the following. Does the perceived low quality of the book come from the assumption that humans can do a better job at (i) generating events/chapters? or instead at (ii) annotating existing books? Knowing this would help us better answer the question.
>
> That being said, our methodology offers unique advantages that human annotation, or even generation, cannot match easily:
>
> - Control and contamination: we generate contamination-free books with deterministic ground truth events, avoiding ambiguity that could exist in annotating real books (real books may be in LLM training data).
>
> - Systematic task design: we are the first to design a comprehensive episodic memory evaluation based on systematic (time, space, entity, details) cue-based recall tasks, grounded in cognitive science principles. This systematic design could, in principle, also benefit human annotated or even human-generated efforts.
>
> - Fine-grained control: our method further enables precise control over (i) the distribution of recurring entities, locations, dates and event content, (ii) question difficulty based on cue precision/specificity and the number of chapters needed for answer retrieval (see for instance the table 12 in the appendix showing the "widespreadness" of the selected questions, that would be difficult to obtain from a human, and prone to error).
>
> - Verifiable ground truth: unlike human annotation which may have subjective interpretations, our generated content has unambiguous ground truth since we inject, generate and verify the events.
>
> - Scalability: finally, a major strength of our synthetic approach is its *scalability*. We can generate benchmarks (book and question/answer pairs) of increasing length. To demonstrate it in this rebuttal, we further generate multiple books from different universes, even producing a 1-million-token book. Such level of scalability would be difficult to achieve with human annotation due to the sheer volume of content and the variability in narrative structures across different books.
>
> If this does not answer the question, it would be helpful if the reviewer could clarify which aspect (e.g. generation vs annotation) they find problematic to better address their concerns.
>
> > Given that the authors are specifying a general strategy for generating benchmarks in section 3, I thought at least a number of randomly generated books would be considered if not also significant diversity within these books.
>
> We agree with this comment. Our framework is indeed designed to generate diverse books by simply updating the universe components (dates, locations, entities, and events, as detailed in appendix B.1.1). To demonstrate this capability, we have now generated additional books* including:
> - World news collections (synthetic fictional news chapters)
> - Science fiction books (chapters set on different planets/moons in year 2200)
>
> While our initial evaluation focused on four similar books ({Claude, GPT4o} × {20, 200 chapters}), this was primarily driven by:
> - API cost considerations
> - The observation that models already struggle with 10k and 100k tokens, making larger contexts less informative at this stage
>
> For the revision, we will evaluate these new diverse books using GPT-4o (our best performing model per Figure 2) to explicitly demonstrate generalization across different domains. We already obtained the performance on recall tasks for the short books, as shown below (first row indicates the number of events matching the cues, with the count of questions between parentheses for respectively the default, the news, and the scifi books).
>
> | Memory | Model | Book | 0 (150) | 1 (150) | 2 (48, 33, 44) | 3-5 (18, 27, 12) |
> |--------|-------|---------|---------|---------|--------|----------|
> | in-context | gpt-4o | default | 0.86±0.35 | 0.96±0.16 | 0.93±0.16 | 0.88±0.16 |
> | in-context | gpt-4o | news | 0.91±0.29 | 0.99±0.06 | 0.89±0.18 | 0.86±0.12 |
> | in-context | gpt-4o | scifi | 0.85±0.36 | 0.99±0.06 | 0.94±0.14 | 0.92±0.15 |
>
> *One excerpt chapter is available in the common paragraph "Illustration of a single world news fictional chapter". Links to the complete books will be available in the reproducibility paragraph

---

> > ### Author Response · Authors · 2024-11-21
> > **Answer to reviewer prP7 (2/5)**
> >
> > > As a result, this benchmark does not yield easy high confidence analysis, which is showcased by massive error bars throughout the main results table (table 3).
> >
> > Thank you for raising this important point about the error bars in Table 3. We want to clarify that these *represent the standard deviation of the F1-score distribution across questions, not confidence intervals in estimating the mean value*. Therefore, *large standard deviations here indicate inherent variability in model performance across different questions, not uncertainty in our measurements*.
> >
> > Let's illustrate this with the first column of Table 3 (questions testing hallucination with no valid answers):
> > - For each question, the F1-score is binary: 1 if the model correctly indicates no answer exists, 0 if it hallucinates
> > - With an observed mean performance p=0.84 for GPT-4o, the standard deviation is mathematically bound to be sqrt(p*(1-p))=0.37, similar to a Bernoulli distribution
> > - Adding more questions would not reduce this standard deviation, as it reflects the inherent variability in model performance
> >
> > The standard deviations actually provide useful insights:
> > - Smaller values (e.g., for 6+ matching events) indicate more consistent model behavior
> > - Larger values suggest the model's performance varies significantly depending on the specific question
> >
> > That said, we agree that adding more questions from diverse books would strengthen our analysis by:
> > 1. Better assessing generalization across different domains
> > 2. Increasing statistical power to differentiate between models (e.g., in Fig. 2, some model pairs like GPT-4o and Claude-3.5-sonnet(RAG) cannot be statistically separated)"
> >
> > > Q1: When the authors write: "The proposed episodic memory benchmark exhibits several desirable properties: it is contamination-free by design, scalable with low human labor, offers unambiguous cues and ground truth, and the ability to model multiple cues and events within a synthetic yet realistic narrative." What does scalable mean here? How is this demonstrated in the paper?
> >
> > 'Scalable' in our framework refers to three complementary aspects:
> >
> > 1. Controlled generation of ground truth:
> > - Events follow our t,s,e,c (time, space, entity, content) structure
> > - Distribution controlled via geometric sampling across universe components
> > - Ground truth remains deterministic and verifiable regardless of scale
> >
> > 2. Systematic question-answer generation:
> > - Questions probe all combinations of episodic memory cues
> > - The difficulty is controlled through cue precision and number of relevant events (0 to 6+)
> > - Automated coverage that is impossible to match through human annotation
> >
> > 3. Automated quality assurance:
> > - Verification procedures detailed in Appendix B
> > - Enforces time-space and time-entity uniqueness constraints
> > - Validates event meta-data requirements and information placement
> >
> > To demonstrate these abilities, we generate a 2000-chapter book (1M+ tokens), for which we select 600+ question-answer pairs. We will upload the additional experiments in our anonymous figshare link.
> >
> > Given the current available context windows (advertised 128k GPT-4o, and 200k Claude) and evaluation costs (~$1500-1800 per model), we do not test the models on this large book.
> >
> > However, as the models expand, our framework can readily generate larger benchmarks to stress-test improved capabilities.

---

> > > ### Author Response · Authors · 2024-11-21
> > > **Answer to reviewer prP7 (3/5)**
> > >
> > > ## Details of the fine-tuning experiment and motivation
> > >
> > > > Table 3: The results of the in-context and RAG models are largely in-line with general expectations in Table 3, so the benchmark does not really lead to new findings in comparison to the current literature. The only interesting finding is with respect to the fine-tuning baseline, but the implementation of this baseline seems flawed. First of all, there is barely enough data for this dataset to be used for evaluation of LLMs, it seems like there is simply not the data required to facilitate fine-tuning. Judging from the appendix, it seems like for some reason only a number of events matching the cues of 1 was used for fine-tuning, which seems to fully explain the results in this row on its own. It is not even clear to me if the 0.83±0.35 is using the same data for training and testing.
> > >
> > > Thank you. We realize that our fine-tuning methodology requires clarification. For this, the structure of our benchmark is key:
> > >
> > > 1. **Book Structure**: Each chapter corresponds to exactly one event, characterized by a (t,s,e,c) tuple:
> > >    - t: a specific time (e.g., "September 22, 2026")
> > >    - s: a location (e.g., "Central Park")
> > >    - e: a main entity (e.g., "Jackson Ramos")
> > >    - c: an event content (e.g., "carnival")
> > >
> > > 2. **Question Types and Event Coverage**:
> > >    - Single-event questions probe one specific chapter/event (e.g., "Where was Jackson Ramos on September 22, 2026?")
> > >    - Multi-event questions require synthesizing information across chapters. For example, "List all places where Jackson Ramos was seen" requires finding and combining information from multiple chapters:
> > >      * Chapter 18: "Jackson Ramos was at High Line on February 27, 2026"
> > >      * Chapter 96: "Jackson Ramos was in Ellis Island on April 09, 2026"
> > >      * Chapter 112: "Jackson Ramos was in Snug Harbor Cultural Center on June 14, 2025"
> > >      * Chapter 163: "Jackson Ramos was in Central Park on September 22, 2026"
> > >      * Chapter 183: "Jackson Ramos was in One World Trade Center on August 24, 2026"
> > >
> > > 3. **Fine-tuning Experiment**:
> > >    - Training data in the fine-tuning experiment: 3,199 single-event questions, each tied to one specific chapter (we cover all possible questions about each chapter)
> > >    - Testing data: 686 questions, including 180 single-event questions (as indicated in Table 2). The 180 single-event questions are included into the 3,199 single-event questions. While the model succeeds on single-event questions (F1=0.83), it fails on multi-event questions (F1≤0.37)
> > >    - This reveals a critical limitation: even though the model learns individual facts (e.g., "Jackson Ramos was in Central Park on September 22, 2026", "Jackson Ramos was in Ellis Island on April 09, 2026", "Jackson Ramos was in One World Trade Center on August 24, 2026", ...), it cannot synthesize across chapters to build a complete picture of Jackson Ramos's movements through time and space to answer questions like "List all the places where Jackson Ramos was seen"
> > >
> > > Please refer to this [figure](https://figshare.com/s/863956f3e6592d3dad34?file=50683632) to visualize the training data in the finetuning experiment.
> > >
> > > The key insight isn't about data insufficiency, but rather that naïve fine-tuning fails to induce the ability to reason across multiple events - a fundamental aspect of episodic memory. Even though all necessary information exists in the atomic facts learned during training, the model cannot combine these facts to answer questions requiring temporal or spatial synthesis across multiple chapters/events.
> > >
> > > This clarifies why our fine-tuning experiment, while simple, reveals an important limitation in current approaches to integrating episodic memory capabilities in LLMs.
> > >
> > > We are adding an end-to-end figure explaining our benchmark creation and we will enhance the writing to better explain the finetuning experiment.

---

> > > > ### Author Response · Authors · 2024-11-21
> > > > **Answer to reviewer prP7 (4/5)**
> > > >
> > > > ## Other questions
> > > >
> > > > > Q2: The authors write that needle in the haystack benchmarks "do not incorporate temporal nor spatial awareness", but isn't this point undermined by the limitation related to "event independence" the authors mention?
> > > >
> > > > Our framework differs fundamentally from needle-in-haystack benchmarks in how it tests temporal and spatial awareness, even with independent events:
> > > >
> > > > 1. **Shared universe structure**: while our events are generated independently, they exist within a shared universe with:
> > > >    - Common set of entities (e.g., "Jackson Ramos")
> > > >    - Common set of locations (within New York in our default book)
> > > >    - Coherent timeline
> > > >    - This allows tracking entities across space and time, even without causal links
> > > >
> > > > 2. **Beyond simple retrieval**: consider a question like "When was Jackson Ramos seen at Central Park?":
> > > >    - Needle-in-haystack: Simply find a single piece of information
> > > >    - Our benchmark:
> > > >      * Must track Jackson's appearances across multiple chapters
> > > >      * Identify which appearances occurred at Central Park
> > > >      * Synthesize multiple date/location pairs
> > > >      * Information may be spread across different paragraphs within chapters
> > > >
> > > > To better illustrate this, we will include the example representing the tracking of a single entity (here Jackson Ramos with red segments) over the chapters (other entities are the grayed segments), for the default book with 200 chapters (which takes place in New York). The corresponding figure is [available at this address](https://figshare.com/s/863956f3e6592d3dad34?file=50682921)
> > > >
> > > > So even without causality, our tasks require, (i) a form of temporal reasoning (tracking entities across different dates), (ii) a form of spatial reasoning, tracking movements between locations, (iii) entity state tracking (what an entity was doing at different times/places, last etc). All this needs to integrate, beyond retrieval, information cross chapters.
> > > >
> > > > While we acknowledge (in the paper) that adding causal links between events would strengthen the benchmark, the current design already provides significant advances over simple retrieval tasks.
> > > >
> > > > > Q3: The authors also write that bABI / bABILong "often involve highly artificial scenarios lacking complexity and realism – opening the door to shortcut reasoning by exploiting dataset biases or patterns", but isn't this point undermined by the limitation related to "temporal representation" the authors mention? Also how are dataset biases / patterns addressed in a way that goes beyond bABI?
> > > >
> > > > Thank you for this question. We believe there may be some misunderstanding about bABI's scope and design goals compared to our benchmark. First, bABI was designed in 2015 for evaluating basic reasoning capabilities in early neural networks. It uses extremely simplified language and artificial scenarios like "John picked up the apple. John went to the office. John dropped the apple." Our benchmark in contrasts evaluates complex episodic memory capabilities in modern LLMs through realistic narratives (as earlier showcased).
> > > >
> > > > In our benchmark, each chapter is written with a consistent narrative voice that gradually reveals information (information carefully spread across different pargaphs), while providing description of the surroundings and the atmosphere. This contrasts with both bABI (that only provides simple atomic statements without a narrative voice) and bABILong (that injects *completely irrelevant* information (e.g. Mary moved to the hallway and John went to the hallway) at different places inside a large book about e.g. software programming).
> > > >
> > > > Let's for example contrast a typical bABI example:
> > > > ```
> > > > Mary moved to the bathroom.
> > > > John went to the hallway.
> > > > Where is Mary? bathroom
> > > > ```
> > > >
> > > > With a pargraph from our benchmark (full chapter available in the common answer, paragraph Illustration of a single world news fictional chapter; other example available in Listing 10 of the draft):
> > > > ```
> > > > In a dramatic turn of events on May 11, 2026, Benjamin Green found himself documenting the rapid transformation of peaceful suburban streets into raging torrents of muddy water. The local meteorological station's emergency sirens blared through the rain-soaked air as Hamza Avila and Koa Berlin, emergency response coordinators, rushed to evacuate residents from the low-lying areas. Rising waters had already submerged vehicles to their windows, while the relentless downpour continued to intensify, creating treacherous conditions across the region.
> > > > ```
> > > >
> > > > The limitations we acknowledge are about making the benchmark even more challenging - they don't undermine its current significant advances over bABI's simplified approach.
> > > >
> > > > Finally, the same example and figure we provided in Q2 above can be used again: even if each chapter is independent (conditionally to the universe), the information within each chapter may be located in different paragraphs.

---

> > > > > ### Author Response · Authors · 2024-11-21
> > > > > **Answer to reviewer prP7 (5/5)**
> > > > >
> > > > > >> Q4: For the point on "limited domain scope" could you explain why more domains or even random variations of the book were not considered in this paper? What roadblocks remain that made the authors position it as future work?
> > > > >
> > > > > As explained earlier, we agree that demonstrating the capability in generating variations of the book was lacking. By updating the components of the universe, we now have added the more diverse 'world news' and 'science fiction' books. What blocks us from generating more of such books is mostly the cost of the APIs to generate and evaluate the books. For example, for the GPT-4o in-context, the ingestion of the 100k tokens necessitates $0.25 per question, summing to $150+ for evaluating all the 600+ questions (the price is higher for Sonnet 3.5 and o1-mini).
> > > > >
> > > > > Our plan is to evaluate each additional book with the gpt-4o model. We would appreciate your thoughts on whether this plan aligns with your expectations.
> > > > >
> > > > > Having said that, the limitation in the domain scope is however still valid when the domains are drastically different. It is thus our intention for future work to extend the benchmark to completely different domains such as software projects. The latter require slight modifications to our existing modelling since the definitions of time, space, entities and contents are drastically different. In software project, time can be the date of modification of the file, space can be folder location or location of the function within the code, entities could be either author of the modification or even the variables whose state changes, etc.

---

> > > > > > ### Comment · Reviewer_prP7 · 2024-11-28
> > > > > >
> > > > > > Thank you for the comprehensive response to the questions and concerns mentioned in my review. It helped me understand the contribution of various aspects of the paper i.e. with respect to the fine-tuning results and comparisons with bABI. I also really appreciate the new experiments focused on generating more content, which addresses my biggest concern about the paper. I feel like the paper is much stronger now after the revisions and have increased my score accordingly.

---

### Author Response · Authors · 2024-11-21
**Global answer summary**

Dear reviewers,

We greatly appreciate your detailed reviews and insightful feedback, which are helping us to significantly improve our work.
We answer in detail to each reviewer in the separate answers, while we provide below a summary of the new experiments performed along with the additional material created.

Sincerely,

The authors

## Generated books and question/answer pairs

- For assessing the impact of the temporal order (highlighted by reviewer 1gS7), we created the chronologically ordered version of the default short and long book,
- For demonstrating the capability in generating variations of the book (concern express by reviewer prP7), we added the more diverse 'world news' and 'science fiction' books, with short (20 chapters) and long (200 chapters) variation for each (one excerpt available below),
- For demonstrating the scalability of our approach, we generated the default book with 2000 chapters, for a total of 1M+ tokens.

In the following table, we show the existing and additional books (together with the related question/answer pairs) that have been generated. The additionally generated benchmarks are indicated with an asterisk *.

| chapters            | 20 | 200 | 2000 |
|---------------------|----|------|-----|
| Claude default         | ✔  |  ✔  | ✔*  |
| Claude default ordered | ✔* |  ✔* | ✘   |
| GPT default            | ✔  |  ✔  | ✘   |
| Claude world news      | ✔* |  ✔* | ✘   |
| Claude scifi           | ✔* |  ✔* | ✘   |

## Additional experiments for providing the answers

- For assessing whether evaluating only the book produced with Claude gives an unfair advantage in the evaluation (highlighted by reviewer 1gS7), we evaluated the short GPT default book on the four (gpt-4o-mini, gpt-4o, claude-3-haiku, claude-3-5-sonnet) models,
- We assessed the impact of the temporal order by evaluating the ordered version of the default short book on the four (gpt-4o-mini, gpt-4o, claude-3-haiku, claude-3-5-sonnet) models,
- We have evaluated the Claude world news and Claude scifi variations with gpt-4o on the short default book.
- ~~We tried to evaluate the benchmark on llama3, but we are facing some issues detailed to reviewer PJrN. We hope that those issues will be solved in order to add this model in the comparison~~ We have evaluated llama-3.1-405b and llama-3.2-3b on the short default book.

In the following tables, we show the additional experiments performed (indicated with an asterisk *).

- Ablations:
|book|gpt-4o-mini|gpt-4o|claude-3-haiku|claude-3-5-sonnet|
|---|---|---|---|---|
| Short Claude default ordered|✔*|✔*|✔*| ✔* |
| Short GPT default|✔*|✔*|✔*|✔*|
| Short Claude world news|✘|✔*|✘|✘|
| Short Claude scifi|✘|✔*|✘|✘|

- New model:
|book|llama-3.1-405b|llama-3.2-3b|
|---|---|---|
|Short Claude default|✔*|✔*|

## Additional experiments

- We evaluated the degree of realism of each produced event (concern express by reviewer 1gS7), and evaluated the difference of performance between the realistic and the non-realistic events,
- We analyzed manually the hallucinations observed in the gpt4o answers when there is 0 matching events

## Visual aids and examples

Please find at the following anonymous addresses:
- The [global flowchart of our generation process](https://figshare.com/s/863956f3e6592d3dad34?file=50683452)
- [An example of the journey of of a single entity within the default long book](https://figshare.com/s/863956f3e6592d3dad34?file=50682921) This example shows the tracking of a single entity (here Jackson Ramos with red segments) over the chapters (other entities are indicated with gray segments)
- [Detailed fine tuning explanation for building the training data in this setting](https://figshare.com/s/863956f3e6592d3dad34?file=50683632)

## For reproducibility

We provide the supplementary [generated benchmark data at this address](https://figshare.com/s/7b634effbf6a71ca722c), while the [following notebooks for reproducing the additional experiments are there](https://figshare.com/s/863956f3e6592d3dad34?file=50784417)

- rebuttal_ablation_on_news_and_scifi_books.ipynb (evaluating the world news and the scifi short books with gpt-4o)
- rebuttal_ablation_with_gpt_book.ipynb (applying the experiment on the GPT generated book)
- rebuttal_generating_book_variations.ipynb (building the world news, the scifi, and the very long default books)
- rebuttal_hallucinations_0_matching_events.ipynb (manual analysis of the hallucinations observed in the gpt-4o answers when there is 0 matching events)
- rebuttal_llama3.ipynb (evaluating the short default book with llama 3.1 405b and llama 3.2 3b)
- rebuttal_map.ipynb (illustration provided)
- rebuttal_ordered_books.ipynb (building and evaluation of the ordered book)
- rebuttal_realistic_partition_and_evaluation.ipynb (assess the degree of realism of each event and evaluation in the difference of performance between the realistic and the non-realistic events),

Edit: adding anonymous links and solving issue for llama3 model

---

> ### Author Response · Authors · 2024-11-21
> **Illustration of a single world news fictional chapter**
>
> ## Illustration of a single world news fictional chapter
>
> We finally provide below a single chapter from one of the additionally synthetically generated book named "world news", following our methodology. This generated chapter is fictitious, and is generated given the event 'May 11, 2026', 'New South Wales', 'Benjamin Green', 'flash flood emergency', with meta-data information being 2 paragraphs, with positions {'location': 2, 'date': 1, 'entity': 1, 'content': 2}.
>
> >In a dramatic turn of events on May 11, 2026, Benjamin Green found himself documenting the rapid transformation of peaceful suburban streets into raging torrents of muddy water. The local meteorological station's emergency sirens blared through the rain-soaked air as Hamza Avila and Koa Berlin, emergency response coordinators, rushed to evacuate residents from the low-lying areas. Rising waters had already submerged vehicles to their windows, while the relentless downpour continued to intensify, creating treacherous conditions across the region.
>
> >As the situation in New South Wales deteriorated, Benjamin witnessed a flash flood emergency that would later be described as unprecedented in its ferocity. Water levels rose at an alarming rate of nearly one meter per hour, prompting Emilia Hooks, a veteran emergency services spokesperson, to declare it a "catastrophic event." The flood's destructive force was evident as debris-laden waters crashed through streets, uprooting trees and damaging infrastructure. Local authorities reported that over 300 residents were evacuated to emergency shelters, while rescue teams conducted more than 50 water rescues throughout the affected areas. The disaster response teams continue to monitor the situation as meteorologists predict additional rainfall in the coming hours.

---

### Author Response · Authors · 2024-11-29
**Changes integrated to the manuscript**

Dear reviewers,

We sincerely thank you all for your thorough feedback throughout the review process. Thanks to your suggestions, we have substantially strengthened our work through methodology clarifications, dataset diversification, and extended evaluations and ablations, enriching our findings and significantly increasing the confidence in our earlier results.

Our improvements encompass three main areas:

## Additional benchmark datasets illustrating our framework's scalability

We are now releasing 11 benchmark datasets (Table 28), including:
- Short (10k tokens), long (100k tokens) and very long (1M+ tokens) Claude-generated books;
- Time-ordered versions of the short and long Claude-generated books;
- Short and long GPT-generated books;
- Two additional diverse universes, world news and sci-fi, that we use to generate additional books (characteristics in Table 26, examples of universe elements and excerpts of a chapter in Appendix G.1).

## Extended evaluation and ablations
We have conducted comprehensive evaluations leading to both novel findings, and increased confidence in our earlier results:
- Applied on both short and long books:
-
  + Evaluation on Llama 3.1 405B instruct, applied on the long book (Figures 3 and 4 and Tables 3 and 4 in the main section), and on the short book (Figure 7 and Table 13 in the appendix). Results show that on the short book, llama-3.1 performance is equivalent to gpt-4o and cl-3.5-sonnet, while on the long book, gpt-4o is still statistically better than both cl-3.5-sonnet and llama-3.1 (that are statistically equivalent). Summarizing table below.

|Model|Book|0|1|2|3-5|6+|
|-|-|-|-|-|-|-|
|llama-3.1|short default|0.91±0.28|0.95±0.18|0.89±0.18|0.83±0.17|n.a.|
|llama-3.1|long default|0.80±0.40|0.49±0.47|0.38±0.33|0.40±0.25|0.45±0.20|

-
  + Evaluation on the world news and the sci-fi books for the gpt-4o model (in Appendix G.2, table also reported below). The figures are inline with our previous universe, confirming a consistent performance decline for queries with two or more matching events.

|Model|Book|0|1|2|3-5|6+|
|-|-|-|-|-|-|-|
|gpt-4o|short default|0.86±0.35|0.96±0.16|0.93±0.16|0.88±0.16|n.a.|
|gpt-4o|short news|0.91±0.29|0.99±0.06|0.89±0.18|0.86±0.12|n.a.|
|gpt-4o|short sci-fi|0.85±0.36|0.99±0.06|0.94±0.14|0.92±0.15|n.a.|
|gpt-4o|long default|0.84±0.37|0.81±0.38|0.60±0.31|0.57±0.21|0.53±0.14|
|gpt-4o|long news|0.96±0.20|0.82±0.38|0.66±0.28|0.54±0.23|0.46±0.20|
|gpt-4o|long sci-fi|0.90±0.30|0.72±0.43|0.62±0.29|0.55±0.22|0.51±0.13|

- Applied on the short book only:
  + Comparative evaluation of Claude- vs GPT-generated short books on four models (gpt-4o-mini, gpt-4o, claude-3-haiku, claude-3-5-sonnet) in Appendix E.5.
  + Comparative evaluation of unordered vs ordered events, in the short books, on four models (gpt-4o-mini, gpt-4o, claude-3-haiku, claude-3-5-sonnet) in Appendix E.6.
  + Comparative evaluation on questions related to a set of realistic vs non-realistic events reported in Tab. 24 for four models (gpt-4o-mini, gpt-4o, claude-3-haiku, claude-3-5-sonnet) in Appendix E.7.

## Methodology clarifications
We have enhanced the paper's clarity through:
- [Flowchart of the book generation process (Fig. 2 in the main section), with an explicit illustration that an item can match many events](https://figshare.com/s/863956f3e6592d3dad34?file=50683452).
- [Illustration of the shared universe structure, with examples of entity tracking and question/answer pairs (Appendix C, and Figure 6)](https://figshare.com/s/863956f3e6592d3dad34?file=50682921).
- Clarification of book generation quality-control layers: (i) exact-match parsing for event requirements and (ii) LLM-based verification for geographical focus, temporal day, main character and main event (Section 4.1, Appendix B.1.6 and B.1.7).
- Evaluation clarifications on LLM-as-a-judge usage and F1-score methodology (Appendix B.3.2, B.4).
- Fine-tuning methodology details in Appendix B.2.5.
- Assessment of event realism (Appendix E.7) and analysis of GPT-4o's empty-answer responses (Appendix E.8).

These improvements significantly strengthen our empirical validation while ensuring full reproducibility for the community. Again, we appreciate the reviewers' guidance in helping us achieve these meaningful enhancements.

Sincerely,

The authors

---

### Meta-Review · Area_Chair_hYMB · 2024-12-20

**Metareview:**

This work introduces a novel, cognitively inspired episodic memory benchmark for LLMs. Episodic memory is a crucial ability for decision making. Even though there have been benchmarks evaluating the decision making abilities of LLMs, there has not been a benchmark that explicitly the episodic memory capacity of LLMs. The extensive evaluation of recent LLMs reveals the limited capacity of current LLMs on episodic memory tasks.

There were several concerns in the initial reviews: the quality of the generated data compared to human annotations, the limited scope of the benchmark, implementation of the finetuning and RAG baselines, the confidence of the results, lack of evaluation of open sourced LLMs, the lack of ambiguous time markers and indirect references, and the lack of detailed analysis of errors and model hallucination. The authors' responses adequately addressed the concerns.

Overall, the benchmark is a good addition to the existing evaluation suite of LLMs, highlighting an important but less explored aspect of LLMs. It could be particularly useful for diagnosing LLMs' limitations in decision making and storytelling. I do want to suggest a human baseline for the benchmark. The benchmark design is inspired by human cognition. Thus, it would be informative to know humans' performance on these tasks to indicate the human-model performance gap.

**Additional Comments On Reviewer Discussion:**

Most reviewers responded to the rebuttal. They were satisfied with the authors' response and raised their ratings accordingly.

---

### Decision · Program_Chairs · 2025-01-22

Accept (Poster)